# COUNTERFACTUAL FAIRNESS FOR PREDICTIONS USING GENERATIVE ADVERSARIAL NETWORKS

## ABSTRACT

Fairness in predictions is of direct importance in practice due to legal, ethical, and societal reasons. It is often achieved through counterfactual fairness, which ensures that the prediction for an individual is the same as that in a counterfactual world under a different sensitive attribute. However, achieving counterfactual fairness is challenging as counterfactuals are unobservable. In this paper, we develop a novel deep neural network called *Generative Counterfactual Fairness Network* (GCFN) for making predictions under counterfactual fairness. Specifically, we leverage a tailored generative adversarial network to directly learn the counterfactual distribution of the descendants of the sensitive attribute, which we then use to enforce fair predictions through a novel counterfactual mediator regularization. We further provide theoretical guarantees that our method is effective in ensuring the notion of counterfactual fairness. Thereby, our GCFN addresses key shortcomings of existing baselines that are based on inferring latent variables, yet which (a) are potentially correlated with the sensitive attributes and thus lead to bias, (b) have weak capability in constructing latent representations and thus low prediction performance, and (c) do not have theoretical guarantees. Across various experiments, our method achieves state-of-the-art performance. Using a real-world case study from recidivism prediction, we further demonstrate that our method makes meaningful predictions in practice.

## 1 INTRODUCTION

Fairness in machine learning is mandated for a large number of practical applications due to legal, ethical, and societal reasons (Angwin et al., 2016; Barocas & Selbst, 2016; De Arteaga et al., 2022; Feuerriegel et al., 2020; Kleinberg et al., 2019; von Zahn et al., 2022). Examples are predictions in credit lending or recidivism prediction, where fairness is mandated by anti-discrimination laws.

In this paper, we focus on the notion of **counterfactual fairness** (Kusner et al., 2017). The notion of counterfactual fairness has recently received significant attention (e.g., Abroshan et al., 2022; Garg et al., 2019; Grari et al., 2023; Kim et al., 2021; Kusner et al., 2017; Ma et al., 2023; Xu et al., 2019). One reason is that counterfactual fairness directly relates to legal terminology in that a prediction is fair towards an individual if the prediction does not change had the individual belonged to a different demographic group defined by some sensitive attribute (e.g., gender, race). However, ensuring counterfactual fairness is challenging as, in practice, counterfactuals are generally unobservable.

Prior works have developed methods for achieving counterfactual fairness in predictive tasks (see Sec. 2). Originally, Kusner et al. (2017) described a conceptual algorithm to achieve counterfactual fairness. Therein, the idea is to first estimate a set of latent (background) variables and then train a prediction model without using the sensitive attribute or its descendants. More recently, the conceptual algorithm has been extended through neural methods, where the latent variables are learned using variational autoencoders (VAEs) (Grari et al., 2023; Kim et al., 2021; Pfohl et al., 2019). However, these methods have key shortcomings: (a) the learned representation can be potentially correlated with the sensitive attributes, which thus leads to bias; (b) VAEs have weak capability in constructing latent representations, which leads to a low prediction performance; and (c) have no theoretical guarantees. We address the shortcomings (a), (b), and (c) in our proposed method through the theoretical properties of our counterfactual mediator regularization and our tailored GAN, respectively.

In this paper, we present a novel deep neural network called *Generative Counterfactual Fairness Network* (GCFN) for making predictions under counterfactual fairness. Our method leverages a tailored generative adversarial network to directly learn the counterfactual distribution of the descendants of the sensitive attribute. We then use the generated counterfactuals to enforce fair predictions through a novel *counterfactual mediator regularization*. We further provide theoretical guarantees that our method is effective in ensuring the notion of counterfactual fairness.

Overall, our **main contributions** are as follows:[1] (1) We propose a novel deep neural network for achieving counterfactual fairness in predictions, which addresses key limitations of existing baselines. (2) We further show that, if the counterfactual distribution is learned sufficiently well, our method is guaranteed to ensure counterfactual fairness. (3) We demonstrate that our GCFN achieves the state-of-the-art performance. We further provide a real-world case study of recidivism prediction to show that our method gives meaningful predictions in practice.

## 2 RELATED WORK

Several research streams are relevant to our work, and we briefly discuss them in the following: (1) fairness notions for predictions, (2) methods for counterfactual fairness, (3) generative models for fair synthetic datasets, and (4) generative models for estimating causal effects.

**Fairness notions for predictions:** Over the past years, the machine learning community has developed an extensive series of fairness notions for predictive tasks so that one can train unbiased machine learning models; see Appendix B for a detailed overview. In this paper, we focus on *counterfactual fairness* (Kusner et al., 2017), due to its relevance in practice (Barocas & Selbst, 2016; De Arteaga et al., 2022).

**Predictions under counterfactual fairness:** Originally, Kusner et al. (2017) introduced a conceptual algorithm to achieve predictions under counterfactual fairness. The idea is to first infer a set of latent background variables and subsequently train a prediction model using these inferred latent variables and non-descendants of sensitive attributes. However, the conceptual algorithm requires knowledge of the ground-truth structural causal model, which makes it impractical.

State-of-the-art approaches build upon the above idea but integrate neural learning techniques, typically by using VAEs. These are mCEVAE (Pfohl et al., 2019), DCEVAE (Kim et al., 2021), and ADVAE (Grari et al., 2023). In general, these methods proceed by first computing the posterior distribution of the latent variables, given the observational data and a prior on latent variables. Based on that, they compute the implied counterfactual distributions, which can either be utilized directly for predictive purposes or can act as a constraint incorporated within the training loss. Further details are in Appendix B. In sum, the methods in Grari et al. (2023); Kim et al. (2021); Pfohl et al. (2019) are our main baselines.

However, the above methods have three main *shortcomings*. (a) The inferred latent variables can be potentially correlated with sensitive attributes because some information from sensitive attributes can leak into latent variables. This could introduce *bias in the prediction*. (b) It is commonly assumed that the prior distribution of latent variables follows a standard Gaussian in VAEs. However, this might be inadequate for capturing complex distributions, potentially leading to imprecise approximations of counterfactual distributions and thus overall *low prediction performance*. (c) The methods have no theoretical guarantees. In particular, the latent variable $U$ in VAE-based methods is *not* identifiable.[2] In our method, we later address (a) through the theoretical properties of our counterfactual mediator regularization and (b) through our tailored GAN. To address (c), we further provide theoretical guarantees that our method is effective in ensuring the notion of counterfactual fairness.

---

[1]Codes are in the anonymous GitHub: `https://anonymous.4open.science/r/gcfn`. Codes will also be available to a public GitHub repository upon acceptance.

[2]In causal inference, "identifiability" refers to a mathematical condition that permits a causal quantity to be measured from observed data (Pearl, 2009). Importantly, identification is *different* from estimation because methods that act as heuristics may return estimates but they do not correspond to the true value. For the latter, see D'Amour (2019) where the authors provide several concerns that, if a latent variable is not unique, it is possible to have local minima, which leads to unsafe results in causal inference.

**Generating fair synthetic datasets:** A different literature stream has used generative models to create fair synthetic datasets (e.g., van Breugel et al., 2021; Xu et al., 2018; Rajabi & Garibay, 2022; Xu et al., 2019). Importantly, the objective here is *different* from ours, namely, predictions under counterfactual fairness. Still, one could potentially adapt these methods to our task by first learning a fair dataset and then training a prediction model. We thus later adapt the procedure for our evaluation. Several methods focus on fairness notions outside of counterfactual fairness and are thus *not* applicable. We are only aware of one method called CFGAN (Xu et al., 2019) which is aimed at counterfactual fairness, and we later use it as a baseline. While CFGAN also employs GANs, its architecture is vastly different from our method. Besides, CFGAN uses the GAN to generate entirely synthetic data, while we use the GAN to generate counterfactuals of the mediator (see Appendix B for details). Moreover, baselines based on fair synthetic datasets have crucial limitations for our task: they learn the predictions not from the original but from the transformed dataset, which leads to information loss and thus a low prediction performance.

**Deep generative models for estimating causal effects:** There are many papers that leverage generative adversarial networks and variational autoencoders to estimate causal effects from observational data (Kocaoglu et al., 2018; Louizos et al., 2017; Pawlowski et al., 2020; Yoon et al., 2018; Bica et al., 2020). We later borrow some ideas of modeling counterfactuals through deep generative models, yet we emphasize that those methods aim at estimating causal effects but *without* fairness considerations.

**Research gap:** Existing baselines that are based on inferring latent variables, can have the problems that latent variables are potentially correlated with the sensitive attributes and thus lead to bias and that have weak capability in constructing latent representations and thus low prediction performance. As a remedy, we develop a novel deep neural network called *Generative Counterfactual Fairness Network* (GCFN) that addresses those shortcomings. We directly learn the counterfactual distribution through a tailored generative adversarial network and enforce counterfactual fairness through a novel *counterfactual mediator regularization*. To the best of our knowledge, ours is the first neural method for counterfactual fair predictions with theoretical guarantees.

## 3 PROBLEM SETUP

**Notation:** Capital letters such as $X, A, M$ denote random variables and small letters $x, a, m$ denote their realizations from corresponding domains $\mathcal{X}, \mathcal{A}, \mathcal{M}$. Further, $\mathbb{P}(M) = \mathbb{P}_M$ is the probability distribution of $M$; $\mathbb{P}(M \mid A = a)$ is a conditional (observational) distribution; $\mathbb{P}(M_a)$ the interventional distribution on $M$ when setting $A$ to $a$; and $\mathbb{P}(M_{a'} \mid A = a, M = m)$ the counterfactual distribution of $M$ had $A$ been set to $a'$ given evidence $A = a$ and $M = m$.

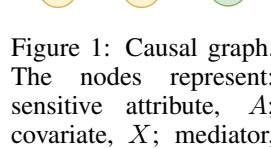

Our causal graph is shown in Fig 1, where the nodes represent: sensitive attribute $A \in \mathcal{A}$; mediators $M \in \mathcal{M}$, which are possibly causally influenced by the sensitive attribute; covariates $X \in \mathcal{X}$, which are not causally influenced by the sensitive attribute; and a target $Y \in \mathcal{Y}$. In our setting, $A$ can be a categorical variable with multiple categories $k$ and $X$ and $M$ can be multi-dimensional. For ease of notation, we use $k = 2$, i.e., $\mathcal{A} = \{0, 1\}$, to present our method below. Our method can be easily extended to scenarios where the sensitive attribute has multiple categories (see Appendix G.

Figure 1: Causal graph. The nodes represent: sensitive attribute, $A$; covariate, $X$; mediator, $M$; target, $Y$. $\longrightarrow$ represents direct causal effect; $\leftarrow \! - \! \rightarrow$ represents potential presence of hidden confounders.[4]

We use the potential outcomes framework (Rubin, 1974) to estimate causal quantities from observational data. Under our causal graph, the dependence of $M$ on $A$ implies that changes in the sensitive attribute $A$ mean also changes in the mediator $M$. We use subscripts such as $M_a$ to denote the potential outcome of $M$ when intervening on $A$. Similarly, $Y_a$ denotes the potential outcome of $Y$. Furthermore, for $k = 2$, $A$ is the factual, and $A'$ is the counterfactual outcome of the sensitive attribute.

---

[4]The dashed line allows for a correlation between $X$ and $A$ in our framework. Note that, if there is no dashed edge between $X$ and $A$, it is actually a stronger assumption, because it forbids the edge between $X$ and $A$ to have any hidden confounders. However, our setting is more general and allows for the existence of confounders (see Appendix H.1 for an experimental analysis).

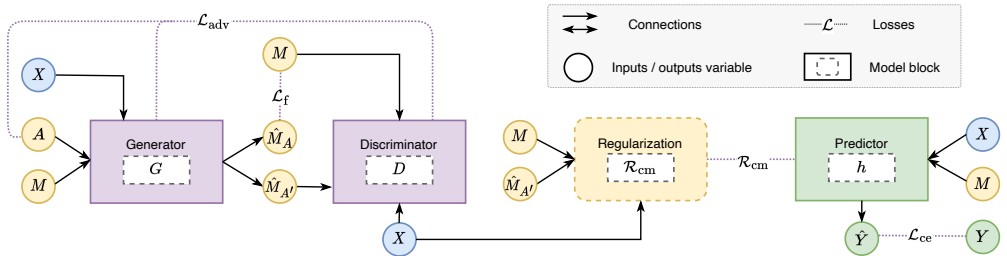

Figure 2: Overview of our GCFN for achieving counterfactual fairness in predictions. *Step 1:* The generator $G$ takes $(X, A, M)$ as input and outputs $\hat{M}_A$ and $\hat{M}_{A'}$. The discriminator $D$ differentiates the observed factual mediator $M$ from the generated counterfactual mediator $\hat{M}_{A'}$. *Step 2:* We then use generated counterfactual mediator $\hat{M}_{A'}$ in our counterfactual mediator regularization $\mathcal{R}_{\text{cm}}$. The counterfactual mediator regularization $\mathcal{R}_{\text{cm}}$ enforces the prediction model $h$ to be counterfactual fairness.

Our model follows standard assumptions necessary to identify causal queries (Rubin, 1974). (1) *Consistency:* The observed mediator is identical to the potential mediator given a certain sensitive attribute. Formally, for each unit of observation, $A = a \Rightarrow M = M_a$. (2) *Overlap:* For all $x$ such that $\mathbb{P}(X = x) > 0$, we have $0 < \mathbb{P}(A = a \mid X = x) < 1$, $\forall a \in \mathcal{A}$. (3) *Unconfoundedness:* Conditional on covariates $X$, the potential outcome $M_a$ is independent of sensitive attribute $A$, i.e. $M_a \perp\!\!\!\perp A \mid X$. We discuss the theoretical guarantee on identifiablity of counterfactuals under bijective generation mechanisms (BGMs) (Nasr-Esfahany et al., 2023; Melnychuk et al., 2023) in Appendix C.

**Objective:** In this paper, we aim to learn the prediction of a target $Y$ to be *counterfactual fair* with respect to some given sensitive attribute $A$ so that it thus fulfills the notion of *counterfactual fairness* (Kusner et al., 2017). Let $h(X, M) = \hat{Y}$ denote the predicted target from some prediction model, which only depends on covariates and mediators. Formally, our goal is to have $h$ achieve counterfactual fairness if under any context $X = x$, $A = a$, and $M = m$, that is,

$$\mathbb{P}\left(h(x, M_a) \mid X = x, A = a, M = m\right) = \mathbb{P}\left(h(x, M_{a'}) \mid X = x, A = a, M = m\right). \tag{1}$$

This equation illustrates the need to care about the counterfactual mediator distribution. Under the consistency assumption, the right side of the equality simplifies to the delta (point mass) distribution $\delta(h(x, m))$.

## 4 GENERATIVE COUNTERFACTUAL FAIRNESS NETWORK

**Overview:** Here, we introduce our proposed method called *Generative Counterfactual Fairness Network* (GCFN). An overview of our method is in Fig. 2. GCFN proceeds in two steps: **Step 1** uses a significantly modified GAN to learn the counterfactual distribution of the mediator. **Step 2** uses the generated counterfactual mediators from the first step together with our *counterfactual mediator regularization* to enforce counterfactual fairness. The pseudocode is in Appendix D.

*Why do we need counterfactuals of the mediator?* Different from existing methods for causal effect estimation (Bica et al., 2020; Yoon et al., 2018), we are *not* interested in obtaining counterfactuals of the target $Y$. Instead, we are interested in counterfactuals for the mediator $M$, which captures the entire influence of the sensitive attribute and its descendants on the target. Thus, by training the prediction model with our *counterfactual mediator regularization*, we remove the information from the sensitive attribute to ensure fairness while keeping the rest useful information of data to maintain high prediction performance. *What is the advantage of using a GAN in our method?* The GAN in our method enables us to directly learn transformations of factual mediators to counterfactuals without the intermediate step of inferring latent variables. As a result, we eliminate the need for the abduction-action-prediction procedure (Pearl, 2009) and avoid the complexities and potential inaccuracies of inferring and then using latent variables for prediction.

## 4.1 STEP 1: GAN FOR GENERATING COUNTERFACTUAL OF THE MEDIATOR

In Step 1, we aim to generate counterfactuals of the mediator (since the ground-truth counterfactual mediator is unavailable). Our generator $G$ produces the counterfactual of the mediators given observational data. Concurrently, our discriminator $D$ differentiates the factual mediator from the generated counterfactual mediators. This adversarial training process encourages $G$ to learn the counterfactual distribution of the mediator.

### 4.1.1 COUNTERFACTUAL GENERATOR $G$

The objective of the generator $G$ is to learn the counterfactual distribution of the mediator, i.e., $\mathbb{P}\left(M_{a'} \mid X = x, A = a, M = m\right)$. Formally, $G : \mathcal{X} \times \mathcal{A} \times \mathcal{M} \to \mathcal{M}$. $G$ takes the factual sensitive attribute $A$, the factual mediator $M$, and the covariates $X$ as inputs, sampled from the joint (observational) distribution $\mathbb{P}_{X,A,M}$, denoted as $\mathbb{P}_\mathrm{f}$ for short. $G$ outputs two potential mediators, $\hat{M}_0$ and $\hat{M}_1$, from which one is factual and the other is counterfactual. For notation, we use $G\left(X, A, M\right)$ to refer to the output of the generator. Thus, we have

$$G\left(X, A, M\right)_a = \hat{M}_a \quad \text{for} \quad a \in \{0, 1\} \tag{2}$$

In our generator $G$, we intentionally output not only the counterfactual mediator but also the factual mediator, even though the latter is observable. The reason is that we can use it to further stabilize the training of the generator. For this, we introduce a reconstructive loss $\mathcal{L}_\mathrm{f}$, which we use to ensure that the generated factual mediator $\hat{M}_A$ is similar to the observed factual mediator $M$. Formally, we define the reconstruction loss

$$\mathcal{L}_\mathrm{f}(G) = \mathbb{E}_{(X,A,M) \sim \mathbb{P}_\mathrm{f}} \left[ \left\| M - G\left(X, A, M\right)_A \right\|_2^2 \right], \tag{3}$$

where $\| \cdot \|_2$ is the $L_2$-norm.

### 4.1.2 COUNTERFACTUAL DISCRIMINATOR $D$

The discriminator $D$ is carefully adapted to our setting. In an ideal world, we would have $D$ discriminate between real vs. fake counterfactual mediators; however, the counterfactual mediators are not observable. Instead, we train $D$ to discriminate between factual mediators vs. generated counterfactual mediators. Note that this is different from the conventional discriminators in GANs that seek to discriminate real vs. fake samples (Goodfellow et al., 2014a). Formally, our discriminator $D$ is designed to differentiate the factual mediator $M$ (as observed in the data) from the generated counterfactual mediator $\hat{M}_{A'}$ (as generated by $G$).

We modify the output of $G$ before passing it as input to $D$: We replace the generated factual mediator $\hat{M}_A$ with the observed factual mediator $M$. We denote the new, combined data by $\tilde{G}\left(X, A, M\right)$, which is defined via

$$\tilde{G}\left(X, A, M\right)_a = \begin{cases} M, & \text{if } A = a, \\ G\left(X, A, M\right)_a, & \text{if } A = a', \end{cases} \quad \text{for } a \in \{0, 1\}. \tag{4}$$

The discriminator $D$ then determines which component of $\tilde{G}$ is the observed factual mediator and thus outputs the corresponding probability. Formally, for the input $(X, \tilde{G})$, the output of the discriminator $D$ is

$$D\left(X, \tilde{G}\right)_a = \hat{\mathbb{P}}\left(M = \tilde{G}_a \mid X, \tilde{G}\right) = \hat{\mathbb{P}}\left(A = a \mid X, \tilde{G}\right) \quad \text{for } a \in \{0, 1\}. \tag{5}$$

### 4.1.3 ADVERSARIAL TRAINING OF OUR GAN

Our GAN is trained in an adversarial manner: (i) the generator $G$ seeks to generate counterfactual mediators in a way that minimizes the probability that the discriminator can differentiate between factual mediators and counterfactual mediators, while (ii) the discriminator $D$ seeks to maximize the probability of correctly identifying the factual mediator. Put simply, by viewing $A$ as the true label in a classification class, our loss is like the cross-entropy loss for the classification task . We thus use an adversarial loss $\mathcal{L}_\mathrm{adv}$

$$\mathcal{L}_{\mathrm{adv}}(G, D) = \mathbb{E}_{(X,A,M)\sim\mathbb{P}_{\mathrm{f}}}\left[\log\left(D(X, \tilde{G}(X, A, M))_A\right)\right]. \tag{6}$$

Overall, our GAN is trained through an adversarial training procedure with a minimax problem as

$$\min_{G}\max_{D}\mathcal{L}_{\mathrm{adv}}(G, D) + \alpha\mathcal{L}_{\mathrm{f}}(G), \tag{7}$$

with a hyperparameter $\alpha$ on $\mathcal{L}_{\mathrm{f}}$.

We provide a theoretical justification for our GAN in Appendix C. Therein, we show that, under mild identifiability conditions, the counterfactual distribution of the mediator, i.e., $\mathbb{P}\left(M_{a'} \mid X = x, A = a, M = m\right)$, is consistently estimated by our GAN.

### 4.2 STEP 2: COUNTERFACTUAL FAIR PREDICTION THROUGH COUNTERFACTUAL MEDIATOR REGULARIZATION

In Step 2, we use the output of the GAN to train a prediction model under counterfactual fairness in a supervised way. For this, we introduce our *counterfactual mediator regularization* that enforces counterfactual fairness w.r.t the sensitive attribute. Let $h$ denote our prediction model (e.g., a neural network). We define our *counterfactual mediator regularization* $\mathcal{R}_{\mathrm{cm}}(h)$ as

$$\mathcal{R}_{\mathrm{cm}}(h) = \mathbb{E}_{(X,A,M)\sim\mathbb{P}_{\mathrm{f}}}\left[\left\|h\left(X, M\right) - h\left(X, \hat{M}_{A'}\right)\right\|_2^2\right]. \tag{8}$$

Our counterfactual mediator regularization has three important characteristics: (1) It is non-trivial. Different from traditional regularization, our $\mathcal{R}_{\mathrm{cm}}$ is not based on the representation of the prediction model $h$ but it *involves a GAN-generated counterfactual $\hat{M}_{A'}$ that is otherwise not observable*. (2) Our $\mathcal{R}_{\mathrm{cm}}$ is not used to constrain the learned representation (e.g., to avoid overfitting) but it is used to change the actual learning objective to achieve the property of counterfactual fairness. (3) Our $\mathcal{R}_{\mathrm{cm}}$ fulfills theoretical properties (see Sec. 4.3). Specifically, we show later that, under some conditions, our regularization actually optimizes against counterfactual fairness and thus should learn our task as desired.

The overall loss $\mathcal{L}(h)$ is as follows. We fit the prediction model $h$ using a cross-entropy loss $\mathcal{L}_{\mathrm{ce}}(h)$. We further integrate the above counterfactual mediator regularization $\mathcal{R}_{\mathrm{cm}}(h)$ into our overall loss $\mathcal{L}(h)$. For this, we introduce a weight $\lambda \geq 0$ to balance the trade-off between prediction performance and the level of counterfactual fairness. Formally, we have

$$\mathcal{L}(h) = \mathcal{L}_{\mathrm{ce}}(h) + \lambda\mathcal{R}_{\mathrm{cm}}(h). \tag{9}$$

A large value of $\lambda$ increases the weight of $\mathcal{R}_{\mathrm{cm}}$, thus leading to a prediction model that is strict with regard to counterfactual fairness, while a lower value allows the prediction model to focus more on producing accurate predictions. As such, $\lambda$ offers additional flexibility to decision-makers as they tailor the prediction model based on the fairness needs in practice.

### 4.3 THEORETICAL ANALYSIS

Below, we provide theoretical analysis to show that our proposed counterfactual mediator regularization is effective in ensuring counterfactual fairness for predictions. Following Grari et al. (2023), we measure the level of counterfactual fairness $CF$ via $\mathbb{E}\left[\left\|(h\left(X, M\right) - h\left(X, M_{A'}\right))\right\|_2^2\right]$. It is straightforward to see that, the smaller $CF$ is, the more counterfactual fairness the prediction model achieves.

We show that by empirically measuring our generated counterfactual of the mediator, we can thus quantify to what extent counterfactual fairness $CF$ is fulfilled in the prediction model. We give an upper bound in the following lemma.

**Lemma 1** (Counterfactual mediator regularization bound). *Given the prediction model $h$ that is Lipschitz continuous with a Lipschitz constant $\mathcal{C}$, we have*

$$\mathbb{E}\left[\left\|(h(X, M) - h(X, M_{A'})\right\|_2^2\right] \leq \mathcal{C}\,\mathbb{E}\left[\left\|M_{A'} - \hat{M}_{A'}\right\|_2^2\right] + \mathcal{R}_{\mathrm{cm}}(h). \tag{10}$$

*Proof.* See Appendix C. □

The inequality in Lemma 1 states that the influence from the sensitive attribute on the target variable is upper-bounded by (i) the estimation of counterfactual mediators (first term) and (ii) the counterfactual mediator regularization (second term).

Given Lemma 1, the natural question arises under which conditions our generator produces consistent counterfactuals and thus leads to the correct estimation of the counterfactual mediator. Therefore, we give a theoretical guarantee for estimating the counterfactual distribution in the following lemma. This is a new theoretical result regarding counterfactual identifiability with GANs. To the best of our knowledge, we are the first to give such a theoretical guarantee on the generated counterfactual with GANs.

**Lemma 2** (Consistent estimation of the counterfactual distribution with GAN). *Let the observational distribution $\mathbb{P}_{X,A,M} = \mathbb{P}_f$ be induced by an SCM $\mathcal{M} = \langle \mathbf{V}, \mathbf{U}, \mathcal{F}, \mathbb{P}(\mathbf{U}) \rangle$ with*

$$\mathbf{V} = \{X, A, M, Y\}, \quad \mathbf{U} = \{U_{XA}, U_M, U_Y\},$$
$$\mathcal{F} = \{f_X(u_{XA}), f_A(x, u_{XA}), f_M(x, a, u_M), f_Y(x, m, u_Y)\}, \quad \mathbb{P}(\mathbf{U}) = \mathbb{P}(U_{XA})\mathbb{P}(U_M)\mathbb{P}(U_Y),$$

*and with the causal graph as in Figure 1. Let $\mathcal{M} \subseteq \mathbb{R}$ and $f_M$ be a bijective generation mechanism (BGM) (Nasr-Esfahany et al., 2023; Melnychuk et al., 2023), i.e., $f_M$ is a strictly increasing (decreasing) continuously-differentiable transformation wrt. $u_M$. Then:*

1. *The counterfactual distribution of the mediator simplifies to one of two possible point mass distributions*

$$\mathbb{P}(M_{a'} \mid X = x, A = a, M = m) = \delta(\mathbb{F}^{-1}(\pm \mathbb{F}(m; x, a) \mp 0.5 + 0.5; x, a')), \quad (11)$$

   *where $\mathbb{F}(\cdot; x, a)$ and $\mathbb{F}^{-1}(\cdot; x, a)$ are a CDF and an inverse CDF of $\mathbb{P}(M \mid X = x, A = a)$, respectively, and $\delta(\cdot)$ is a Dirac-delta distribution;*

2. *If the generator of GAN is a continuously differentiable function with respect to $M$, then it consistently estimates the counterfactual distribution of the mediator, $\mathbb{P}(M_{a'} \mid X = x, A = a, M = m)$.*

*Proof.* See Appendix C. □

**Remark 1.** *We proved that the generator converges to one of the two BGM solutions in Eq. 11. Notably, the difference between the two solutions is negligibly small, when the conditional standard deviation of the mediator is small.*

Lemma 2 states that our generator consistently estimates the counterfactual distribution of the mediator $\mathbb{P}(M_{a'} \mid X = x, A = a, M = m)$. It gives a guarantee that the generated counterfactual mediators are correctly estimated, which is the first term of the upper bound in Lemma 1. Hence, by reducing the second term $\mathcal{R}_{cm}$ through minimizing our training loss in Eq. 9, we can effectively enforce the predictor to be more counterfactual fair.[5]

## 5 EXPERIMENTS

### 5.1 SETUP

**Baselines:** We compare our method against the following state-of-the-art approaches: (1) **CFAN** (Kusner et al., 2017): Kusner et al.'s algorithm with additive noise where only non-descents of sensitive attributes and the estimated latent variables are used for prediction; (2) **CFUA** (Kusner et al., 2017): a variant of the algorithm which does not use the sensitive attribute or any descents of the sensitive attribute; (3) **mCEVAE** (Pfohl et al., 2019): adds a maximum mean discrepancy to regularize the generations in order to remove the information the inferred latent variable from sensitive information; (4) **DCEVAE** (Kim et al., 2021): a VAE-based approach that disentangles the exogenous uncertainty into two variables; (5) **ADVAE** (Grari et al., 2023): adversarial neural learning approach which should be more powerful than penalties from maximum mean discrepancy but is aimed the continuous setting; (6) **HSCIC** (Quinzan et al., 2022): originally designed to enforces the predictions to remain invariant to changes of sensitive attributes using conditional kernel mean

---

[5]Details how we ensure Lipschitz continuity in $h$ are in Appendix F.

embeddings but which we adapted for counterfactual fairness. We also adapt applicable baselines from fair dataset generation: (7) **CFGAN** (Xu et al., 2019): which we extend with a second-stage prediction model. Details are in Appendix F.

**Performance metrics:** Methods for causal fairness aim at both: (i) achieve high accuracy while (ii) ensuring causal fairness, which essentially yields a multi-criteria decision-making problem. To this end, we follow standard procedures and reformulate the multi-criteria decision-making problem using a utility function $U_\gamma(accuracy, CF) : \mathbb{R}^2 \mapsto \mathbb{R}$, where $CF$ is the metric for measuring counterfactual fairness from Sec. 4.3. We define the utility function as $U_\gamma(accuracy, CF) = accuracy - \gamma \times CF$ with a given utility weight $\gamma$. A larger utility $U_\gamma$ is better. The weight $\gamma$ depends on the application and is set by the decision-maker; here, we report results for a wide range of weights $\gamma \in \{0.1, \ldots, 1.0\}$.

**Implementation details:** We implement our GCFN in PyTorch. Both the generator and the discriminator are designed as deep neural networks. We use LeakyReLU, batch normalization in the generator for stability, and train the GAN for 300 epochs with 256 batch size. The prediction model is a multilayer perceptron, which we train for 30 epochs at a 0.005 learning rate. Since the utility function considers two metrics, the weight $\lambda$ is set to 0.5 to get a good balance. More implementation details and hyperparameter tuning are in Appendix F.

## 5.2 RESULTS FOR (SEMI-)SYNTHETIC DATASETS

We explicitly focus on (semi-)synthetic datasets, which allow us to compute the true counterfactuals and thus validate the effectiveness of our method.

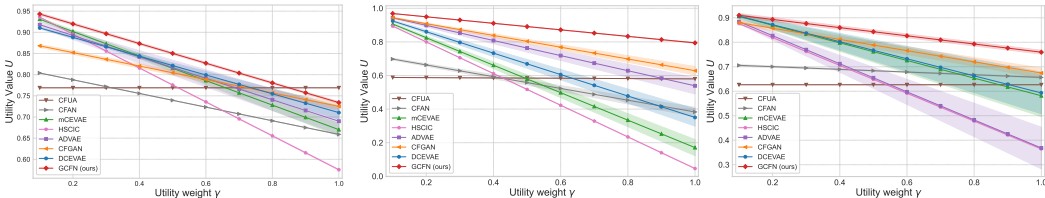

Figure 3: Results for (semi-)synthetic datasets. *Left:* fully-synthetic dataset. *Center:* semi-synthetic dataset with sigmoid function. *Right:* with sin function. A larger utility is better. Shown: mean ± std over 5 runs.

**Setting:** (1) We follow previous works that simulate a fully synthetic dataset for performance evaluations (Kim et al., 2021; Quinzan et al., 2022). We simulate sensitive attributes and target to follow a Bernoulli distribution with the sigmoid function while the mediator is generated from a function of the sensitive attribute, covariates, and some Gaussian noise. (2) We use the Law School (Wightman, 1998) dataset to predict whether a candidate passes the bar exam and where *gender* is the sensitive attribute. The mediator uses a linear combination together with a sigmoid function. The target variable is generated from the Bernoulli distribution with a probability calculated by a function of covariates, mediator, and noise. (3) We follow the previous dataset but, instead of a sigmoid function, we generate the mediator via a sin function. The idea behind this is to have a more flexible data-generating function, which makes it more challenging to learn latent variables. For all datasets, we use 20% as a test set. Further details are in Appendix E.

**Results:** Results are shown in Fig. 3. We make the following findings. (1) Our GCFN performs best. (2) Compared to the baselines, the performance gain from our GCFN is large (up to ~30%). (3) The performance gain for our GCFN tends to become larger for larger $\gamma$. (4) Most baselines in the semi-synthetic dataset with sin function have a large variability across runs as compared to our GCFN, which further demonstrates the robustness of our method. (5) Conversely, the strong performance of our

Table 1: Our GCFN can learn the distribution of the counterfactual mediator. The normalized $MSE(M_{A'}, \hat{M}_{A'})$ is $\approx 0$, showing the generated counterfactual mediator is similar to the ground-truth counterfactual mediator. In contrast, both the factual and the generated counterfactual mediator are highly dissimilar.

|  | Synthetic | Semi-syn. (sigmoid) | Semi-syn. (sin) |
|---|---|---|---|
| $MSE(M, M_{A'})$ | $1.00_{\pm 0.00}$ | $1.00_{\pm 0.00}$ | $1.00_{\pm 0.00}$ |
| $MSE(M, \hat{M}_{A'})$ | $1.00_{\pm 0.005}$ | $1.00_{\pm 0.003}$ | $1.01_{\pm 0.004}$ |
| $MSE(M_{A'}, \hat{M}_{A'})$ | $0.01_{\pm 0.002}$ | $0.02_{\pm 0.001}$ | $0.03_{\pm 0.001}$ |

$M$: ground-truth factual mediator; $M_{A'}$: ground-truth counterfactual mediator; $\hat{M}_{A'}$: generated counterfactual mediator

GCFN in the semi-synthetic dataset with sin function demonstrates that our tailed GAN can even capture complex counterfactual distributions.

**Additional insights:** As an additional analysis, we now provide further insights into how our GCFN operates. Specifically, one may think that our GCFN simply learns to reproduce factual mediators in the GAN rather than actually learning the counterfactual mediators. However, this is *not* the case. To show this, we compare the (1) the factual mediator $M$, (2) the ground-truth counterfactual mediator $M_{A'}$, and (3) the generated counterfactual mediator $\hat{M}_{A'}$. The normalized mean squared error (MSE) between them is in Table 1. We find: (1) The factual mediator and the generated counterfactual mediator are highly dissimilar. This is shown by a normalized $\text{MSE}(M, \hat{M}_{A'})$ of $\approx 1$. (2) The ground-truth counterfactual mediator and our generated counterfactual mediator are highly similar. This shown by a normalized $\text{MSE}(M_{A'}, \hat{M}_{A'})$ of close to zero. In sum, our GCFN is effective in learning counterfactual mediators (and does *not* reproduce the factual data).

### 5.3 RESULTS FOR REAL-WORLD DATASETS

We now demonstrate the applicability of our method to real-world data. Since ground-truth counterfactuals are unobservable for real-world data, we refrain from benchmarking, but, instead, we now provide additional insights to offer a better understanding of our method.

#### 5.3.1 RESULTS FOR UCI ADULT DATASET

**Setting:** We use UCI Adult (Asuncion & Newman, 2007) to predict if individuals earn a certain salary but where *gender* is a sensitive attribute. Further details are in Appendix E.

**Insights:** To better understand the role of our counterfactual mediator regularization, we trained prediction models both with and without applying $\mathcal{R}_{\text{cm}}$. Our primary focus is to show the shifts in the distribution of the predicted target variable (salary) across the sensitive attribute (gender). The corresponding density plots are in Fig. 4. One would expect the distributions for males and females should be

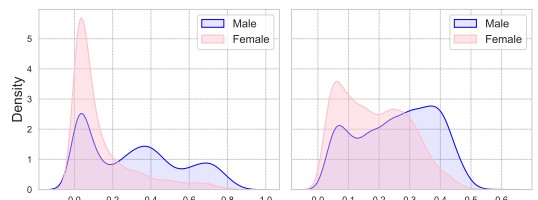

Figure 4: Density of the predicted target variable (salary) across male vs. female, where the idea is that similar distributions should be fairer. Left: w/o our counterfactual mediator regularization. Right: w/ our counterfactual mediator regularization.

more similar if the prediction is fairer. However, we do not see such a tendency for a prediction model without our counterfactual mediator regularization. In contrast, when our counterfactual mediator regularization is used, both distributions are fairly similar as desired. Further visualizations are in Appendix H.

**Accuracy and fairness trade-off:** We vary the fairness weight $\lambda$ from 0 to 1 to see the trade-off between prediction performance and the level of counterfactual fairness. Since the ground-truth counterfactual is not available for the real-world dataset, we use the generated counterfactual to measure counterfactual fairness on the test dataset. The results are in Fig. 5. In line with our expectations, we see that larger values for $\lambda$ lead the predictions to be more strict w.r.t counterfactual fairness, while lower values allow the predictions to have greater accuracy. Hence, the fairness weight $\lambda$ offers flexibility to decision-makers, so that they can tailor our method to the fairness needs in practice.

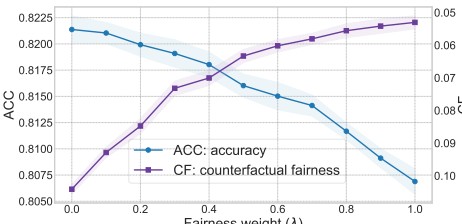

Figure 5: Trade-off between accuracy (ACC) and counterfactual fairness (CF) across different fairness weights $\lambda$. ACC: the higher (↑) the better. CF: the lower (↓) the better. Shown: mean ± std over 5 runs.

#### 5.3.2 RESULTS ON COMPAS DATASET

**Setting:** We use the COMPAS dataset (Angwin et al., 2016) to predict recidivism risk of criminals and where *race* is a sensitive attribute. The dataset also has a COMPAS score for that purpose, yet it was revealed to have racial biases (Angwin et al., 2016). In particular, black defendants were

frequently overestimated of their risk of recidivism. Motivated by this finding, we focus our efforts on reducing such racial biases. Further details about the setting are in Appendix E.

**Insights:** We first show how our method adds more fairness to real-world applications. For this, we compare the recidivism predictions from the criminal justice process against the actual reoffenses two years later. Specifically, we compute (i) the accuracy of the official COMPAS score in predicting reoffenses and (ii) the accuracy of our GCFN in predicting the outcomes.

Table 2: Comparison of predictions against actual reoffenses.

| Method | ACC | PPV | FPR | FNR |
|---|---|---|---|---|
| COMPAS | 0.6644 | 0.6874 | 0.4198 | 0.2689 |
| GCFN (ours) | 0.6753 | 0.7143 | 0.3519 | 0.3032 |

ACC (accuracy); PPV (positive predictive value); FPR (false positive rate); FNR (false negative rate).

comes. The results are in Table 2. We see that our GCFN has a better accuracy. More important is the false positive rate (FPR) for black defendants, which measures how often black defendants are assessed at high risk, even though they do not recidivate. Our GCFN reduces the FPR of black defendants from 41.98% to 35.19%. In sum, our method can effectively decrease the bias towards black defendants.

We now provide insights at the defendant level to better understand how black defendants are treated differently by the COMPAS score vs. our GCFN. Fig. 6 shows the number of such different treatments across different characteristics of the defendants. (1) Our GCFN makes oftentimes different predictions for black defendants with a medium COMPAS score around 4 and 5. However, the predictions for black defendants with a very high or low COMPAS

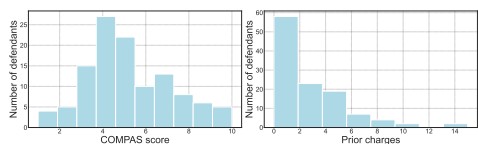

Figure 6: Distribution of black defendants that are treated differently using our GCFN. Left: COMPAS score. Right: Prior charges.

score are similar, potentially because these are 'clear-cut' cases. (2) Our method arrives at significantly different predictions for patients with low prior charges. This is expected as the COMPAS score overestimates the risk and is known to be biased (Angwin et al., 2016). Further insights are in the Appendix H.

To exemplify the above, Fig. 7 shows two defendants from the data. Both primarily vary in their race (black vs. white) and their number of prior charges (2 vs. 7). Interestingly, the COMPAS score coincides with race, while our method makes predictions that correspond to the prior charges.

## 6 DISCUSSION

**Flexibility:** Our method works with various data types. In particular, it works with both discrete and multi-dimensional sensitive attributes (see G). It can also be straightforwardly extended to, e.g., continuous target variables. Our further method offers to choose the fairness-accuracy trade-off according to needs in practice. By choosing a large counterfactual fairness weight $\lambda$, our method enforces counterfactual fairness in a strict manner. Nevertheless, by choosing $\lambda$ appropriately, our method supports applications where practitioners seek a trade-off between performance and fairness.

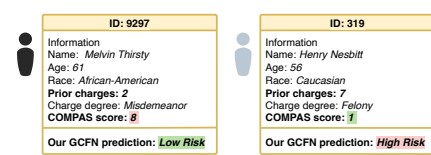

Figure 7: Frequency of how often the predictions from the COMPAS score and our GCFN are different. Shown are the frequency across different COMPAS scores (left) and different numbers of prior charges (right).

**Limitations.** We acknowledge that our method for counterfactual fairness rests on mathematical assumptions, in line with prior work. Further, as with all research on algorithmic fairness, we usher for a cautious, responsible, and ethical use. Sometimes, unfairness may be historically ingrained and require changes beyond the algorithmic layer.

**Conclusion:** Our work provides a novel method for achieving predictions under counterfactual fairness. Thanks to our combination of the counterfactual mediator regularization with GAN, our GCFN addresses key shortcomings of existing baselines that are based on inferring latent variables, and our GCFN thus achieves state-of-the-art performance. To the best of our knowledge, ours is the first neural method for counterfactual fair predictions with theoretical guarantees.

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

## A  MATHEMATICAL BACKGROUND

**Notation:** Capital letters such as $U$ denote a random variable and small letters $u$ its realizations from corresponding domains $\mathcal{U}$. Bold capital letters such as $\mathbf{U} = \{U_1, \ldots, U_n\}$ denote finite sets of random variables. Further, $\mathbb{P}(Y)$ is the distribution of a variable $Y$.

**SCM:** A structural causal model (SCM) (Pearl, 2009) is a 4-tuple $\langle \mathbf{V}, \mathbf{U}, \mathcal{F}, \mathbb{P}(\mathbf{U})\rangle$, where $\mathbf{U}$ is a set of exogenous (background) variables that are determined by factors outside the model; $\mathbf{V} = \{V_1, \ldots, V_n\}$ is a set of endogenous (observed) variables that are determined by variables in the model (i.e., by the variables in $\mathbf{V} \cup \mathbf{U}$ ); $\mathcal{F} = \{f_1, \ldots, f_n\}$ is the set of structural functions determining $\mathbf{V}, v_i \leftarrow f_i\left(\mathrm{pa}\left(v_i\right), u_i\right)$, where $\mathrm{pa}\left(V_i\right) \subseteq \mathbf{V} \backslash V_i$ and $U_i \subseteq \mathbf{U}$ are the functional arguments of $f_i$; $\mathbb{P}(\mathbf{U})$ is a distribution over the exogenous variables $\mathbf{U}$.

**Potential outcome:** Let $X$ and $Y$ be two random variables in $\mathbf{V}$ and $\mathbf{u} = \{u_1, \ldots, u_n\} \in \mathcal{U}$ be a realization of exogenous variables. The potential outcome $Y_x(\mathbf{u})$ is defined as the solution for $Y$ of the set of equations $\mathcal{F}_x$ evaluated with $\mathbf{U} = \mathbf{u}$ (Pearl, 2009). That is, after $\mathbf{U}$ is fixed, the evaluation is deterministic. $Y_x(\mathbf{u})$ is the value variable $Y$ would take if (possibly contrary to observed facts) $X$ is set to $x$, for a specific realization $\mathbf{u}$. In the rest of the paper, we use $Y_x$ as the short for $Y_x(\mathbf{U})$.

**Observational distribution:** A structural causal model $\mathcal{M} = \langle \mathbf{V}, \mathbf{U}, \mathcal{F}, \mathbb{P}(\mathbf{U})\rangle$ induces a joint probability distribution $\mathbb{P}(\mathbf{V})$ such that for each $Y \subseteq \mathbf{V}$, $\mathbb{P}^{\mathcal{M}}(Y = y) = \sum_u \mathbb{1}(Y(\mathbf{u}) = y)\mathbb{P}(\mathbf{U} = \mathbf{u})$ where $Y(\mathbf{u})$ is the solution for $Y$ after evaluating $\mathcal{F}$ with $\mathbf{U} = \mathbf{u}$ (Bareinboim et al., 2022).

**Counterfactual distributions:** A structural causal model $\mathcal{M} = \langle \mathbf{V}, \mathbf{U}, \mathcal{F}, \mathbb{P}(\mathbf{U})\rangle$ induces a family of joint distributions over counterfactual events $Y_x, \ldots, Z_w$ for any $Y, Z, \ldots, X, W \subseteq \mathbf{V}$ : $\mathbb{P}^{\mathcal{M}}\left(Y_x = y, \ldots, Z_w = z\right) = \sum_u \mathbb{1}\left(Y_x(\mathbf{u}) = y, \ldots, Z_w(\mathbf{u}) = z\right)\mathbb{P}(\mathbf{U} = \mathbf{u})$ (Bareinboim et al., 2022). This equation contains variables with different subscripts, which syntactically represent different potential outcomes or counterfactual worlds.

**Causal graph:** A graph $\mathcal{G}$ is said to be a causal graph of SCM $\mathcal{M}$ if represented as a directed acyclic graph (DAG), where (Pearl, 2009; Bareinboim et al., 2022) each endogenous variable $V_i \in \mathbf{V}$ is a node; there is an edge $V_i \longrightarrow V_j$ if $V_i$ appears as an argument of $f_j \in \mathcal{F}$ ($V_i \in \mathrm{pa}(V_j)$); there is a bidirected edge $V_i \leftarrow\!\dashrightarrow V_j$ if the corresponding $U_i, U_j \subset \mathbf{U}$ are correlated ($U_i \cap U_j \neq \varnothing$) or the corresponding functions $f_i, f_j$ share some $U_{ij} \in \mathbf{U}$ as an argument.

# B EXTENDED RELATED WORK

## B.1 FAIRNESS

Recent literature has extensively explored different fairness notions (e.g., Feldman et al., 2015; Di Stefano et al., 2020; Dwork et al., 2012; Grgic.Hlaca et al., 2016; Hardt et al., 2016; Joseph et al., 2016; Pfohl et al., 2019; Salimi et al., 2019; Zafar et al., 2017; Wadsworth et al., 2018; Celis et al., 2019; Chen et al., 2019; Zhang et al., 2018; Madras et al., 2019; Di Stefano et al., 2020; Madras et al., 2018). For a detailed overview, we refer to Makhlouf et al. (2020) and Plecko & Bareinboim (2022). There have been also theoretical advances (e.g., Fawkes et al., 2022; Rosenblatt & Witter, 2023) but these are orthogonal to ours.

Existing fairness notions can be loosely classified into notions for group- and individual-level fairness, as well as causal notions, some aim at path-specific fairness (e.g., Nabi & Shpitser, 2018; Chiappa, 2019). We adopt the definition of counterfactual fairness from Kusner et al. (2017).

**Counterfactual fairness** (Kusner et al., 2017): Given a predictive problem with fairness considerations, where $A, X$ and $Y$ represent the sensitive attributes, remaining attributes, and output of interest respectively, for a causal model $\mathcal{M} = \langle \mathbf{V} = \{A, X, Y\}, \mathbf{U}, \mathcal{F}, \mathbb{P}(\mathbf{U}) \rangle$, prediction model $\hat{Y} = h(X, A, \mathbf{U})$ is counterfactual fair, if under any context $X = x$ and $A = a$,

$$\mathbb{P}\left(\hat{Y}_a(\mathbf{U}) \mid X = x, A = a\right) = \mathbb{P}\left(\hat{Y}_{a'}(\mathbf{U}) \mid X = x, A = a\right), \tag{12}$$

for any value $a'$ attainable by $A$. This is equivalent to the following formulation:

$$\mathbb{P}\left(h(X_a(\mathbf{U}), a, \mathbf{U}) \mid X = x, A = a\right) = \mathbb{P}\left(h(X_{a'}(\mathbf{U}), a', \mathbf{U}) \mid X = x, A = a\right). \tag{13}$$

Our paper adapts the later formulation by doing the following. First, we make the prediction model independent of the sensitive attributes $A$, as they could only make the predictive model unfairer. Second, given the general non-identifiability of the posterior distribution of the exogenous noise, i.e., $\mathbb{P}(\mathbf{U} \mid X = x, A = a)$, we consider only the prediction models dependent on the observed covariates. Third, we split observed covariates $X$ on pre-treatment covariates (confounders) and post-treatment covariates (mediators). Thus, we yield our definition of a fair predictor in Eq. 1.

**Benefits over latent variable baselines:** Importantly, the latent variable baselines for counterfactual fairness (e.g., mCEVAE (Pfohl et al., 2019), DCEVAE (Kim et al., 2021), and ADVAE (Grari et al., 2023)) are far from being easy as they do not rely on off-the-shelf methods. Rather, they also learn a latent variable in non-trivial ways. The inferred latent variable $U$ should be independent of the sensitive attribute $A$ while representing all other useful information from the observation data. However, there are two main challenges: (1) The latent variable $U$ is *not* identifiable. (2) It is very *hard* to learn such $U$ to satisfy the above independence requirement, especially for high-dimensional or other more complicated settings. Hence, we argue that baselines based on some custom latent variables are highly challenging.

Because of (1) and (2), there are **no** theoretical guarantees for the VAE-based methods. Hence, it is mathematically unclear whether they actually learn the correct counterfactual fair predictions. In fact, there is even rich empirical evidence that VAE-based methods are often *suboptimal*. VAE-based methods use the estimated variable $U$ in the first step to learn the counterfactual outcome $\mathbb{P}\left(\hat{Y}_{a'}(\mathbf{U}) \mid X = x, A = a, M = m\right)$. The inferred, non-identifiable latent variable can be correlated with the sensitive attribute which may harm fairness, or it might not fully represent the rest of the information from data and harm prediction performance.

Importantly, the latent variable baselines do *not* allow for identifability. In causal inference, "identifiability" refers to a mathematical condition that permits a causal quantity to be measured from observed data (Pearl, 2009). Importantly, identification is *different* from estimation because methods that act as heuristics may return estimates but they do not correspond to the true value. For the latter, see D'Amour (2019) where the authors provide several concerns that, if a latent variable is not unique, it is possible to have local minima, which leads to unsafe results in causal inference.

Non-identifiable for VAE-based methods have been shown in prior works of literature. In a recent paper Xia et al. (2022), the authors show that VAE-based counterfactual inference do *not* allow for identifiability. The results directly apply to variational inference-based methods, which *do* not have proper identification guarantees. Also, the result from non-linear ICA (which is the task of variational autoencoders) shows that the latent variables are *non-identifiable* Khemakhem et al. (2020).

In simple words, VAE-based methods can estimate the latent variable but it is *not* guaranteed that it can be correctly identified, leading to risks that the latent variable is often estimated incorrectly. Note that non-identifiability of the latent variables means non-identifiability of the counterfactual queries. We refer to paper Melnychuk et al. (2023), which show that the non-identifiability of the latent variables means non-identifiability of the counterfactual queries. Hence, VAE-based methods can **not** ensure that they correctly learn counterfactual fairness, only our method does so.

## B.2  DIFFERENCE FROM CFGAN

Even though CFGAN also employs GANs, it is vastly **different** from our method.

1. *Different tasks:* CFGAN is designed for fair data generation tasks, while our model is designed for learning predictors to be counterfactual fairness. Hence, both address **different tasks**.

2. *Different architectures:* CFGAN employs **two** generators, each aimed at simulating the original causal model and the interventional model, and two discriminators, which ensure data utility and causal fairness. We only employ a streamlined architecture with a **single** generator and discriminator. Further, fairness enters both architectures at **different places**. In CFGAN, fairness is ensured through the GAN setup, whereas our method ensures fairness in a second step through our counterfactual mediator regularization.

3. *Different training objectives:* The training objectives are **different**: CFGAN learns to **mimic factual data**. In our method, the generator **learns the counterfactual distribution of the mediator** through the discriminator distinguishing factual from counterfactual mediators.

4. *No theoretical guarantee for CFGAN:* CFGAN is proposed to synthesize a dataset that satisfies counterfactual fairness. However, a recent paper (Abroshan et al., 2022) has shown that CFGAN is actually considering interventions (=level 2 in Pearl's causality ladder) and **not** counterfactuals (=level 3).[6] Hence, CFGAN does **not** fulfill the counterfactual fairness notion, but a different notion based on do-operator (intervention). For details, we refer to (Abroshan et al., 2022), Definition 5 therein, called "Discrimination avoiding through causal reasoning"): A generator is said to be fair if the following equation holds: for any context $A = a$ and $X = x$, for all value of $y$ and $a' \in \mathcal{A}$, $P(Y = y \mid X = x, do(A = a)) = P(Y = y \mid X = x, do(A = a'))$, which is different from the counterfactual fairness $P(\hat{Y}_a = y \mid X = x, A = a) = P(\hat{Y}_{a'} = y \mid X = x, A = a)$.. Moreover, **CFGAN lacks theoretical support** for its methodology (no identifiable guarantee or counterfactual fairness level). In contrast, our method strictly satisfies the principles of counterfactual fairness and provides theoretical guarantees on the counterfactual fairness level. In sum, **only our method offers theoretical guarantees** for the task at hand.

5. *Suboptimal performance of CFGAN:* Even though CFGAN can, in principle, be applied to counterfactual fairness prediction, it is **suboptimal**. The reason is the following. Unlike CFGAN, which generates complete synthetic data under causal fairness notions, our method only generates counterfactuals of the mediator as an intermediate step, resulting in minimal information loss and better inference performance than CFGAN. Furthermore, since CFGAN needs to train the dual-generator and dual-discriminator together and optimize two adversarial losses, it is more difficult for stable training, and thus its method is less robust than ours.

In sum, even though CFGAN also employs GANs, it is **vastly different from our method**.

---

[6] In the context of Pearl's causal hierarchy** , interventional and counterfactual queries are completely different concepts Bareinboim et al. (2022). (1) Interventional queries are located on level 2 of Pearl's causality ladder. Interventional queries are of the form $P(y \mid do(x))$. Here, the typical question is "What if? What if I do X?", where the activity is "doing". (2) Counterfactual queries are located on level 3 of Pearl's causality ladder. Counterfactual queries are of the form $P(y_x \mid x', y')$, where $x'$ and $y'$ are different values that $X, Y$ took before. Here, the typical question is "What if I had acted differently?", where the activity is "imagining" had a different treatment selected been made in the beginning. Hence, the main difference is that the counterfactual of $y$ is conditioned on the post-treatment outcome (factual outcome) of $y$ and a different $x$ (where $x$ takes a different value than $x'$). For details, we kindly refer to paper Bareinboim et al. (2022); Pearl (2009) for a more technical definition of why intervention and counterfactual are two entirely different concepts.

# C  THEORETICAL RESULTS

Here, we prove Lemma 1 from the main paper, which states that our counterfactual regularization achieves counterfactual fairness if our generator consistently estimates the counterfactuals.

## C.1  PROOF OF LEMMA 1

**Lemma 3** (Counterfactual mediator regularization bound)**.** *Given the prediction model $h$ that is Lipschitz continuous with a Lipschitz constant $\mathcal{C}$, we have*

$$\mathbb{E}\left[\left\|(h(X,M)-h(X,M_{A'})\right\|_2^2\right] \le \mathcal{C}\,\mathbb{E}\left[\left\|M_{A'}-\hat{M}_{A'}\right\|_2^2\right]+\mathcal{R}_{\mathrm{cm}}. \tag{14}$$

*Proof.* Using triangle inequality, we yield

$$\mathbb{E}\left[\left\|h(X,M)-h(X,M_{A'})\right\|_2^2\right] \tag{15}$$

$$=\mathbb{E}\left[\left\|h(X,M)-h(X,M_{A'})+h(X,\hat{M}_{A'})-h(X,\hat{M}_{A'})\right\|_2^2\right] \tag{16}$$

$$\le\mathbb{E}\left[\left\|h(X,M)-h(X,\hat{M}_{A'})\right\|_2^2\right]+\mathbb{E}\left[\left\|h(X,\hat{M}_{A'})-h(X,M_{A'})\right\|_2^2\right] \tag{17}$$

$$=\mathbb{E}\left[\left\|h(X,\hat{M}_{A'})-h(X,M_{A'})\right\|_2^2\right]+\mathcal{R}_{\mathrm{cm}} \tag{18}$$

$$\le\mathcal{C}\,\mathbb{E}\left[\left\|(X,\hat{M}_{A'})-(X,M_{A'})\right\|_2^2\right]+\mathcal{R}_{\mathrm{cm}} \tag{19}$$

$$=\mathcal{C}\,\mathbb{E}\left[\left\|M_{A'}-\hat{M}_{A'}\right\|_2^2\right]+\mathcal{R}_{\mathrm{cm}}. \tag{20}$$

$\square$

## C.2  RESULTS ON COUNTERFACTUAL CONSISTENCY

Given Lemma 1, the natural question arises under which conditions our generator produces consistent counterfactuals. In the following, we provide a theory based on bijective generation mechanisms (BGMs) (Nasr-Esfahany et al., 2023; Melnychuk et al., 2023).

**Lemma 4** (Consistent estimation of the counterfactual distribution with GAN)**.** *Let the observational distribution $\mathbb{P}_{X,A,M}=\mathbb{P}_{\mathrm{f}}$ be induced by an SCM $\mathcal{M}=\langle\mathbf{V},\mathbf{U},\mathcal{F},\mathbb{P}(\mathbf{U})\rangle$ with*

$$\mathbf{V}=\{X,A,M,Y\},\quad \mathbf{U}=\{U_{XA},U_M,U_Y\},$$
$$\mathcal{F}=\{f_X(u_{XA}),f_A(x,u_{XA}),f_M(x,a,u_M),f_Y(x,m,u_Y)\},\quad \mathbb{P}(\mathbf{U})=\mathbb{P}(U_{XA})\mathbb{P}(U_M)\mathbb{P}(U_Y),$$

*and with the causal graph as in Figure 1. Let $\mathcal{M}\subseteq\mathbb{R}$ and $f_M$ be a bijective generation mechanism (BGM) (Nasr-Esfahany et al., 2023; Melnychuk et al., 2023), i.e., $f_M$ is a strictly increasing (decreasing) continuously-differentiable transformation wrt. $u_M$. Then:*

1. *The counterfactual distribution of the mediator simplifies to one of two possible point mass distributions*

   $$\mathbb{P}(M_{a'}\mid X=x,A=a,M=m)=\delta(\mathbb{F}^{-1}(\pm\mathbb{F}(m;x,a)\mp 0.5+0.5;x,a')), \tag{21}$$

   *where $\mathbb{F}(\cdot;x,a)$ and $\mathbb{F}^{-1}(\cdot;x,a)$ are a CDF and an inverse CDF of $\mathbb{P}(M\mid X=x,A=a)$, respectively, and $\delta(\cdot)$ is a Dirac-delta distribution;*

2. *If the generator of GAN is a continuously differentiable function with respect to $M$, then it consistently estimates the counterfactual distribution of the mediator, $\mathbb{P}(M_{a'}\mid X=x,A=a,M=m)$, i.e., converges to one of the two solutions in Eq. equation 21.*

*Proof.* The first statement of the theorem is the main property of bijective generation mechanisms (BGMs), i.e., they allow for deterministic (point mass) counterfactuals. For a more detailed proof, we refer to Lemma B.2 in (Nasr-Esfahany et al., 2023) and to Corollary 3 in (Melnychuk et al.,

2023). Importantly, under mild conditions[7], this result holds in the more general class of BGMs with non-monotonous continuously differentiable functions.

The second statement can be proved in two steps. (i) We show that, given an optimal discriminator, the generator of our GAN estimates the distribution of potential mediators for counterfactual sensitive attributes, i.e., $\mathbb{P}(G(x, a, M_a)_{a'} \mid X = x, A = a) = \mathbb{P}(M_{a'} \mid X = x, A = a)$ in distribution. (ii) Then, we demonstrate that the outputs of the deterministic generator, conditional on the factual mediator $M = m$, estimate $\mathbb{P}(M_{a'} \mid X = x, A = a, M = m)$.

(i) Let $\pi_a(x) = \mathbb{P}(A = a \mid X = x)$ denote the propensity score. The discriminator of our GAN, given the covariates $X = x$, tries to distinguish between generated counterfactual data and ground truth factual data. The adversarial objective from Eq. 6 could be expanded with the law of total expectation wrt. $X$ and $A$ in the following way:

$$\mathbb{E}_{(X,A,M) \sim \mathbb{P}_f} \left[ \log \left( D(X, \tilde{G}(X, A, M))_A \right) \right] \tag{22}$$

$$= \mathbb{E}_{X \sim \mathbb{P}(X)} \mathbb{E}_{(A,M) \sim \mathbb{P}(A,M|X)} \left[ \log \left( D(X, \tilde{G}(X, A, M))_A \right) \right] \tag{23}$$

$$= \mathbb{E}_{X \sim \mathbb{P}(X)} \Big[ \mathbb{E}_{M \sim \mathbb{P}(M|X,A=0)} \left[ \log \left( D(X, \tilde{G}(X, 0, M))_0 \right) \right] \pi_0(X) \tag{24}$$

$$+ \mathbb{E}_{(M \sim \mathbb{P}(M|X,A=1))} \left[ \log \left( D(X, \tilde{G}(X, 1, M))_1 \right) \right] \pi_1(X) \Big]$$

$$= \mathbb{E}_{X \sim \mathbb{P}(X)} \Big[ \mathbb{E}_{M \sim \mathbb{P}(M|X,A=0)} \left[ \log \left( D(X, \{M, G(X, 0, M)_1\})_0 \right) \right] \pi_0(X) \tag{25}$$

$$+ \mathbb{E}_{M \sim \mathbb{P}(M|X,A=1)} \left[ \log \left( 1 - D(X, \{G(X, 1, M)_0, M\})_0 \right) \right] \pi_1(X) \Big].$$

Let $Z_0 = \{M, G(X, 0, M)_1\}$ and $Z_1 = \{G(X, 1, M)_0, M\}$ be two random variables. Then, using the law of the unconscious statistician, the expression can be converted to a weighted conditional GAN adversarial loss (Mirza & Osindero, 2014), i.e.,

$$\mathbb{E}_{X \sim \mathbb{P}(X)} \Big[ \mathbb{E}_{Z_0 \sim \mathbb{P}(Z_0|X,A=0)} \left[ \log \left( D(X, Z_0)_0 \right) \right] \pi_0(X) \tag{26}$$

$$+ \mathbb{E}_{Z_1 \sim \mathbb{P}(Z_1|X,A=1)} \left[ \log \left( 1 - D(X, Z_1)_0 \right) \right] \pi_1(X) \Big]$$

$$= \mathbb{E}_{X \sim \mathbb{P}(X)} \Big[ \int_{\mathcal{Z}} \Big( \log \left( D(X, z)_0 \right) \pi_0(X) \mathbb{P}(Z_0 = z \mid X, A = 0) \tag{27}$$

$$+ \log \left( 1 - D(X, z)_0 \right) \pi_1(X) \mathbb{P}(Z_1 = z \mid X, A = 1) \Big) \mathrm{d}z \Big],$$

where $\mathcal{Z} = \mathcal{M} \times \mathcal{M}$. Notably, the weights of the loss, i.e., $\pi_0(X)$ and $\pi_1(X)$, are greater than zero, due to the overlap assumption. The second term follows analogously. Following the theory from the standard GANs (Goodfellow et al., 2014b), for any $(a, b) \in \mathbb{R}^2 \setminus 0$, the function $y \mapsto \log(y)a + \log(1 - y)b$ achieves its maximum in $[0, 1]$ at $\frac{a}{a+b}$. Therefore, for a given generator, an optimal discriminator is

$$D(x, z)_0 = \frac{\mathbb{P}(Z_0 = z \mid X = x, A = 0) \pi_0(x)}{\mathbb{P}(Z_0 = z \mid X = x, A = 0) \pi_0(x) + \mathbb{P}(Z_1 = z \mid X = x, A = 1) \pi_1(x)}. \tag{28}$$

Both conditional densities used in the expression above can be expressed in terms of the potential outcomes densities due to the consistency and unconfoundedness assumptions, namely

$$\mathbb{P}(Z_0 = z \mid X = x, A = 0) = \mathbb{P}(\{M = m_0, G(x, 0, M)_1 = m_1\} \mid X = x, A = 0) \tag{29}$$

$$= \mathbb{P}(\{M_0 = m_0, G(x, 0, M_0)_1 = m_1\} \mid X = x),$$

$$\mathbb{P}(Z_1 = z \mid X = x, A = 1) = \mathbb{P}(\{G(x, 1, M)_0 = m_0, M = m_1\} \mid X = x, A = 1) \tag{30}$$

$$= \mathbb{P}(\{G(x, 1, M_1)_0 = m_0, M_1 = m_1\} \mid X = x).$$

Thus, an optimal generator of the GAN then minimizes the following conditional propensity-weighted Jensen–Shannon divergence (JSD)

$$\mathrm{JSD}_{\pi_0(x), \pi_1(x)} \Big( \mathbb{P}(\{M_0, G(x, 0, M_0)_1\} \mid X = x) \big\| \mathbb{P}(\{G(x, 1, M_1)_0, M_1\} \mid X = x) \Big), \tag{31}$$

---

[7]If the conditional density of the mediator has finite values.

where $\mathrm{JSD}_{w_1,w_2}(\mathbb{P}_1 \| \mathbb{P}_1) = w_1\,\mathrm{KL}(\mathbb{P}_1 \| w_1\,\mathbb{P}_1 + w_2\,\mathbb{P}_2) + w_2\,\mathrm{KL}(\mathbb{P}_2 \| w_1\,\mathbb{P}_1 + w_2\,\mathbb{P}_2)$ and where $\mathrm{KL}(\mathbb{P}_1 \| \mathbb{P}_1)$ is Kullback–Leibler divergence. The Jensen–Shannon divergence is minimized, when $G(x,0,M_0)_1 = M_1$ and $G(x,1,M_1)_0 = M_0$ conditioned on $X = x$ (in distribution), since, in this case, it equals to zero, i.e.,

$$\mathbb{P}(G(x,a,M_a)_{a'} \mid X = x) = \mathbb{P}(M_{a'} \mid X = x). \tag{32}$$

Finally, due to the unconfoundedness assumption, the generator of our GAN estimates the potential mediator distributions with counterfactual sensitive attributes, i.e.,

$$\mathbb{P}(G(x,a,M_a)_{a'} \mid X = x, A = a) = \mathbb{P}(M_{a'} \mid X = x, A = a) \tag{33}$$

in distribution.

(ii) For a given factual observation, $X = x, A = a, M = m$, our generator yields a deterministic output, i.e.,

$$\mathbb{P}(G(x,a,M_a)_{a'} \mid X = x, A = a, M = m) = \mathbb{P}(G(x,a,m)_{a'} \mid X = x, A = a, M = m) \tag{34}$$

$$= \delta(G(x,a,m)_{a'}). \tag{35}$$

At the same time, this counterfactual distribution is connected with the potential mediators' distributions with counterfactual sensitive attributes, $\mathbb{P}(M_{a'} = m' \mid X = x, A = a)$, via the law of total probability:

$$\mathbb{P}(M_{a'} = m' \mid X = x, A = a) = \mathbb{P}(G(x,a,M)_{a'} = m' \mid X = x, A = a) \tag{36}$$

$$= \int_{\mathcal{M}} \delta(G(x,a,m)_{a'} - m')\,\mathbb{P}(M = m \mid X = x, A = a)\,\mathrm{d}m \tag{37}$$

$$= \sum_{m:\,G(x,a,m)_{a'}=m'} |\nabla_m G(x,a,m)_{a'}|^{-1}\,\mathbb{P}(M = m \mid X = x, A = a). \tag{38}$$

Due to the unconfoundedness and the consistency assumptions, this is equivalent to

$$\mathbb{P}(M = m' \mid X = x, A = a') = \sum_{m:\,G(x,a,m)_{a'}=m'} |\nabla_m G(x,a,m)_{a'}|^{-1}\,\mathbb{P}(M = m \mid X = x, A = a). \tag{39}$$

The equation above has only two solutions wrt. $G(x,a,\cdot)$ in the class of the continuously differentiable functions (Corollary 3 in (Melnychuk et al., 2023)), namely:[8]

$$G(x,a,m)_{a'} = \mathbb{F}^{-1}(\pm\mathbb{F}(m;x,a) \mp 0.5 + 0.5; x, a'), \tag{40}$$

where $\mathbb{F}(\cdot; x, a)$ and $\mathbb{F}^{-1}(\cdot; x, a)$ are a CDF and an inverse CDF of $\mathbb{P}(M \mid X = x, A = a)$. Thus, the generator of GAN exactly matches one of the two BGM solutions from (i). This concludes that our generator consistently estimates the counterfactual distribution of the mediator, $\mathbb{P}(M_{a'} \mid X = x, A = a, M = m)$. $\qquad\square$

**Corollary 1.** *The results of the Lemma 4 naturally generalize to sensitive attributes with more categories, i.e., $\mathcal{A} = \{0, 1, \dots, k-1\}, k > 2$.*

*Proof.* We want to show that, when $\mathcal{A} = \{0, 1, \dots, k-1\}, k > 2$, the generator is still able to learn the potential mediator distributions with the counterfactual distributions. For that, we follow the same derivation steps, as in part (i) of the proof of Lemma 4. This brings us to the following equality for the loss of the discriminator:

$$\mathbb{E}_{(X,A,M)\sim\mathbb{P}_{\mathrm{f}}} \left[ \log\left(D(X, \tilde{G}(X, A, M))_A\right) \right] \tag{41}$$

$$=\mathbb{E}_{X\sim\mathbb{P}(X)} \Bigg[ \int_{\mathcal{Z}} \Big( \log\left(D(X, z)_0\right) \pi_0(X)\mathbb{P}(Z_0 = z \mid X, A = 0) \tag{42}$$

$$+ \log\left(D(X, z)_1\right) \pi_1(X)\mathbb{P}(Z_1 = z \mid X, A = 1) \tag{43}$$

$$\dots \tag{44}$$

$$+ \log\left(D(X, z)_{k-2}\right) \pi_{k-2}(X)\mathbb{P}(Z_{k-2} = z \mid X, A = k-2) \tag{45}$$

$$+ \log\left(1 - \sum_{j=0}^{k-2} D(X, z)_j\right) \pi_{k-1}(X)\mathbb{P}(Z_{k-1} = z \mid X, A = k-1) \Big)\,\mathrm{d}z \Bigg], \tag{46}$$

---

[8]Under mild conditions, the counterfactual distributions cannot be defined via the point mass distribution with non-monotonous functions, even if we assume the extension of BGMs to all non-monotonous continuously differentiable functions.

where

$$Z_0 = \{M, G(X, 0, M)_1, G(X, 0, M)_2, \ldots, G(X, 0, M)_{k-1})\}, \tag{47}$$

$$Z_1 = \{G(X, 1, M)_0, M, G(X, 1, M)_2, \ldots, G(X, 1, M)_{k-1}\}, \tag{48}$$

$$\ldots \tag{49}$$

$$Z_{k-1} = \{G(X, k-1, M)_0, G(X, k-1, M)_1, \ldots, M\}. \tag{50}$$

Then, it is easy to see that, for a given generator, an optimal discriminator is (analogously to Eq. 28)

$$D(x, z)_a = \frac{\mathbb{P}(Z_a = z \mid X = x, A = a)\, \pi_a(x)}{\sum_{j=0}^{k-1} \mathbb{P}(Z_j = z \mid X = x, A = j)\, \pi_j(x)} \quad \text{for all } a \in \mathcal{A}. \tag{51}$$

This happens, as, for any $(a_0, \ldots, a_{k-1}) \in \mathbb{R}^k \smallsetminus 0$, the function $(y_0, y_1, \ldots y_{k-2}) \mapsto \log(y_0)a_0 + \log(y_1)a_1 + \cdots + \log(y_{k-2})a_{k-2} + \log(1 - \sum_{j=0}^{k-2} y_j)a_{k-1}$ achieves its maximum in $[0, 1]$ at $\left(\frac{a_0}{\sum_{j=0}^{k-1} a_j}, \frac{a_1}{\sum_{j=0}^{k-1} a_j}, \ldots, \frac{a_{k-2}}{\sum_{j=0}^{k-1} a_j}\right)$. Then, an optimal generator of the GAN aims to minimize the propensity-weighted multi-distribution JSD, i.e.,

$$\begin{aligned}
\mathrm{JSD}_{\pi_0(x), \pi_1(x), \ldots, \pi_{k-1}(x)} \Big( &\mathbb{P}(\{M_0, G(x, 0, M_0)_1, G(x, 0, M_0)_2, \ldots, G(x, 0, M_0)_{k-1}\} \mid X = x), \\
&\mathbb{P}(\{G(x, 1, M_1)_0, M_1, G(x, 1, M_1)_2, \ldots, G(x, 1, M_1)_{k-1}\} \mid X = x), \\
&\ldots \\
&\mathbb{P}(\{G(x, k-1, M_{k-1})_0, G(x, k-1, M_{k-1})_1, \ldots, M_{k-1}\} \mid X = x)\Big).
\end{aligned} \tag{52}$$

The JSD is minimized, when all the distributions are equal. If we additionally look at the marginalized distributions, the following equalities will hold

$$\mathbb{P}(G(x, a, M_a)_{a'} \mid X = x) = \mathbb{P}(M_{a'} \mid X = x) \quad \text{for all } a \neq a' \in \mathcal{A}. \tag{53}$$

This concludes the proof of the Corollary, as all additional steps are analogous to the Lemma 4. $\qquad\square$

**Remark 2.** *We proved that the generator converges to one of the two BGM solutions in Eq. 21. Which solution the generator exactly returns depends on the initialization and the optimizer. Notably, the difference between the two solutions is negligibly small, when the variability of the mediator is low. To demonstrate this, we assume (without the loss of generality) that the ground-truth counterfactual mediator follows one of the BGM solutions, e.g., $\mathbb{P}(M_{a'} \mid X = x, A = a, M = m) = \delta(\mathbb{F}^{-1}(\mathbb{F}(m; x, a); x, a'))$; and our GAN estimates another, i.e., $\mathbb{P}(M_{a'} \mid X = x, A = a, M = m) = \delta(\mathbb{F}^{-1}(1 - \mathbb{F}(m; x, a); x, a'))$. Then, assuming a perfect fit of the GAN, the conditional expectation of the squared difference between ground-truth counterfactual mediator and estimated mediator is*

$$\mathbb{E}\left[\left\|M_{A'} - \hat{M}_{A'}\right\|_2^2 \mid X = x, A = a\right] \tag{54}$$

$$= \mathbb{E}\left[\left|\mathbb{F}^{-1}(\mathbb{F}(M; x, a); x, a') - \mathbb{F}^{-1}(1 - \mathbb{F}(M; x, a); x, a')\right| \mid X = x, A = a\right] \tag{55}$$

$$= \mathbb{E}\left[\left|\mathbb{F}^{-1}(U; x, a') - \mathbb{F}^{-1}(1 - U; x, a')\right|\right] \tag{56}$$

$$= \int_0^1 \left|\mathbb{F}^{-1}(u; x, a') - \mathbb{F}^{-1}(1 - u; x, a')\right| du \tag{57}$$

$$\leq \int_0^1 \left|\mathbb{F}^{-1}(u; x, a') - \mu(x, a')\right| du + \int_0^1 \left|\mathbb{F}^{-1}(1 - u; x, a') - \mu(x, a')\right| du \tag{58}$$

$$= 2\,\mathbb{E}\left[\left|M - \mu(x, a')\right| \mid X = x, A = a'\right] \tag{59}$$

$$\overset{(*)}{\leq} 2\sqrt{\mathrm{Var}\left[M \mid X = x, A = a'\right]}, \tag{60}$$

*where $(*)$ holds as an inequality between the mean absolute deviation and the standard deviation. This result also holds for high-dimensional mediators, where there is a continuum of solutions in the class of continuously differentiable functions (Chen & Gopinath, 2000). Thus, if the conditional standard deviation of the mediator is high, a combination of multiple GANs might be used to enforce a worst-case counterfactual fairness.*

# D  TRAINING ALGORITHM OF GCFN

---

**Algorithm 1** Training algorithm of GCFN

---

1: **Input.** Training dataset $\mathcal{D}$; fairness weight $\lambda$; number of training GAN epoch $e_1$; number of training prediction model epoch $e_2$; minibatch of size $n$; training supervised loss weight $\alpha$

2: **Init.** Generator $G$ parameters: $\theta_g$; discriminator $D$ parameters: $\theta_d$; prediction model $h$ parameters: $\theta_h$

3: *Step 1: Training GAN to learn to generate counterfactual mediator*

4: **for** $e_1$ **do**

5:     **for** $k$ steps **do**                                            ▷ Training the discriminator $D$

6:         Sample minibatch of $n$ examples $\left\{x^{(i)}, a^{(i)}, m^{(i)}\right\}_{i=1}^{n}$ from $\mathcal{D}$

7:         Compute generator output $G\left(x^{(i)}, a^{(i)}, m^{(i)}\right)_a = \hat{m}_a^{(i)}$   for $a \in \{0, 1\}$

8:         Modify $G$ output to $\tilde{G}_a^{(i)} = \begin{cases} m^{(i)}, & \text{if } a^{(i)} = a, \\ \hat{m}_a^{(i)}, & \text{if } a^{(i)} = a' \end{cases}$   for $a \in \{0, 1\}$

9:         Update the discriminator via stochastic gradient ascent

$$\nabla_{\theta_d} \frac{1}{n} \sum_{i=1}^{n} \left[ \log\left( D(x^{(i)}, \tilde{G}^{(i)})_{a^{(i)}} \right) \right]$$

10:     **end for**

11:     **for** $k$ steps **do**                                              ▷ Training the generator $G$

12:         Sample minibatch of $n$ examples $\left\{x^{(i)}, a^{(i)}, m^{(i)}\right\}_{i=1}^{n}$ from $\mathcal{D}$

13:         Compute generator output $G\left(x^{(i)}, a^{(i)}, m^{(i)}\right)_a = \hat{m}_a^{(i)}$   for $a \in \{0, 1\}$

14:         Modify $G$ output to $\tilde{G}_a^{(i)} = \begin{cases} m^{(i)}, & \text{if } a^{(i)} = a, \\ \hat{m}_a^{(i)}, & \text{if } a^{(i)} = a' \end{cases}$   for $a \in \{0, 1\}$

15:         Update the generator via stochastic gradient descent

$$\nabla_{\theta_g} \frac{1}{n} \sum_{i=1}^{n} \left[ \log\left( D(x^{(i)}, \tilde{G}^{(i)})_{a^{(i)}} \right) + \log\left( 1 - D(x^{(i)}, \tilde{G}^{(i)})_{1-a^{(i)}} \right) + \alpha \left\| m^{(i)} - G\left(x^{(i)}, a^{(i)}, m^{(i)}\right)_{a^{(i)}} \right\|_2^2 \right]$$

16:     **end for**

17: **end for**

18: *Step 2: Training prediction model with counterfactual mediator regularization*

19: **for** $e_2$ **do**                                                 ▷ Training the prediction model $h$

20:     Sample minibatch of $n$ examples $\left\{x^{(i)}, a^{(i)}, m^{(i)}, y^{(i)}\right\}_{i=1}^{n}$ from $\mathcal{D}$

21:     Generate $\hat{m}^{(i)}$ from $G\left(x^{(i)}, a^{(i)}, m^{(i)}\right)$

22:     Compute counterfactual mediator regularization

$$\mathcal{R}_{\text{cm}} = \left\| h(x^{(i)}, m^{(i)}) - h(x^{(i)}, \hat{m}_{a'^{(i)}}^{(i)}) \right\|_2^2$$

23:     Update the prediction model via stochastic gradient descent

$$\nabla_{\theta_h} \frac{1}{n} \sum_{i=1}^{n} \left[ y^{(i)} \log\left( h(x^{(i)}, m^{(i)}) \right) + (1 - y^{(i)}) \log\left( 1 - h(x^{(i)}, m^{(i)}) \right) + \lambda \mathcal{R}_{\text{cm}} \right]$$

24: **end for**

25: **Output.** Counterfactually fair prediction model $h$

---

# E  DATASET

## E.1  SYNTHETIC DATA

Analogous to prior works that simulate synthetic data for benchmarking (Kim et al., 2021; Kusner et al., 2017; Quinzan et al., 2022), we generate our synthetic dataset in the following way. The covariates $X$ is drawn from a standard normal distribution $\mathcal{N}(0,1)$. The sensitive attribute $A$ follows a Bernoulli distribution with probability $p$, determined by a sigmoid function $\sigma$ of $X$ and a Gaussian noise term $U_A$. We then generate the mediator $M$ as a function of $X$, $A$, and a Gaussian noise term $U_M$. Finally, the target $Y$ follows a Bernoulli distribution with probability $p_y$, calculated by a sigmoid function of $X$, $M$, and a Gaussian noise term $U_Y$. $\beta_i$ ($i \in [1,6]$) are the coefficients. Let $\sigma(x) = \frac{1}{1+e^{-x}}$ represent the sigmoid function. Formally, we yield

$$
\begin{cases}
X = U_X & U_X \sim \mathcal{N}(0,1) \\
A \sim \text{Bernoulli}\left(\sigma(\beta_1 X + U_A)\right) & U_A \sim \mathcal{N}(0,0.01) \\
M = \beta_2 X + \beta_3 A + U_M & U_M \sim \mathcal{N}(0,0.01) \\
Y \sim \text{Bernoulli}\left(\sigma(\beta_5 X + \beta_6 M + U_Y)\right) & U_Y \sim \mathcal{N}(0,0.01)
\end{cases}
\tag{61}
$$

We sample 10,000 observations and use 20% as the test set.

## E.2  SEMI-SYNTHETIC DATA

**LSAC dataset.** The Law School (LSAC) dataset (Wightman, 1998) contains information about the law school admission records. We use the LSAC dataset to construct two semi-synthetic datasets. In both, we set the sensitive attribute to *gender*. We take *resident* and *race* from the LSAC dataset as confounding variables. The *LSAT* and *GPA* are the mediator variables, and the *admissions decision* is our target variable. We simulate 101,570 samples and use 20% as the test set. We denote $M_1$ as GPA score, $M_2$ as LSAT score, $X_1$ as resident, and $X_2$ as race. Further, $w_{X_1}, w_{X_2}, w_A, w_{M_1}, w_{M_2}$ are the coefficients. $U_{M_1}, U_{M_2}, U_Y$ are the Gaussian noise. Let $\sigma(x) = \frac{1}{1+e^{-x}}$ represent the sigmoid function. Note that the generation of $M$ in our datasets is non-deterministically.

We then produce the two different semi-synthetic datasets as follows. The main difference is whether we use a rather simple sigmoid function or a complex sinus function that could make extrapolation more challenging for our GCFN.

■ Semi-synthetic dataset "sigmoid":

$$
\begin{cases}
M_1 = w_{M_1}\left(\sigma(w_A A + w_{X_1} X_1 + w_{X_2} X_2 + U_{M_1})\right) & U_{M_1} \sim \mathcal{N}(0,0.01) \\
M_2 = w_{M_2} + w_{M_1}\left(\sigma(w_A S + w_{X_1} X_1 + w_{X_2} X_2 + U_{M_2})\right) & U_{M_2} \sim \mathcal{N}(0,0.01) \\
Y \sim \text{Bernoulli}\left(\sigma(w_{M_1} M_1 + w_{M_2} M_2 + w_{X_1} X_1 + w_{X_2} X_2 + U_Y)\right) & U_Y \sim \mathcal{N}(0,0.01)
\end{cases}
\tag{62}
$$

■ Semi-synthetic "sin":

$$
\begin{cases}
M_1 = w_A \cdot A - \sin\left(\pi \times (w_{X_1} X_1 + w_{X_2} X_2 + U_{M_1})\right) & U_{M_1} \sim \mathcal{N}(0,0.01) \\
M_2 = w_A \cdot A - \sin\left(\pi \times (w_{X_1} X_1 + w_{X_2} X_2 + U_{M_2})\right) & U_{M_2} \sim \mathcal{N}(0,0.01) \\
Y \sim \text{Bernoulli}\left(\sigma(w_{M_1} M_1 + w_{M_2} M_2 + w_{X_1} X_1 + w_{X_2} X_2 + U_Y)\right) & U_Y \sim \mathcal{N}(0,0.01)
\end{cases}
\tag{63}
$$

## E.3  REAL-WORLD DATA

**UCI Adult dataset:** The UCI Adult dataset (Asuncion & Newman, 2007) captures information about 48,842 individuals including their sociodemographics. Our aim is to predict if individuals earn more than USD 50k per year. We follow the setting of earlier research (Kim et al., 2021; Nabi & Shpitser, 2018; Quinzan et al., 2022; Xu et al., 2019). We treat *gender* as the sensitive attribute and set mediator variables to be *marital status*, *education level*, *occupation*, *hours per week*, and *work class*. The causal graph of the UCI dataset is in Fig. 8. We take 20% as the test set.

**COMPAS dataset:** COMPAS (Correctional Offender Management Profiling for Alternative Sanctions) (Angwin et al., 2016) was developed as a decision support tool to score the likelihood of a person's recidivism. The score ranges from 1 (lowest risk) to 10 (highest risk). The dataset further contains information about whether there was an actual recidivism (reoffended) record within 2

years after the decision. Overall, the dataset has information about over 10,000 criminal defendants in Broward County, Florida. We treat *race* as the sensitive attribute. The mediator variables are the features related to prior convictions and current charge degree. The target variable is the *recidivism* for each defendant. The causal graph of the COMPAS dataset is in Fig. 9. We take 20% as test set.

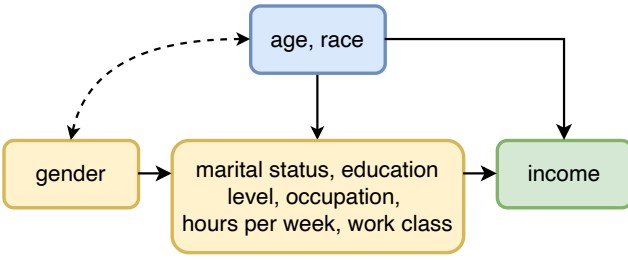

Figure 8: Causal graph of UCI dataset.

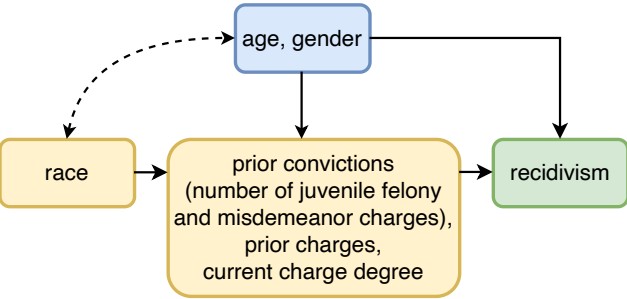

Figure 9: Causal graph of COMPAS dataset.

In practice, it is common and typically straightforward to choose which variables act as mediators $M$ through domain knowledge (Nabi & Shpitser, 2018; Kim et al., 2021; Plecko & Bareinboim, 2022). Hence, mediators $M$ are simply all variables that can potentially be influenced by the sensitive attribute. All other variables (except for $A$ and $Y$) are modeled as covariates $X$. For example, consider a job application setting where we want to avoid discrimination by gender. Then $A$ is gender, and $Y$ is the job offer. Mediators are, for instance, education level or work experience, as both are potentially influenced by gender. In contrast, age is a covariate because it is not influenced by gender.

# F    IMPLEMENTATION DETAILS

## F.1    IMPLEMENTATION OF BENCHMARKS

We implement **CFAN** (Kusner et al., 2017) in PyTorch based on the paper's source code in R and Stan on `https://github.com/mkusner/counterfactual-fairness`. We use a VAE to infer the latent variables. For **mCEVAE** (Pfohl et al., 2019), we follow the implementation from `https://github.com/HyemiK1m/DCEVAE/tree/master/Tabular/mCEVAE_baseline`. We implement **CFGAN** (Xu et al., 2019) in PyTorch based on the code of Abroshan et al. (2022) and the TensorFlow source code of (Xu et al., 2019). We implement **ADVAE** (Grari et al., 2023) in PyTorch. For **DCEVAE** (Kim et al., 2021), we use the source code of the author of **DCEVAE** (Kim et al., 2021). We use **HSCIC** (Quinzan et al., 2022) source implementation from the supplementary material provided on the OpenReview website `https://openreview.net/forum?id=ERjQnrmLKH4`. We performed rigorous hyperparameter tuning for all baselines.

**Hyperparameter tuning.** We perform a rigorous procedure to optimize the hyperparameters for the different methods as follows. For DCEVAE (Kim et al., 2021) and mCEVAE (Pfohl et al., 2019), we follow the hyperparameter optimization as described in the supplement of Kim et al. (2021). For ADVAE (Grari et al., 2023) and CFGAN (Xu et al., 2019), we follow the hyperparameter optimization as described in their paper. For both HSCIC and our GCFN, we have an additional weight that introduces a trade-off between accuracy and fairness. This provides additional flexibility to decision-makers as they tailor the methods based on the fairness needs in practice (Quinzan et al., 2022). We then benchmark the utility of different methods across different choices of $\gamma$ of the utility function in Sec. 5.1. This allows us thus to optimize the trade-off weight $\lambda$ inside HSCIC and our GCFN using grid search. For HSCIC, we experiment with $\lambda = 0.1, 0.5, 1, 5, 10, 15, 20$ and choose the best for them across different datasets. For our method, we experiment with $\lambda = 0.1, 0.5, 1, 1.5, 2$. Since the utility function considers two metrics, across the experiments on (semi-)synthetic dataset, the weight $\lambda$ is set to $0.5$ to get a good balance for our method.

## F.2    IMPLEMENTATION OF OUR METHOD

Our GCFN is implemented in PyTorch. Both the generator and the discriminator in the GAN model are designed as deep neural networks, each with a hidden layer of dimension $64$. LeakyReLU is employed as the activation function and batch normalization is applied in the generator to enhance training stability. The GAN training procedure is performed for $300$ epochs with a batch size of $256$ at each iteration. We set the learning rate to $0.0005$. Following the GAN training, the prediction model, structured as a multilayer perceptron (MLP), is trained separately. This classifier can incorporate spectral normalization in its linear layers to ensure Lipschitz continuously. It is trained for $30$ epochs, with the same learning rate of $0.005$ applied. The training time of our GCFN on (semi-) synthetic dataset is comparable to or smaller than the baselines.

## G  GENERALIZATION TO MULTIPLE SOCIAL GROUPS

### G.1  STEP 1: GAN FOR GENERATING COUNTERFACTUAL OF THE MEDIATOR

Our method can easily extended to scenarios with multiple social groups. Suppose we have $k$ categories, then the sensitive attribute $A \in \mathcal{A}$, where $\mathcal{A} = \{0, 1, \ldots, k-1\}$ and $k > 2$.

The output of the generator $G$ is $k$ potential mediators, i.e., $\hat{M}_0, \hat{M}_1, \ldots, \hat{M}_{k-1}$, from which one is factual and the others are counterfactual.

$$G(X, A, M)_a = \hat{M}_a \quad \text{for} \quad a \in \{0, 1, ..., k-1\} \tag{64}$$

The reconstruction loss of the generator is the same as the binary case,

$$\mathcal{L}_{\text{f}}(G) = \mathbb{E}_{(X,A,M)\sim\mathbb{P}_{\text{f}}} \left[ \|M - G(X, A, M)_A\|_2^2 \right], \tag{65}$$

where $\|\cdot\|_2$ is the $L_2$-norm.

The discriminator $D$ is designed to differentiate the factual mediator $M$ (as observed in the data) from the $k-1$ generated counterfactual mediators (as generated by $G$).

We modify the output of $G$ before passing it as input to $D$: We replace the generated factual mediator $\hat{M}_A$ with the observed factual mediator $M$. We denote the new, combined data by $\tilde{G}(X, A, M)$, which is defined via

$$\tilde{G}(X, A, M)_a = \begin{cases} M, & \text{if } A = a, \\ G(X, A, M)_a, & \text{Otherwise}, \end{cases} \quad \text{for } a \in \{0, 1, ..., k-1\}. \tag{66}$$

The discriminator $D$ then determines which component of $\tilde{G}$ is the observed factual mediator and thus outputs the corresponding probability. Formally, for the input $(X, \tilde{G})$, the output of the discriminator $D$ is

$$D(X, \tilde{G})_a = \hat{\mathbb{P}}(M = \tilde{G}_a \mid X, \tilde{G}) = \hat{\mathbb{P}}(A = a \mid X, \tilde{G}) \quad \text{for } a \in \{0, 1, ..., k-1\}. \tag{67}$$

Our GAN is trained in an adversarial manner: (i) the generator $G$ seeks to generate counterfactual mediators in a way that minimizes the probability that the discriminator can differentiate between factual mediators and counterfactual mediators, while (ii) the discriminator $D$ seeks to maximize the probability of correctly identifying the factual mediator. We thus use an adversarial loss $\mathcal{L}_{\text{adv}}$ by

$$\mathcal{L}_{\text{adv}}(G, D) = \mathbb{E}_{(X,A,M)\sim\mathbb{P}_{\text{f}}} \left[ \log \left( D(X, \tilde{G}(X, A, M))_A \right) \right]. \tag{68}$$

Overall, our GAN is trained through an adversarial training procedure with a minimax problem as

$$\min_G \max_D \mathcal{L}_{\text{adv}}(G, D) + \alpha \mathcal{L}_{\text{f}}(G), \tag{69}$$

with a hyperparameter $\alpha$ on $\mathcal{L}_{\text{f}}$.

### G.2  STEP 2: COUNTERFACTUAL FAIR PREDICTION THROUGH COUNTERFACTUAL MEDIATOR REGULARIZATION

We use the output of the GAN to train a prediction model $h$ under counterfactual fairness in a supervised way. Our counterfactual mediator regularization $\mathcal{R}_{\text{cm}}(h)$ thus is

$$\mathcal{R}_{\text{cm}}(h) = \mathbb{E}_{(X,A,M)\sim\mathbb{P}_{\text{f}}} \left[ \frac{1}{(k-1)} \sum_{\substack{a=0 \\ a \neq A}}^{k-1} \|h(X, M) - h(X, \hat{M}_a)\|_2^2 \right]. \tag{70}$$

The training loss is

$$\mathcal{L}(h) = \mathcal{L}_{\text{ce}}(h) + \lambda \mathcal{R}_{\text{cm}}(h). \tag{71}$$

### G.3 THEORETICAL INSIGHTS

We provide proof that our method can naturally generalize to sensitive attributes with more categories in Appendix. C, Corollary. 1.

### G.4 RESULTS GENERALIZATION TO MULTIPLE SOCIAL GROUPS

Since some baselines are not adaptable to various social groups, they cannot be benchmarked effectively in this context. Consequently, our focus is on demonstrating the generalization capabilities of our method across multiple social groups. We carried out experiments on the synthetic dataset where $k = 4$. Our GCFN attained an accuracy of approximately 0.92.

# H ADDITIONAL EXPERIMENTAL RESULTS

## H.1 RESULTS FOR THE PRESENCE OF CONFOUNDERS BETWEEN COVARIATES AND THE SENSITIVE ATTRIBUTE

In our causal graph Fig. 1, we allow the potential existence of the hidden confounders between covariates $X$ and the sensitive attribute $A$. We now perform additional experiments where explore the performance in the presence of confounders, and, to this end, we intentionally introduce confounders into our semi-synthetic dataset, and the corresponding causal graph is shown Fig. 10. The experiment results are shown in Fig. 11. The result shows that having correlations between $X$ and $A$ does not affect the counterfactual level in our prediction in $Y$, which is consistent with our setting.

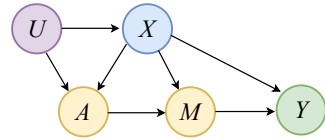

Figure 10: Causal graph in the presence of confounders between covariates and the sensitive attribute.

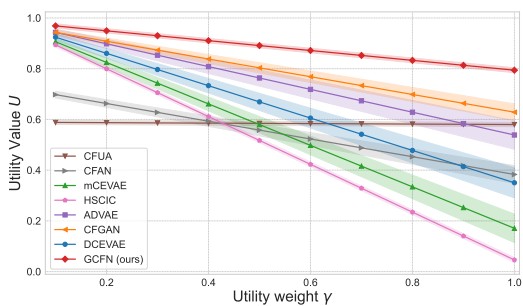

Figure 11: Utility function value $U$ on dataset contains correlations between the covariate $X$ and sensitive attribute $A$. A larger utility is better. Shown: mean ± std over 5 runs.

## H.2 RESULTS FOR REAL-WORLD DATASET

We now provide an additional analysis for the UCI real-world dataset. Here, we seek to benchmark the performance of the different methods in terms of utility. However, by definition, the true counterfactuals for real-world datasets are unavailable, which naturally arises in causal inference. As a remedy, we use our generated counterfactual as the ground-truth counterfactual on the test dataset. This then allows us to report the utility function value analogous to synthetic datasets. The experiment results are shown in Fig. 12. We again find that our method is highly effective.

## H.3 RESULTS FOR (SEMI-)SYNTHETIC DATASET

We compute the average value of the utility function $U$ over varying utility weights $\gamma \in \{0.1, \ldots, 1.0\}$ on the synthetic dataset (Fig. 13) and two different semi-synthetic datasets (Fig. 14 and Fig. 15).

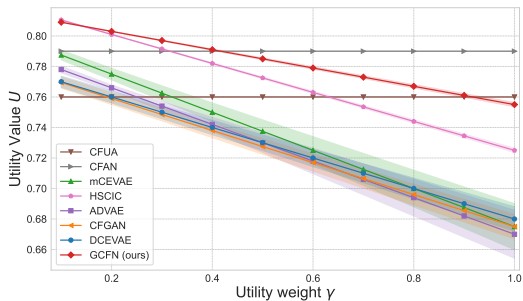

Figure 12: Average utility function value $U$ across different utility weights $\gamma$ on synthetic dataset.

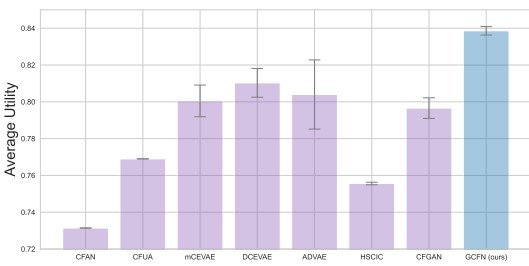

Figure 13: Average utility function value $U$ across different utility weights $\gamma$ on synthetic dataset.

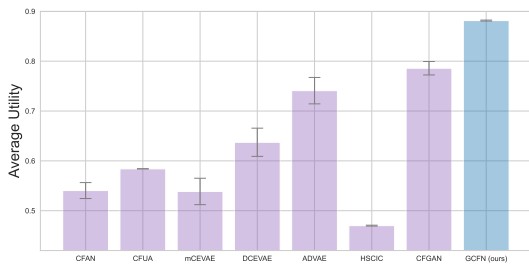

Figure 14: Average utility function value $U$ across different utility weight $\gamma$ on semi-synthetic (sigmoid) dataset.

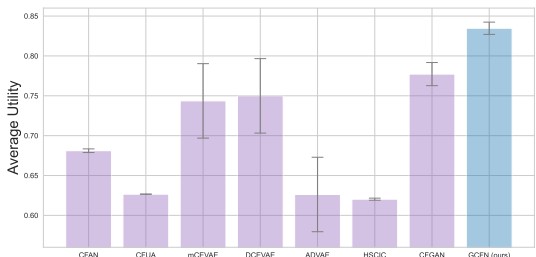

Figure 15: Average utility function value $U$ across different utility weight $\gamma$ on semi-synthetic (sin) dataset.

### H.4 Results for UCI Adult dataset

We now examine the results for different fairness weights $\lambda$. For this, we report results from $\lambda = 0.5$ (Fig. 16) to $\lambda = 1000$ (Fig. 19). In line with our expectations, we see that larger values for fairness weight $\lambda$ lead the distributions of the predicted target to overlap more, implying that counterfactual fairness is enforced more strictly. This shows that our regularization $\mathcal{R}_{\mathrm{cm}}$ achieves the desired behavior.

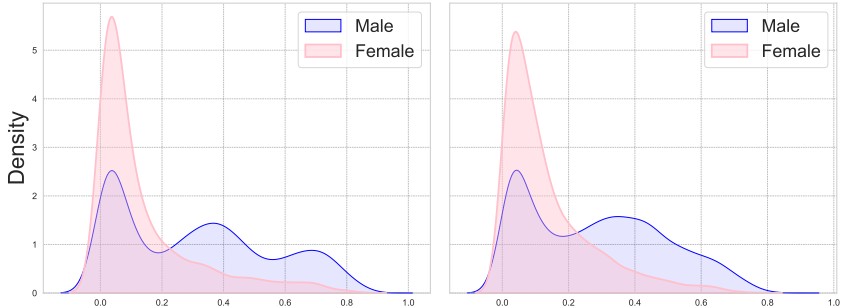

Figure 16: Density plots of the predicted target on UCI Adult dataset. Left: fairness weight $\lambda = 0$. Right: fairness weight $\lambda = 0.5$.

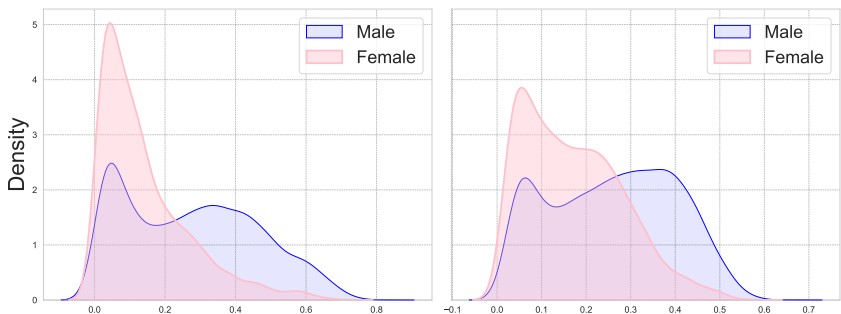

Figure 17: Density plots of the predicted target on UCI Adult dataset. Left: fairness weight $\lambda = 1$. Right: fairness weight $\lambda = 5$.

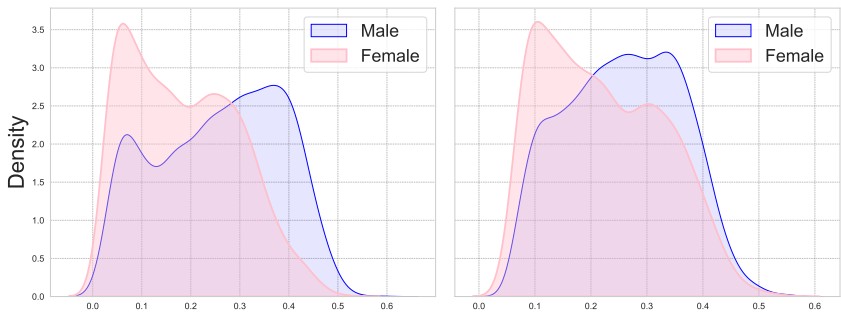

Figure 18: Density plots of the predicted target on UCI Adult dataset. Left: fairness weight $\lambda = 10$. Right: fairness weight $\lambda = 100$.

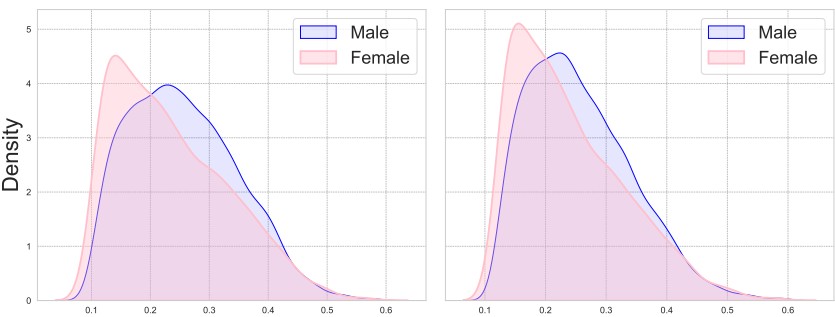

Figure 19: Density plots of the predicted target on UCI Adult dataset. Left: fairness weight $\lambda = 500$. Right: fairness weight $\lambda = 1000$.

## H.5 RESULTS FOR COMPAS DATASET

In Sec. 5, we show how black defendants are treated differently by the COMPAS score vs. our GCFN. Here, we also show how white defendants are treated differently by the COMPAS score vs. our GCFN; see Fig. 20. We make the following observations. (1) Our GCFN makes oftentimes different predictions for white defendants with a low and high COMPAS score, which is different from black defendants. (2) Our method also arrives at different predictions for white defendants with low prior charges, similar to black defendants.

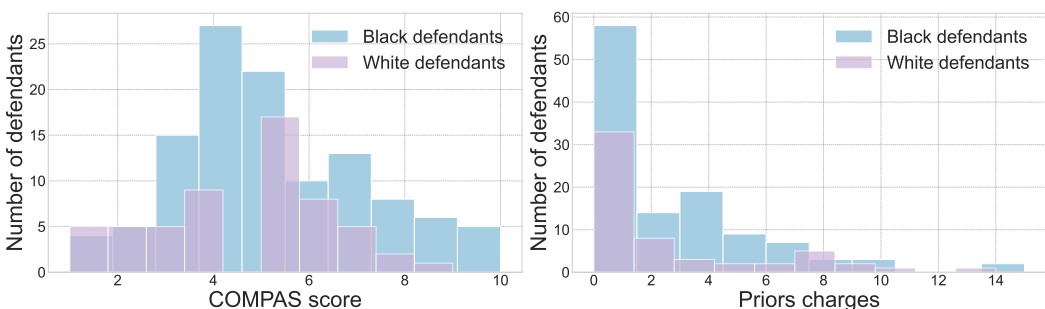

Figure 20: Distribution of white and black defendants that are treated differently using our GCFN. Left: COMPAS score. Right: Prior charges.

# I   DETAILS ON BASELINE METHODS

We provide details for CFGAN (Xu et al., 2019) and ADVAE (Grari et al., 2023) in the following. Note that these two methods aim at a different task and a different causal graph setting, respectively. Hence, we also lay out below how we adapted them to make them applicable for comparison in our experiments.

## I.1   CFGAN

Note that CFGAN (Xu et al., 2019) is designed for fair data generation tasks, while our model is designed for learning predictors to achieve counterfactual fairness. Hence, both address different tasks. We made CFGAN applicable for a baseline by first generating the synthetic dataset and then training a predictor on this dataset.

In CFGAN (Xu et al., 2019), what is referred to as $S$ aligns with the sensitive attribute $A$ in our paper, and the target variable is both denoted as $Y$. Since they do not specify the causal graph for their dataset, we align the set of all other variables $\mathbf{X}$ in CFGAN with covariates $X$ plus mediators $M$ in our paper. Below, we use the notation from CFGAN for ease of description of their method.

The goal of CFGAN is to generate new data $(\hat{\mathbf{x}}, \hat{y}, \hat{s})$ which maintains the distribution of all attributes in the real data and ensures that the generated data $\hat{S}$ has no discriminatory effect on $\hat{Y}$. The variables with hat denote the fake data generated by the generator. CFGAN considers $S$ as a binary variable, where $s^+$ denotes $S = 1$ and $s^-$ denotes $S = 0$.

CFGAN adopts two generators $(G^1, G^2)$ and two discriminators $(D^1, D^2)$. The generator $G^1$ aims to mimic the real observational distribution, and the generator $G^2$ aims to generate interventional data. The discriminator $D^1$ tries to distinguish the generated data from the real data, and the discriminator $D^2$ tries to distinguish the two intervention distributions under $\mathrm{do}\,(S = s^+)$ and $\mathrm{do}\,(S = s^-)$.

The generator $G^1$ is designed to agree with the causal graph $\mathcal{G} = (\mathbf{V}, \mathbf{E})$. It consists of $|\mathbf{V}|$ sub-neural networks, where each of them corresponds to a node in $\mathbf{V}$. All sub-neural networks are connected following the connections in $\mathcal{G}$. The generator $G^2$ is designed to agree with the interventional graph $\mathcal{G}_s = \left(\mathbf{V}, \mathbf{E} \setminus \{V_j \to S\}_{V_j \in \mathbf{Pa}_S}\right)$, where all incoming edges to $S$ are deleted under intervention $\mathrm{do}(S = s)$. The structure of $G^2$ is similar to that of $G^1$, except for that the sub-neural network $G^2_S$ is set to $G^2_S \equiv 1$ if $s = s^+$, and $G^2_S \equiv 0$ if $s = s^-$.

After training, $G^1$ should generate samples from the observational distribution, and $G^2$ generates samples from two interventional distributions, i.e., $(\hat{\mathbf{x}}, \hat{y}, \hat{s}) \sim P_{G^1}(\mathbf{X}, Y, S)$, $(\hat{\mathbf{x}}_{s^+}, \hat{y}_{s^+}) \sim P_{G^2}(\mathbf{X}_{s^+}, Y_{s^+})$, if $s = s^+$, $(\hat{\mathbf{x}}_{s^-}, \hat{y}_{s^-})\,P_{G^2}(\mathbf{X}_{s^-}, Y_{s^-})$, if $s = s^-$. The discriminator $D^1$ is designed to distinguish between the real observational data $(\mathbf{x}, y, s) \sim P_{\mathrm{data}}(\mathbf{X}, Y, S)$ and the generated fake observational data $(\hat{\mathbf{x}}, \hat{y}, \hat{s}) \sim P_{G^1}(\mathbf{X}, Y, S)$. The discriminator $D^2$ is designed to distinguish between the two interventional distributions $\hat{y}_{s^+} \sim P_{G^2}(Y_{s^+})$ and $\hat{y}_{s^-} \sim P_{G^2}(Y_{s^-})$.

During training, the generator $G^1$ plays an adversarial game with the discriminator $D^1$, and generator $G^2$ plays an adversarial game with the discriminator $D^2$. The overall minimax game is

$$\min_{G^1, G^2} \max_{D^1, D^2} J\left(G^1, G^2, D^1, D^2\right) = J_1\left(G^1, D^1\right) + \lambda J_2\left(G^2, D^2\right), \tag{72}$$

where

$$J_1\left(G^1, D^1\right) = \mathbb{E}_{(\mathbf{x}, y, s) \sim P_{\mathrm{data}}(\mathbf{X}, Y, S)}\left[\log D^1(\mathbf{x}, y, s)\right]$$
$$+ \mathbb{E}_{(\hat{\mathbf{x}}, \hat{y}, \hat{s}) \sim P_{G^1}(\mathbf{X}, Y, S)}\left[1 - \log D^1(\hat{\mathbf{x}}, \hat{y}, \hat{s})\right]$$
$$J_2\left(G^2, D^2\right) = \mathbb{E}_{\hat{y}_{s^+} \sim P_{G^2}(Y_{s^+})}\left[\log D^2\left(\hat{y}_{s^+}\right)\right]$$
$$+ \mathbb{E}_{\hat{y}_{s^-} \sim P_{G^2}(Y_{s^-})}\left[1 - \log D^2\left(\hat{y}_{s^-}\right)\right]$$

and $\lambda$ is a hyperparameter that controls a trade-off between the utility and the fairness of data generation. The first function $J_1$ aims to achieve $P_{G^1}(\mathbf{X}, Y, S) = P_{\mathrm{data}}(\mathbf{X}, Y, S)$. The second function $J_2$ aims to achieve $P_{G^2}(Y_{s^+}) = P_{G^2}(Y_{s^-})$. In generating a dataset for counterfactual fairness, the intervention is performed by conditioning on a subset of variables $\mathbf{O} = \mathbf{o}$. For each noise vector $\mathbf{z}$, CFGAN first generates the observational sample by using $G^1$, and observes whether in the sample they have $\mathbf{O} = \mathbf{o}$. Only for those noise vectors with $\mathbf{O} = \mathbf{o}$ in the generated samples, they generate

interventional samples using $G^2$. The interventional distribution generated by $G^2$ is conditioned on $\mathbf{O} = \mathbf{o}$, denoted by $P_{G^2}(\mathbf{X}_s, Y_s \mid \mathbf{o})$. The discriminator $D^2$ is designed to distinguish between $\hat{y}_{s^+} \mid \mathbf{o} \sim P_{G^2}(Y_{s^+} \mid \mathbf{o})$ and $\hat{y}_{s^-} \mid \mathbf{o} \sim P_{G^2}(Y_{s^-} \mid \mathbf{o})$, producing the function that aims to achieve $P_{G^2}(Y_{s^+} \mid \mathbf{o}) = P_{G^2}(Y_{s^-} \mid \mathbf{o})$.

After generating the synthetic dataset, we train a predictor $h$ on the data by minimizing the cross-entropy loss for a classification task. The overall training process is shown in Algorithm 2

---

**Algorithm 2** CFGAN (Xu et al., 2019)

1: **Input:** Observational data $(\mathbf{X}, Y, S)$, fairness weight $\lambda$
2: **Output:** Generated fair dataset $(\hat{\mathbf{X}}, \hat{Y}, \hat{S})$
3: Initialize two generators $(G^1, G^2)$ and two discriminators $(D^1, D^2)$.
4: **while** $G^1, G^2$ has not converged **do**
5:     Update discriminators $(D^1, D^2) \leftarrow \arg\max_{D^1, D^2} J(G^1, G^2, D^1, D^2) = J_1(G^1, D^1) + \lambda J_2(G^2, D^2)$
6:     Update generators $G^1, G^2 \leftarrow \arg\min_{G^1, G^2} J(G^1, G^2, D^1, D^2) = J_1(G^1, D^1) + \lambda J_2(G^2, D^2)$
7: **end while**
8: Train a predictor $h$ on the generated dataset $(\hat{\mathbf{X}}, \hat{Y}, \hat{S})$ and minimize the cross-entropy loss.
9: **return** $h$

---

*Why CFGAN is different from our GCFN doing counterfactual fairness*

In this section, we clarify that CFGAN considers interventions (=level 2 in Pearl's causality ladder) and not counterfactuals (=level 3). We refer Bareinboim et al. (2022) for a detailed explanation of Pearl's causality ladder.

In the CFGAN paper, the authors say that $G^2$ is generating interventional data (see CFGAN paper Section 3.2 and Figure 2). Besides, the authors give a Definition 4 of counterfactual effect, $CE(x_2, x_1 \mid \mathbf{o}) = P(Y_{x_2} \mid \mathbf{o}) - P(Y_{x_1} \mid \mathbf{o})$. Then, the authors reduce the counterfactual effect to achieve counterfactual fairness. However, in their paper, $\mathbf{o}$ is just the sensitive attribute (see the caption of Table 2 in their paper). That means. CFGAN just generates outcome $Y$ conditioned on the sensitive attribute but *not* conditioned on the post-treatment variable (i.e., factual (observation) of $Y$). This shows that CFGAN works on intervention not counterfactual. Because if you want to get the counterfactual of a variable taking another different value, you need to conditional on the factual results of the exact same variable, which CFGAN does *not* do.

However, in our method, we generate a counterfactual mediator based on the above definition of counterfactuals. Different from CFGAN, we thus operate not only on level 2 of Pearl's causality ladder but on the more complex level 3. To do so, we generate the counterfactual mediator from the distribution $P(M_{a'} \mid X = x, A = a, M = m)$, that is, we generate counterfactual $M_{a'}$ based on the factual of the mediator.

In summary, we are learning the counterfactual distribution and generating counterfactuals, whereas CFGAN is doing intervention and generating interventional samples.

## I.2 ADVAE

Note that the causal graph for ADVAE (Grari et al., 2023) is different from our setting. We show the causal graph from ADVAE in Fig. 21. Our setting allows for the existence of the confounders between the sensitive attribute $A$ and its descendant $M$, denoted as $X$ in our paper (see our causal graph in Fig. 1. This is *differnet* from ADVAE Grari et al. (2023), where such variables are *not* explicitly considered.

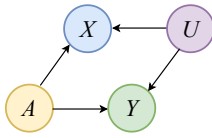

Figure 21: Causal graph in ADVAE (Grari et al., 2023).

In what follows, the notation for the sensitive attribute is consistent with our paper by using $A$ and for the target variable by using $Y$. To make the ADVAE method applicable to our setting for comparison, we regard that the descendants of the sensitive attribute (denoted by $X$ in ADVAE) correspond to $M$ in our paper. Below, we use the notation from ADVAE for ease of description.

The first step of ADVAE is counterfactual inference. The common way of using variational autoencoding (VAE) to infer the latent variable $U$ is by optimizing the lower bound (ELBO)

$$\mathcal{L}_{\text{ELBO}} = -\mathbb{E}_{(x,y,a)\sim\mathcal{D},u\sim q_\phi(u|x,y,a)}\left[\log p_\theta(x,y \mid u,a)\right] + D_{\text{KL}}\left(q_\phi(u \mid x,y,a)\|p(u)\right) \tag{73}$$

where $D_{\text{KL}}$ denotes the Kullback-Leibler divergence of the posterior $q_\phi(u \mid x,y,a)$ from a prior $p(u)$, typically a standard Gaussian distribution $\mathcal{N}(0,I)$. The posterior $q_\phi(u \mid x,y,a)$ is represented by a deep neural network with parameters $\phi$, which typically outputs the mean $\mu_\phi$ and the variance $\sigma_\phi$ of a diagonal Gaussian distribution $\mathcal{N}(\mu_\phi,\sigma_\phi I)$. The likelihood term factorizes as $p_\theta(x,y \mid u,a) = p_\theta(x \mid u,a)p_\theta(y \mid u,a)$, which are defined as neural networks with parameters $\theta$.

ADVAE employs a variant of the ELBO optimization, where the $D_{\text{KL}}\left(q_\phi(u \mid x,y,a)\|p(u)\right)$ term is replaced by an MMD term $\mathcal{L}_{\text{MMD}}\left(q_\phi(u)\|p(u)\right)$ between the aggregated posterior $q_\phi(u)$ and the prior. Their counterfactual inference is by minimizing

$$\mathcal{L} = -\mathop{\mathbb{E}}_{(x,y,a)\sim\mathcal{D},u\sim q_\phi(u|x,y,a)}\left[\begin{array}{l}\lambda_x\log\left(p_\theta(x \mid u,a)\right) + \\ \lambda_y\log\left(p_\theta(y \mid u,a)\right)\end{array}\right] + \lambda_{\text{MMD}}\mathcal{L}_{\text{MMD}}\left(q_\phi(u)\|p(u)\right)$$
$$+ \lambda_{\text{ADV}}\frac{1}{m_a}\sum_{a_k\in\Omega_A}\mathcal{L}_{\text{MMD}}\left(q_\phi(u \mid a = a_k)\|p(u)\right)$$

where $\lambda_x,\lambda_y,\lambda_{\text{MMD}},\lambda_{\text{ADV}}$ are scalar hyperparameters. ADVAE later proposes to employ an adversarial learning framework. The goal is to find some parameters $\phi$ which minimize the loss to reconstruct $X$ and $Y$, while maximizing the reconstruction loss of $A$ according to the best decoder $p_\psi(A \mid U)$, that is, $\arg\min_{\theta,\phi}\max_\psi\mathcal{L}_{\text{ADV}}(\theta,\phi,\psi)$ with

$$\mathcal{L}_{\text{ADV}}(\theta,\phi,\psi) = -\mathop{\mathbb{E}}_{(x,y,a)\sim\mathcal{D},u\sim q_\phi(u|x,y,a)}\left[\begin{array}{l}\lambda_x\log\left(p_\theta(x \mid u,a)\right) + \\ \lambda_y\log\left(p_\theta(y \mid u,a)\right)\end{array}\right]$$
$$+ \lambda_{\text{MMD}}\mathop{\mathcal{L}_{\text{MMD}}}\left(q_\phi(u)\|p(u)\right)$$
$$+ \lambda_{\text{ADV}}\mathop{\mathbb{E}}_{(x,a)\sim\mathcal{D},u\sim q_\phi(u|x,y,a)}\left[\log\left(p_\psi(a \mid u)\right)\right]$$

where $\lambda_x,\lambda_y,\lambda_{\text{MMD}},\lambda_{\text{ADV}}$ are scalar hyperparameters.

To learn a fair predictive function $h_\theta$, the second step is to minimize the loss $\mathcal{L} = \frac{1}{m}\sum_i^m l\left(h_\theta(x_i),y_i\right) + \lambda\mathcal{L}_{\mathcal{CF}}(\theta)$, where $\lambda$ is a hyperparameter that controls the impact of the counterfactual loss in the optimization, $l\left(h_\theta(x_i,a_i),y_i\right)$ is the cross-entropy loss and $\mathcal{L}_{\mathcal{CF}}(\theta)$ is the counterfactual unfairness estimation term. It is defined as

$$\mathcal{L}_{CF}(\theta) = \frac{1}{m}\sum_i^m l\left(h_\theta(x_i),y_i\right) + \lambda\mathop{\mathbb{E}}_{u\sim P(u|x_i,a_i,y_i),\tilde{x}\sim P(x|u_i,a_i),a'\sim P'(A),x'\sim P(x|u,a')}\left[\left(h_\theta(\tilde{x}) - h_\theta(x')\right)^2\right],$$

where $\Delta$ is a loss function that compares two predictions. Again, they consider a two-player adversarial game, which is formulated as $\arg\min_\theta\arg\max_\phi\mathcal{L}_{\text{DynCF}}(\theta,\phi)$ with

$$\mathcal{L}_{\text{DynCF}}(\theta,\phi) = \frac{1}{m}\sum_i^m l\left(h_\theta(x_i),y_i\right) + \lambda\mathop{\mathbb{E}}_{u\sim P(u|x_i,a_i,y_i),\tilde{x}\sim P(x|u,a_i),a'\sim P_\phi(a|u),x'\sim P(x|u,a')}\left[\left(h_\theta(\tilde{x}) - h_\theta(x')\right)^2\right].$$

This formulation considers an adversarial sampling distribution $P_\phi(A \mid U)$ rather than a uniform static distribution $P'(A)$. It takes the form of a neural network that outputs the parameters of the sampling distribution for a given individual representation $U$. The authors use a diagonal logit-normal distribution $\text{sigmoid}\left(\mathcal{N}\left(\mu_\phi(u),\sigma_\phi^2(u)I\right)\right)$, where $\mu_\phi(u)$ and $\sigma_\phi^2(u)$ stand for the mean and variance parameters provided by the network for the latent code $U$. As done for adversarial learning in the first step, all parameters are learned cjointly, by alternating steps for the adversarial maximization and steps of global loss minimization.

