# OpenReview forum: "Counterfactual Fairness for Predictions using Generative Adversarial Networks"
_ICLR.cc/2024/Conference — Submitted to ICLR 2024_

### Official Review · Reviewer_Bc4s · 2023-10-23

**Soundness:** 2 fair
**Presentation:** 3 good
**Contribution:** 2 fair
**Rating:** 5
**Confidence:** 3

**Summary:**

The paper proposes a method that learns predictor under counterfactual fairness. It first learns the counterfactual distribution of mediator (variables that are causally affected by sensitive attribute), based on which establish counterfactual mediator regularizer. Finally, the predictor is trained by enforcing such a regularizer. The paper conducts experiments to validates the proposed method on three datasets (sythentic, Adult, COMPAS).

**Strengths:**

1. Learning counterfactual fair predictors is an important problem. Unlike existing methods that learn counterfactual samples, the paper learns counterfactual distributions which are novel to my knowledge.
2. The paper is in general well-written and organized.

**Weaknesses:**

1. The paper focuses on binary-sensitive attributes. It is unclear whether the proposed method is applicable to settings where sensitive attribute has multiple categories. Specifically, for each $a$ sensitive attribute may take, do we need a GAN to learn the corresponding counterfactual distribution? Or do we only need a single GAN to learn “averaged counterfactual distribution?”
2. While the paper establishes an upper bound on the violation of fairness, the bound seems to be very trivial. The strength of fairness is controlled empirically by hyper-parameter $\lambda$.
3. The paper claims that existing works that rely on inferring latent variables may hurt prediction performance by introducing bias, and it argues the proposed method can mitigate such an issue. However, learning counterfactual distribution is often more challenging than generating counterfactual samples, and it can also be hard to stabilize the training of GAN. Although the paper empirically shows on synthetic data that the proposed method can attain a better utility-fairness trade-off than baselines, it is unclear how it performs in more complicated settings.

**Questions:**

1. Because only factual mediators are observable, how can the generator ensure the generated counterfactual mediators are accurate? To my understanding, the generator’s accuracy is ensured via construction loss between generated factual mediators and actual factual mediators. Can it imply the accuracy of counterfactual mediators?
2. How can the method be generalized to settings with more than 2 social groups?

---

> ### Author Response · Authors · 2023-11-20
> **Response to Reviewer Bc4s**
>
> We greatly appreciate your insightful feedback! We are thankful that you find our method important and novel.
>
> ### Responses to "Weaknesses"
>
>  **1. Applicability beyond binary sensitive attributes**
>
> Thank you. Our method can **easily extended to scenarios where the sensitive attribute has multiple categories**. (We only opted for the binary case for ease of readability.) To this end, we improved our paper in the following ways:
>
> * We added a **new description of how our method can be applied to a sensitive attribute with multiple categories** (see our **new Appendix G**). Importantly, we do **not** need several GANs for each category $a$. Instead, we show how a single GAN is sufficient. The reason for that is that we can learn the joint distribution for all counterfactuals.
> * We added a **mathematical proof** that a single GAN can effectively learn the joint distribution for all counterfactuals (see our **new Corollary 1**).
> * We added **new experiments** to show that our method is also effective in settings with multiple categories (see our **new Appendix G.4**).
>
>
> **2. New theoretical guarantees beyond Lemma 1**
>
> Thank you. We have greatly reworked our theoretical analysis:
> * We added a **new Lemma 2** with **theoretical guarantees** to show that our generator provides **consistent estimates** of the counterfactual distribution of mediators. This ensures that we learn the **correct** counterfactuals. Together with Lemma 1, this ensures mathematically that we achieve counterfactual fairness.
>
> * Importantly, the proof of our new Lemma 2 is **non-trivial**. We kindly refer to our **extensive proof in Supplement D.2**, where we leverage ideas from [1,2] that were published only recently at NeurIPS 2023.
>
> * The hyperparameter $\lambda$ controls the strength of counterfactual fairness. Hence, by $\lambda$->$\infty$, we ensure that the predictions adhere to counterfactual fairness more strictly. One may think that one could make the hyperparameter $\lambda$ similarly arbitrarily large to ensure counterfactual fairness. However, this would undermine the accuracy of the predictions. Rather, $\lambda$ is a hyperparameter that is used to trade-off counterfactual fairness vs. accuracy.
>
> * To the best of our knowledge, ours is the **first** neural method that offers theoretical guarantees that the method is effective in learning counterfactual fairness. In fact, many baselines (e.g., [CFGAN]) have even been shown to learn incorrect objectives that act as heuristics but fail to recover counterfactual fairness. To the best of our knowledge, our method is the first neural approach that ensures to correctly learn counterfactual fairness predictions with theoretical guarantees.
>
> References:
>
> [1] Arash Nasr-Esfahany, Mohammad Alizadeh, and Devavrat Shah. Counterfactual identifiability of bijective causal models. In ICML, 2023
>
> [2] Valentyn Melnychuk, Dennis Frauen, and Stefan Feuerriegel. Partial counterfactual identification of continuous outcomes with a curvature sensitivity model. In NeurIPS, 2023.

---

> ### Author Response · Authors · 2023-11-20
> **Response to Reviewer Bc4s**
>
> ### Responses to "Weaknesses"
>
> **3. Methods inferring latent variables may hurt prediction performance by introducing bias**
>
> Thank you for giving us the opportunity to clarify the benefits of our methods over the baselines. Below, we first explain why even the baseline methods with latent variables are fairly tricky, and we then discuss why our GAN-based approach is superior.
>
> *Why is the use of baseline methods based on latent variables tricky?*
>
> Importantly, even the baselines for counterfactual fairness are far from being easy as they do not rely on off-the-shelf methods. Rather, they also learn a latent variable in non-trivial ways. More specifically, the inferred latent variable $U$ should be independent of sensitive attribute $A$ while representing all other useful information from the observation data. However, there are two main challenges: (1) The latent variable $U$ is **not** identifiable. (2) It is very hard to learn such $U$ to satisfy the above independence requirement, especially for high-dimensional or other more complicated settings. Hence, we argue that baselines based on some custom latent variables are highly challenging.
>
> Because of (1) and (2), there are **no** theoretical guarantees for the VAE-based methods. Hence, it is mathematically **unclear** whether they actually learn the correct counterfactual fair predictions. In fact, there is even rich empirical evidence that VAE-based methods are often **suboptimal**. VAE-based methods use the estimated variable $U$ in the first step to learn the counterfactual outcome $\mathbb{P}\left(\hat{Y}_{ a'}(\mathbf{U}) \mid X=x, A=a, M=m \right).$ The inferred, non-identifiable latent variable can be correlated with the sensitive attribute which may harm fairness, or it might not fully represent the rest of the information from data and harm prediction performance.
>
> How our method overcomes the above challenges: We address the above challenges by learning the counterfactuals directly. Thus, our method eliminates the need for a two-step process that learns $U$. To this end, we avoid the complexities and potential inaccuracies of inferring and then using latent variables. More formally, we generate counterfactual samples from the learned counterfactual distribution. For each specific input data $(X=x, A=a,M=m)$, we then generate the counterfactual mediator from the distribution $\mathbb{P}(M_{a'} \mid X=x, A=a, M=m)$. This results in overall more accurate and robust predictions.
>
> More importantly, our **new Lemma 2** even provides **theoretical guarantees** that we learn the correct counterfactual mediators and, therefore, ensure counterfactual fair predictions. To the best of our knowledge, none of the baselines based on latent variables have such guarantees.
>
> As for GAN training, we followed best practices to ensure stability. For example, we employed common techniques for GAN training such as batch normalization, label smoothing, etc. On top of that, counterfactual mediators are commonly low-dimensional in practice. Recall that only a subset of all variables are outputs of our GAN (i.e., the mediators but not the covariates, not the treatment, and not the target variable). This is a crucial difference from many other applications of GANs where the outputs are high-dimensional data such as images and videos. This is one reason why we find our GAN framework to perform well throughout all of our experiments.

---

> ### Author Response · Authors · 2023-11-20
> **Response to Reviewer Bc4s**
>
> ### Responses to "Questions":
>
> **1. Correctness of counterfactual mediators**
>
> Thank you for raising this important question. You are right in pointing out that, during training, we cannot directly learn the counterfactual mediators in a supervised way as they are unobservable. Instead, we can only leverage the reconstruction loss on the factual mediators. The reason why we can still learn the correct counterfactual mediators is due to the adversarial training process of the generator. By training the discriminator to differentiate between factual and generated counterfactual mediators, the generator is guided to learn the correct counterfactual distribution.
>
> Motivated by your question, we now added a **new Lemma 2** where we **show theoretically that we learn the correct counterfactual mediators**. Specifically, our new Lemma 2 offers a theoretical guarantee that our generator consistently estimates the counterfactual distribution of the mediator. This first provides a theoretical justification behind the design of our generator for the counterfactual mediators. On top of that, Lemma 2 addresses your question by establishing convergence to the correct counterfactual distribution (point mass distribution).
>
> **2. Generalization to multiple social groups**
>
> Thank you. Our method can **easily extended to scenarios with multiple social groups.** To demonstrate this, we improved our paper in the following ways:
>
> * We added a description of **how our method can be applied to a sensitive attribute with multiple categories** (see our **new Appendix G**). Importantly, we do not need several GANs for each category $a$. Instead, we show how a single GAN is sufficient. The reason for that is that we can learn the joint distribution for all counterfactuals.
>
> * We **added mathematical proof** of how a single GAN learns the joint distribution for all counterfactuals (see our **new Corollary 1**).
>
> * We added **new experiments** to show that our method is also effective in settings with multiple categories (see our new Appendix G4).
>
> We also updated the codes in our anonymous GitHub repository accordingly.

---

### Official Review · Reviewer_RxKF · 2023-10-27

**Soundness:** 2 fair
**Presentation:** 4 excellent
**Contribution:** 3 good
**Rating:** 5
**Confidence:** 4

**Summary:**

The paper "Counterfactual Fairness for Predictions using Generative Adversarial Networks" tackles one of the most challenging definition of fairness from the fair ML community, the counterfactual fairness, which is also probably the most conceptually appealing if effective. The goal is to insure for any individual that the outcome would be the same if the individual has a different value of a sentifive attribute (e.g., gender). As many attributes of the individual can be influenced by the sensitive, it implies to build a mechanism to imagine how the individual would look like for the other - counterfactual - sensitive value. Authors of the paper propose to build on a GAN architecture, which has already been employed for that purpose, but in a different way where the approach was to generate causal and interventional data, from which counterfactual fairness could be achieved. The proposed approach is more simple, proposing to generating a counterfactual representation of individual more directly, with a specific min max objective.

**Strengths:**

Clarity : I feel the paper very well-written and structured (maybe some introductional discussion and examples  could be given to help a uneducated reader with challenges with counterfactual fairness), very easy to read from the clear definitions of every component. It is nice to read such a self-contained paper, many papers I had to review recently had a lot of ill-defined notations, I was happy to read this one.

The tackled setting is also very appealing and challenging.

The method looks to achieve interesting results.

**Weaknesses:**

I have some concerns about soundness of the approach, positioning and the experiments. see questions below.

**Questions:**

Positioning w.r.t. state of the Art

1. Related work say that VAE based methods fail because they capture correlations between the sensitive and the latent representation. Authors should at least mention that most cited approaches (e.g. Grari et al.2023) have regularizers that seek at removing these correlations. I cannot see why the proposed approach would be really better on that point, as I feel the discriminator makes some similar job as, for instance, VAE with adversarial loss that seek at reducing mutual information between the latent and the sensitive of these approaches.

2. One mentioned limitation of VAE is "It is commonly assumed that the prior distribution of latent variables follows a standard Gaussian in VAEs", that would hinder the prediction ability. While I agree that VAE is more constrained than GAN since there is a need for choosing a distribution family that strongly affects the results if is does not well fit the data, this problem is about the decoding distribution, not the encoding one, as having a latent space structured using a gaussian is something usually desirable. Note that GAN also use a  sampling from a standard gaussian in the latent space... What do you think ?

3. CFGAN: In appendix B3, authors point out than one of the main difference with CFGAN is that this approach only achieves interventional fairness, not counterfactual. From my point of view this is not true since, as explained in their section 3.5 counterfactual samples are generated by 1) generating samples from the causal model  2) selecting the z that led to the generation of a given class of the sensitive and finally 3) generate a new sample for each selected z, by interventioning on the sensitive, to get a counterfactual sample for each of them. Thus, a predictor can be learned on that data, with a regularization that ensures that both versions of individual lead to similar outputs. It is countefactual fairness as every varariables depending on the sensitive can be impacted. That is not only P(Y|X,do(A=a))=P(Y|X,do(A=a') since X and Y are regenerated after intervention (using the same z in factual and interventional world allows to follow the counterfactual fairness objective that is mentioned in the paper).

Soundness of the objectives

1. Authors consider a causal graph where X->A, but suggest in fig1 that there can exist some correlations between X and A, implying some cofounder that causes both. If there are in the data distribution for instance more old women than old men, and assuming Y is lower for women than for men, then I suspect that the regularization term in (7) cannot have any strong impact on that bias. X remains the same for factual and counterfactual, only M changes, which can be ignored by h if most of the information for outputting Y is in X. Please discuss that remark.

2. I am surprised to see that the proposed adversarial objective is greatly asymmetric, as it never considers counterfactuals produced for individuals from the A=1 class. The adversarial loss is indeed weighted by A for the factual part and 1-A for the counterfactual one (which is 0 in the case A=1). Denoting as M_{i,j} the output of the generator for an individual from class i and intervention using j as the sensitive value, we M_1,1 steered toward M from Lf and far from M from Ladv; M_0,1   steered toward M from Ladv; M_1,0   steered toward M from Lf; and M_1,0 is totally free... Isn't it a problem ? I suspect that it can report most of the fairness effort on the A=0 class. Also, I am not sure if this does correspond to a well-defined optimization problem since there may exist many equivalent optima, no ? having M_1,1 = M   looks very likely at the end of the optimization (with G  outputting a very different value for M_1,0 which is unconstrained).


Experiments

1. Comparative experiments are performed on semi-synthetic datasets, where we can get both factual and counterfactual versions of the data. However, I feel that the generative process that are considered are too easy to fully analyze, as 1) there is not correlation between X and A (which can be unfair regarding the previous remarks above) and 2) M is deterministically deduced from X and A, which appears as a really easy setting : First,  authors show in table 1 that counterfactual M can be generated accurately from their generator. However, this is not fully convincing (and does not reassure me about the asymmetry issue mentioned above), as it is easy to fully understand the relation between (X,A) and M only from samples from the 0 class (A=0). M_A' is well generated for the class A=1 thanks to this, but I am really not sure that is would applied in more difficult settings. Second, for such a setting we could design a very simpler approach that learn h(X,A) to be close to Y while limiting the distance between h(X,A) and h(X,A'), which would be at least as effective.

2. Comparison with competitors only performed on semi-synthetic- not fully convincing - datasets. I know that countefactual fairness is difficult to fully evaluate, but wouldn't it be possible to conduct some analysis on the results of some competitors also, to see if there is impact of the improvements observed on synthetic data ?

3. To be fully self-contained and reproducible, as every competitor has many variants inside their paper, with sometimes non trivial application for the considered setting, it would be nice to have in appendix the pseudo-code of each of the considered approach, at least ADVAE and CFGAN.

Minor : the "," should be replaced by a "-" in every regularization (7) or metric (9) formulations.

**Details Of Ethics Concerns:**

The paper is about removing biases.

---

> ### Author Response · Authors · 2023-11-20
> **Response to Reviewer RxKF**
>
> We appreciate the positive feedback on our paper as well as the suggestions for further improvement! As you will see below, we took your feedback at heart and have carefully and thoroughly revised our paper to incorporate your comments.
>
> In addition to comments, we have also added a **new theoretical analysis**. Specifically, we added a **new Lemma 2** to show that our method correctly identifies the counterfactual mediators. Specifically, we prove that our method allows for **consistent estimation of the counterfactual distribution**. The lemma thus gives a guarantee on the correctness of the generated counterfactual mediators and, therefore, also a guarantee on the upper bound in Lemma 1.
>
> Our new theoretical analysis is important for two reasons. First, it provides a **theoretical justification behind the design of our method**. Second, it ensures that we correctly learn counterfactual fairness. To the best of our knowledge, ours is the **first** neural method that offers such guarantees.  Note that the VAE-based baselines do **not** have similar theoretical support, which provides an additional theoretical explanation why our method is effective and even superior.
>
> ### Responses to "Positioning w.r.t. state of the Art":
>
> * Questions related to the baselines
>
> **1. Addressing concerns about the VAE-based methods**
>
> Thank you. We followed your suggestion and added the reference to VAE-based methods with regularization (e.g. Grari et al.2023) that seek to remove correlations. However, we would still clarify why VAE-based methods for our task can be problematic and why our proposed method is superior – both theoretically and numerically.
>
> *Why are VAE-based methods not optimal? *VAE-based methods learn a latent variable in non-trivial ways. More specifically, the inferred latent variable $U$ should be independent of sensitive attribute $A$ while representing all other useful information from the observation data. However, there are two main challenges: (1) The latent variable $U$ is not identifiable. (2) it is very **hard** to learn such $U$ to satisfy the above independence requirement, especially for high-dimensional or other more complicated settings. As a result, VAE-based methods act as heuristics, and, hence, there is no theoretical guarantee that VAE-based methods are actually effective. That means, there is **no theoretical guarantee** on the fairness level in the final prediction.
>
> Because of (1) and (2), there are **no** theoretical guarantees for the VAE-based methods. Hence, it is mathematically **unclear** whether they actually learn the correct counterfactual fair predictions. In fact, there is even rich empirical evidence that VAE-based methods are often **suboptimal**. VAE-based methods use the estimated variable $U$ in the first step to learn the counterfactual outcome $\mathbb{P}\left(\hat{Y}_{ a'}(\mathbf{U}) \mid X=x, A=a, M=m \right).$ The inferred, non-identifiable latent variable can be correlated with the sensitive attribute which may harm fairness, or it might not fully represent the rest of the information from data and harm prediction performance.
>
> How our method overcomes the above challenges: We address the above challenges by learning the counterfactuals directly. Thus, our method eliminates the need for a two-step process that learns $U$ but, instead, we learn $U$ directly in a single step through our GAN. To this end, we avoid the complexities and potential inaccuracies of inferring and then using latent variables. More formally, we generate counterfactual samples from the learned counterfactual distribution. For each specific input data $(X=x, A=a,M=m)$, we then generate the counterfactual mediator from the distribution $\mathbb{P}(M_{a'} \mid X=x, A=a, M=m)$. This results in overall more accurate and robust predictions. Further, our **new Lemma 2** even shows that we correctly learn the counterfactual mediators and, unlike VAE-based methods, **we thereby prove that our method is effective in achieving counterfactual fairness.**

---

> > ### Comment · Reviewer_RxKF · 2023-11-21
> > **Cannot see why U in VAE would not be identifiable, while the mediator in the proposed approach would**
> >
> > I cannot see why U in VAE would not be identifiable, while the mediator in the proposed approach would. You still need to learn a counterfactuel distribution, which looks to me at least as hard as inferring U decorrelated from A and the generate counterfactuals from this....

---

> ### Author Response · Authors · 2023-11-20
> **Response to Reviewer RxKF**
>
> ### Responses to "Positioning w.r.t. state of the Art":
>
> * Questions related to the baselines
>
>  **2. Addressing concerns about the VAE-based methods**:
>
> Thank you. Although a standard Gaussian is a common choice for the prior in VAEs, recent research, such as [2,3], has highlighted that this choice can be **suboptimal**. The standard Gaussian prior can lead to over-regularization, contributing to poorer density estimation performance. This issue is often referred to as the posterior-collapse phenomenon, as discussed by [4].
>
> Regarding GANs, it is true that they often utilize sampling from a standard Gaussian in the latent space. However, this is not a strict requirement. Various studies have demonstrated effective GAN models that operate without sampling from a Gaussian distribution [5]. Our method diverges from the standard practice as well; in our GAN model, we do not rely on sampling from a standard Gaussian. Instead, we focus on generating counterfactuals based on the learned counterfactual distribution. Thus, our method is more robust by directly learning the transformation.
>
> Motivated by your question, we **added a new Lemma 2**, which provides **theoretical justification for our method**. Therein, we prove that our method allows for **consistent estimation of the counterfactual distribution**. This is important for two reasons. First, it provides a theoretical justification behind the design of our GAN method. Second, it ensures that we correctly learn counterfactual fairness. To the best of our knowledge, ours is the **first** neural method for counterfactual fairness that offers such guarantees. Importantly, this is a major difference to the VAE-based baselines, which do **not** have similar guarantees.
>
>
> References:
>
> [1] Depeng Xu, Yongkai Wu, Shuhan Yuan, Lu Zhang, and Xintao Wu. Achieving causal fairness through generative adversarial networks. In IJCAI, 2019.
>
> [2] Takahashi, Hiroshi and Iwata, Tomoharu and Yamanaka, Yuki and Yamada, Masanori and Yagi, Satoshi. "Variational autoencoder with implicit optimal priors." In AAAI, 2019.
>
> [3] Hoffman, Matthew D., and Johnson, Matthew J. ELBO surgery: yet another way to carve up the variational evidence lower bound. In Workshop in Advances in Approximate Bayesian Inference,  NeurIPS. 2016.
>
> [4] Aaron van den Oord, Oriol Vinyals, koray kavukcuoglu. Neural Discrete Representation Learning. In  In NeurIPS.2017
>
> [5] Ledig C, Theis L, Huszár F, et al. Photo-realistic single image super-resolution using a generative adversarial network. Proceedings of the IEEE conference on computer vision and pattern recognition. 2017

---

> ### Author Response · Authors · 2023-11-20
> **Response to Reviewer RxKF**
>
> ### Responses to "Positioning w.r.t. state of the Art":
>
>
> **3. Addressing concerns about the CFGAN [1]**:
>
> Thank you. In response to the query regarding CFGAN and its relation to interventional versus counterfactual fairness, we would like to direct the reviewer's attention to the paper [6]. Therein, the authors show that CFGAN actually satisfies the definition of Discrimination avoiding through causal reasoning. This means: a generator is said to be fair if the following equation holds  $P(Y=y \mid X=x, d o(A=a))=P\left(Y=y \mid X=x, d o\left(A=a^{\prime}\right)\right)$. This is **different from counterfactual fairness**. In counterfactual fairness, a generator producing samples $(X, A, Y)$ with distribution $P$ is said to be counterfactually fair if: $P\left(Y_{a}=y \mid X=x, A=a\right)=P\left(Y_{a’ }=y \mid X=x, A=a\right),$ for all $y \in \mathcal{Y}, x \in \mathcal{X}, a, a’ \in \mathcal{A}$. As such, CFGAN actually considers **interventional queries** (=**level 2** in Pearl’s causality ladder), while counterfactual fairness involves **counterfactual queries** (=**level 3** in Pearl’s causality ladder).
>
> For details of the above, we kindly refer to paper [6], which provides a comprehensive explanation of the actual distribution CFGAN learns and the distinction between counterfactual fairness, along with detailed proofs.
>
> Reassuringly, we remind that CFGAN is **vastly different from our method** (see our discussion in **Appendix B.2**). (1) Different tasks:  CFGAN is designed for fair data generation tasks, while our model is designed for learning predictors to be counterfactual fairness. Hence, both address **different tasks**. (2) Different architectures: CFGAN employs two generators, while our method has a single generator and discriminator. Further, fairness enters both architectures at **different** places. In CFGAN, fairness is ensured through the GAN setup, whereas our method ensures fairness in a second step through our counterfactual mediator regularization. (3) Different training objectives: The training objectives are **different**: CFGAN learns to mimic factual data. In our method, the generator learns the counterfactual distribution of the mediator through the discriminator distinguishing factual from counterfactual mediators. (4) No theoretical guarantee for CFGAN: Unlike our method, CFGAN does not have theoretical guarantees to correctly learn counterfactual fair predictions. **Only our method** offers theoretical guarantees for the task at hand. In sum, even though CFGAN also employs GANs, it is vastly different from our method.
>
>
> Reference:
>
> [6] Mahed Abroshan, Mohammad Mahdi Khalili, and Andrew Elliott. Counterfactual fairness in synthetic data generation. In Neurips 2022 SyntheticData4ML Research, 2022.

---

> ### Author Response · Authors · 2023-11-20
> **Response to Reviewer RxKF**
>
> ### Responses “Soundness of the objectives”
>
> **1.Addressing concern about the spurious effect**
>
> Thank you for asking questions about the potential spurious effects in the correlation between $X$ and $A$, which can have an impact on $Y$ We welcome the opportunity to provide a clarification. It is true that if $X$ and $A$ are correlated, there would occur a spurious effect ($A$->$X$->$Y$) in our causal graph. However, note that the counterfactual fairness notion [8] only addresses direct effect ($A$->$Y$) and indirect effect ($A$->$M$->$Y$), but does not include the spurious effect ($A$->$X$->$Y$). This is because the spurious effect ($A$->$X$->$Y$) does not influence the counterfactual fairness in the prediction.
>
> We kindly refer to paper [9], which provides an in-depth and detailed discussion of the difference between various fairness notions and causal effects. Figure 4.5 in paper [9] explains clearly that counterfactual fairness is the unit-level fairness notion and, thus, does not include spurious effects. This makes counterfactual fairness different from other fairness notions (such as causal fairness).
> To illustrate the above, let us draw upon your example. In the given scenario, $X$ represents age, and $A$ represents gender. We further assume that there is correlation between them ( typically, women are older than men). A company aiming for gender fairness, thus chooses not to use gender directly in its hiring decisions. Since age is not a sensitive attribute, they use age for making decisions, preferring younger candidates. This method, although seemingly unbiased w.r.t gender, inadvertently leads to a higher number of male employees than female employees due to the correlation between age and gender. This example points to the reviewer's concern: such correlations do indeed influence outcomes at the population level, creating an average effect.
>
> However, from the perspective of counterfactual fairness, the focus is on the individual level. For each person, it examines the situation where altering an individual's gender (e.g., from male to female) wouldn't change the employment outcome of this person. Thus, even though a correlation between age and gender exists, it does not affect each individual's final employment decision.
> Therefore, having correlations between $X$ and $A$ does not affect the counterfactual level in our prediction in Y, which is consistent with our setting.
>
>
> **2.Addressing concerns about the adversarial loss**
>
> Thank you. We would like to clarify that our adversarial objective is indeed **symmetric**, addressing counterfactual scenarios for both $A=1$ and $A=0$ classes. There is no preference for the class and the loss formula is correct. Since the sum of the discriminator output is 1 (i.e., a sum of the probability), we merge the two items in the original version of the loss equation into one (see our updated Eq. 6 for the loss). If you view $A$ as the true label, it is exactly how cross-entropy loss is computed. The counterfactual output of the generator is always fed into the discriminator. The generator of the GAN then minimizes the conditional propensity-weighted Jensen–Shannon divergence (JSD).
> To show the above more explicitly, we also added a new Lemma 2 and a corresponding proof. We kindly refer the reviewer to our proof, which should make the above more clear.  (see Eq. (31) in Appendix D.2).
>
> References:
>
> [9] Drago Plecko and Elias Bareinboim. Causal fairness analysis. In arXiv preprint, 2022.

---

> > ### Comment · Reviewer_RxKF · 2023-11-21
> > **1.Addressing concern about the spurious effect**
> >
> > ok my example was maybe bad.
> > Thanks for this answer

---

> > ### Comment · Reviewer_RxKF · 2023-11-21
> > **2.Addressing concerns about the adversarial loss**
> >
> > Thanks, but from my point of view the current presentation (which is very different from the former one) is even worse: nothing considers counterfactuals in that new formulation  of (6), I suspect there is a mistake like a A in place of a A' ?

---

> ### Author Response · Authors · 2023-11-20
> **Response to Reviewer RxKF**
>
> ### Responses to “experiments”
>
> Thank you. We followed your suggestions and included several new experiments (see **new Appendix X.1 and X.2**).
>
> **1**.
> *(a). Correlation Between $X$ and $A$:*
>
> We added a new experiment where we included correlation between $X$ and $A$. The experiment results show again that we achieved the best performance (see our **new Appendix H.1**). Importantly, these results confirm that introducing a correlation between $X$ and $A$ does not adversely affect the counterfactual level of our predictions for $Y$. This finding aligns with our above response to the initial concern on spurious effects, showing that having correlation between X and A does not affect the counterfactual fairness level in our prediction in Y.
>
> *(b). Generation of M from X and A*:
>
> Thank you for asking the questions. We would like to clarify that $M$ is **not** deterministically deduced from $X$ and $A$. In our dataset, the generation of $M$ incorporates stochastic elements, specifically through the inclusion of noise $U_M$, which makes the process non-deterministic. We have thus revised our paper to spell out explicitly that the generation of $M$ is non-deterministically.
> In response to your query regarding the evaluation of counterfactual fairness: This limitation is widely recognized in the field, and as a result, the use of synthetic or semi-synthetic datasets for such evaluations is a well-established norm, as is the quasi-standard in the literature [7,8,10].
>
> Regarding the simplicity of the setting: our setting is intentionally designed **analogous to prior research** [8, 10] and especially [7]. Therein, the setting for modeling fairness violations is similar to ours, so that we allow for better comparability.
> Lastly, it is crucial to note that our prediction model **does** not explicitly utilize the sensitive attribute. The inputs for our predictor $h$ are $X$ and $M$. This is done to adhere to ethical guidelines in practice and to reduce the risk of biased outcomes. We have revised our paper to spell out clearly that **we follow your comment and do not use the sensitive attribute in our predictor**.
>
> **2**.  We followed your suggestion and **added new performance comparisons** for our real-world dataset (see our **new Appendix H.2**). Due to the unavailability of counterfactuals for real-world data, we use our generated counterfactual as the ground-truth counterfactual and show the utility function value as on synthetic datasets. As a result, we find that our proposed method is highly effective.
>
> **3**. Thank you. We paid great attention to make our work self-contained and reproducible. Wherever possible, we used the original source codes from the authors for fair comparison (and we contacted all authors whenever the code was not public in the first place). To address your comment, we have improved our papers as follows:
> We extended our implementation details and now state clearly where the baseline codes are from (see our revised **Appendix F.1**).
> We have also added detailed descriptions including pseudocode for ADVAE and CFGAN (see our new materials in **Appendix I**).
> We have made both our code and all datasets publicly available (see the GitHub link in our paper). Thereby, we ensure that all experiments reported in our study are indeed self-contained and reproducible.
>
> Reference:
>
> [7] Ioana Bica, James Jordon, and Mihaela van der Schaar. Estimating the effects of continuous-valued interventions using generative adversarial networks. In NeurIPS, 2020.
>
> [8] Matt J Kusner, Joshua Loftus, Chris Russell, and Ricardo Silva. Counterfactual fairness. In NeurIPS, 2017.
>
> [10] Francesco Quinzan, Cecilia Casolo, Krikamol Muandet, Niki Kilbertus, and Yucen Luo. Learning counterfactually invariant predictors. In arXiv preprint, 2022.

---

> > ### Comment · Reviewer_RxKF · 2023-11-21
> > **thanks for new experiments**
> >
> > thanks for new experiments that look very useful to me.
> >
> > I slightly increased my score consequently, but still have some concerns with the paper (see my other comments).

---

> ### Comment · Reviewer_RxKF · 2023-11-21
> **disagree on cfgan interpretation**
>
> Thanks for this reference but I stil think that CFGAN proposes a way to achieve counterfactual fairness, not only interventional one, as they produce both versions of the individuals, with modified Y and X according to the sensitive attribute (using same Z as seed of the generation). From this it is possible to learn a predictor that achieves comparable outcomes for both versions...

---

> > ### Author Response · Authors · 2023-11-21
> > **Addressing concern about CFGAN interpretation 1/2 (Response to Reviewer RxKF)**
> >
> > ### Addressing concern about "CFGAN interpretation": (1/2)
> >
> > Thank you very much! We appreciate your help in improving the paper and your constructive feedback!
> >
> > We would kindly **clarify that “intervention” and “counterfactual” are different concepts with specific meanings in the context of Pearl’s causal hierarchy** [1]. We have the feeling that the reviewer may have a background in fairness and not in causal machine learning, because of which the questions seem to mix up both.
> >
> > * **Interventional** queries are located on level 2 of Pearl’s causality ladder. Interventional queries are of the form $P(y \mid do(x)$. Here, the typical question is “What if? What if I do X?”, where the activity is “doing”.
> >
> > * **Counterfactual** queries are located on level 3 of Pearl’s causality ladder. Counterfactual queries are of the form $ P(y_x \mid x’, y’)$, where $x’$ and $y’$ are different values that $X, Y$ took beforehand. Here, the typical question is” What if I had acted differently?”, where the activity is “imagining” had a different treatment selected been made in the beginning. Hence, the main difference is that the counterfactual of $y$ is conditioned on the post-treatment outcome (factual outcome) of $y$ and a different $x$ (where $x$ takes a different value than $x’$).
> >
> > For illustration, let us consider an example from credit lending to show the differences between interventional vs. counterfactual queries:
> >
> > -Interventional query. Here, we are interested in finding the effect of gender on default. A common question would be: “What would be the risk of default if the person increases now her salary from $10 to $20.” Here, the approach is that we actively change the variable (salary) and observe the outcome in terms of default risk.
> >
> >
> > -Counterfactual query. Here, we are interested in hypothetical backward reasoning. A common question would be: "What would have been the load approval decision if the person would have had a higher salary?” Here, the approach requires imagining a scenario that didn’t happen but could have, given the circumstances. It involves considering the person's credit risk under a hypothetical situation where they had a different salary.
> >
> >
> > This example illustrates how interventional queries are about the outcomes of real changes we can make, while counterfactual queries explore hypothetical scenarios and their possible outcomes.  In summary, interventional queries are concerned with the effects of deliberately changing a variable. In contrast, counterfactual queries deal with hypothetical alterations of past events and their potential outcomes. As you can also see in the above example, counterfactual fairness is a counterfactual query (and not an interventional query) because we want to ensure similar outcomes in a hypothetical example where we imagine that certain personal characteristics are different.
> >
> > We understand that the naming of the two concepts in the existing causality literature may not be directly intuitive. We thus **added a clarification that interventions and counterfactuals are not the same**. Furthermore, we kindly refer to papers [1], [2], [3] for the detailed definition of why intervention and counterfactual are two **entirely different concepts**. In particular, we personally find reference [1] (“On Pearl’s Hierarchy and the Foundations of Causal Inference”) especially helpful, where the author gives a very clear explanation of Pearl’s Hierarchy.
> >
> > Reference:
> >
> > [1] Bareinboim E, Correa J D, Ibeling D, et al. On Pearl’s hierarchy and the foundations of causal inference. Probabilistic and causal inference: the works of Judea Pearl. 2022.
> >
> > [2] Pearl, Judea. Causal inference in statistics: An overview. 2009.
> >
> > [3]Peters, Jonas, Dominik Janzing, and Bernhard Schölkopf. Elements of causal inference: foundations and learning algorithms. The MIT Press, 2017.

---

> > ### Author Response · Authors · 2023-11-21
> > **Addressing concern about CFGAN interpretation 2/2 (Response to Reviewer RxKF)**
> >
> > ### Addressing concern about CFGAN interpretation (2/2)
> >
> > In the following, we now build upon the above concepts that interventions vs. counterfactuals are different concepts. We then continue by discussing that CFGAN [4] actually involves **interventions** (=level 2 in Pearl’s causality ladder) and **not counterfactuals** (=level 3). Again, we apologize for the common nomenclature in the causality literature, which is unfortunately misleading: CFGAN includes the name ‘counterfactual’ but, in terms of the underlying query, **it is not counterfactual but only interventional**.
> >
> > Let us explain the above difference more formally. It is true that CFGAN generates different $Y$ and $X$ of sensitive attribute takes 1 and 0, and the two generators $G^1$ and generator $G^2$ use the same noise seed $z$. However, it does **not** mean CFGAN is generating counterfactual outcomes.
> >
> >
> > We acknowledge that, in many works, the terms counterfactual and potential outcomes are used interchangeably. However, from the point of Pear's SCM approach to causal inference, the "counterfactual" effect from the CFGAN paper is an interventional effect and lies on layer 2 of Pearl's hierarchy. This is because
> >
> > (1) In the CFGAN paper [4], the authors say that $G^2$ is generating interventional data (see CFGAN paper Section 3.2 and Figure 2).
> >
> > (2) In the CFGAN paper [4], the author give their definition of counterfactual effect (see CFGAN paper Definition 4) as $CE\left(x_2, x_1 \mid \mathbf{o}\right)=P\left(Y_{x_2} \mid \mathbf{o}\right)-P\left(Y_{x_1} \mid \mathbf{o}\right)$. Then, the authors reduce the counterfactual effect to achieve counterfactual fairness. However, in their paper,  $\mathbf{o}$ is just the sensitive attribute (see the caption of Table 2 in their paper).
> >
> > That means: CFGAN just generates outcome $Y$ only conditioned on the sensitive attribute, but not conditioned on the post-treatment variable ( here is the factual (observation) of $Y$). This shows that **CFGAN is interventional but not counterfactual** (according to Pearl’s causal hierarchy).  Because if you want to get the counterfactual of a variable taking another different value, you need to conditional on the factual results of the exact same variable, which CFGAN does not do.
> >
> > However, in our method, **we generate a counterfactual mediator based on the above definition of counterfactuals**. Different from CFGAN, we thus operate not only on level 2 of Pearl’s causality ladder **but on the more complex level 3**. To do so, we generate the counterfactual mediator from the distribution $P(M_{a'} \mid X=x, A=a, M=m)$, that is, we generate counterfactual $M_{a'}$ based on the factual of the mediator.
> >
> > In summary, we are learning counterfactual distribution and generating counterfactuals while CFGAN is doing intervention and generating interventional samples.
> >
> > **Action (in blue font):** We added a clarification to our paper that interventional and counterfactual queries are different. We then build upon the differences in the terminology and explain why CFGAN is focused on not solving our task (see our revised **Appendix B.1 and I.1**).
> >
> >
> > Reference:
> >
> >
> > [4]Depeng Xu, Yongkai Wu, Shuhan Yuan, Lu Zhang, and Xintao Wu. Achieving causal fairness through generative adversarial networks. In IJCAI, 2019.

---

> > > ### Comment · Reviewer_RxKF · 2023-11-22
> > > **Addressing concern about CFGAN interpretation 2/2**
> > >
> > > Many thanks to the authors for their detailed answers. I acknowledge that maybe I miss something, but I still think that CFGAN could be applied for counterfactual fairness, although I agree that in their paper authors use it differently, by considering groups of individuals defined according to a subset of variables o, that remain unchanged in the factual and counterfactual world (o is not necessarily the sensitive contrary to what authors indicate, at least I did not understand it like this).
> > >
> > > I previously spent time to capture how we could leverage CFGAN for counterfactual fairness from this, and my understanding was that if you consider the generation seed z as o, you can generate all variables for the same individual with a=1 and a=0, and then regulate to achieve counterfactual fairness. z is the essence of the individual, which controls everything given the sensitive A. So, while it is true that CFGAN only reports results that compare outcomes for groups of individual that have similar properties o in both world, reducing o to the generation seed would comes down to consider CFGAN for counterfactual fairness no ? From my point of view, this is like this that it should be considered in experiments to get a comparable setting as yours.

---

> > > > ### Author Response · Authors · 2023-11-22
> > > > **Further discussion on CFGAN (Response to Reviewer RxKF)**
> > > >
> > > > ### Further discussion on CFGAN
> > > >
> > > > Thank you. CFGAN [1]  is a relevant baseline for our task. Originally, CFGAN  [1] was proposed as a method for synthetic data generation under counterfactual fairness. We thus included CFGAN as a baseline in our paper but adapted to our task using ideas from [3]. That is, we first use CFGAN to generate fair synthetic data and then train a machine learning model on it. (This is typically referred to as pre-processing in the algorithmic fairness literature [3], while our method uses in-processing).
> > > >
> > > > **We report the results for the above CFGAN approach in our main paper and in all extended analyses (see Figure 3 and Figure 11)**. We find that our proposed method outperforms CFGAN by a large main. (We refer to our previous answers that highlight the theoretical reasons behind that.)
> > > >
> > > > We additionally followed the idea from the reviewer to modify CFGAN by considering the $z$ as $\mathbf{o}$ in the generation process. In the following, we now refer to this second, modified version of CFGAN as “modified CFGAN”. We ran the experiment on our semi-synthetic dataset. The accuracy and fairness of the modified CFGAN (as proposed by the reviewer) and our method GCFN are shown below:
> > > >
> > > > | Methods         | Utility ($\gamma=1$) | Counterfactual Fairness |
> > > > |-----------------|-----------------------|-------------------------|
> > > > | Original CFGAN  | 0.6285                | 0.3137                  |
> > > > | Modified CFGAN  | 0.4086                | 0.3679                  |
> > > > | Our GCFN        | 0.7941                | 0.1955                  |
> > > >
> > > >
> > > > (Notes: Utility => higher the better; counterfactual fairness => lower the better)
> > > >
> > > > As you can see from the above results, the modified version of CFGAN is slightly inferior to the two-step approach [3] for CFGAN. We thus report the baseline result from the latter in our paper. We believe that this is beneficial as it is more consistent with the original approach in [1] and thus allows for a more fair comparison.
> > > >
> > > > Finally, we would like to emphasize that the above results are not surprising: CFGAN was originally designed for fair synthetic dataset generation, while our method is carefully designed for a different task, namely, predictions under counterfactual fairness. Importantly, **only our method has a theoretical guarantee** that ensures that we correctly learn predictions under counterfactual fairness.
> > > >
> > > >
> > > >
> > > > Reference:
> > > >
> > > > [1] Depeng Xu, Yongkai Wu, Shuhan Yuan, Lu Zhang, and Xintao Wu. Achieving causal fairness through generative adversarial networks. In IJCAI, 2019.
> > > >
> > > > [2] Hyemi Kim, Seungjae Shin, JoonHo Jang, Kyungwoo Song, Weonyoung Joo, Wanmo Kang, and Il.Chul Moon. Counterfactual fairness with disentangled causal effect variational autoencoder. In AAAI, 2021.
> > > >
> > > > [3]Plecko, Drago, and Elias Bareinboim. Causal fairness analysis. arXiv 2022.

---

> > > > > ### Comment · Reviewer_RxKF · 2023-11-23
> > > > > **Further discussion on CFGAN**
> > > > >
> > > > > Thanks for considering the proposal. But I still have some concerns :
> > > > > - authors set cfgan in the pre-processing category of fairness approaches. From my point of view, this is not fully true as they generate data that are not fair but which can be leveraged in counterfactual or interventional fairness by getting both versions of each individual z or group o. The bias mitigation is performed afterwards, on that dataset, as any in-processing approach
> > > > > - I cannot understand how the original CFGAN is used by authors. you use the sensitive as o that's it ? you generate many samples from the generator with both versions of the sensitive and then mitigate biases between s=0 and s=1 ? indeed, that is group fairness, not individual, so I don't think this baseline is really relevant
> > > > > - Thanks having considered (very quickly !) my proposal of using z as o. But I am surprised of such so bad results. You generate a dataset with both versions of each individual using same z for s=0 and s=1. Then you train a predictor that considers a regularization constraint that minimizes $|P(y_0|s_0, x_0)-P(y_1|s_1, x_1)|^2$ ? If yes, what could explain as bad results, I do not feel this process very different from what authors propose, except the use of a mediator that is the only changing part while cfgan can change everything acting on s...

---

> > > > > > ### Author Response · Authors · 2023-11-23
> > > > > >
> > > > > > Thank you. To summarize, CFGAN can essentially predict two variants $y_0$ and $y_1$ for individual from different social groups $s_0$ and $s_1$. Hence, as you nicely summarize, the question is now how should we learn predictions that are fair. Note that the CFGAN does **not** have a prediction model. Hence, we could picture two remedies to adapt CFGAN to our task. One is to use preprocessing-processing where train an additional classifier on top of the data as (in suggested by Barenboim et al) and which we used. The other is to develop an in-processing approach along your idea by directly incorporating the fairness constraint in the GAN. Hence, this would not allow for training arbitrary prediction model but would our model to coincide with a GAN to make predictions. Given the complexity and little flexibility of GANs, this would lead to a **low** performance of GANs in our prediction task.

---

> ### Author Response · Authors · 2023-11-21
> **Addressing concern on "identifiable for VAE-based method” (Response to Reviewer RxKF)**
>
> ### Addressing concern on “identifiable for VAE-based method”:
>
> We break down the reviewer's questions and answer it point by point below:
>
> **1. What does identification mean**:
>
> First, “identification” does **not** mean “estimation“. In causal inference, you can think about identifiability as the mathematical condition that permits a causal quantity to be measured from observed data. We kindly refer to papers [2][3] which all discuss clearly what identification means in causality. Importantly, identification is different from estimaton because methods that act as heuristics may return estimates but they do not correspond to the true value. (see paper [1] where the authors provide several concerns that, if it is not unique, it is possible to have local minima, which leads to unsafe results.
>
> **2. Non-identifiable for VAE based methods have been shown in prior works of literature**:
>
> We kindly refer the reviewer to this paper [7] [Neural Causal Models for Counterfactual Identification and Estimation] ” In this paper, the authors show that VAE-based counterfactual inference do not allow for identifiability. The results directly apply to variational inference-based methods they listed (VACA Sanchez-Martin et al. (2021), DeepSCM Pawlowski et al. (2020)), which do not have proper identification guarantees. Also, the result from non-linear ICA (which is the task of variational autoencoders) shows that the latent variables are non-identifiable [6]. In simple words, VAE-based methods **can estimate the latent variable but it is not guaranteed that it can be correctly identified**, leading to risks that the latent variable is often estimated incorrectly.
>
>  **3. Non-identifiability of the latent variables means non-identifiability of the counterfactual queries:**
>
> We kindly refer to paper [4] (Partial Counterfactual Identification of Continuous Outcomes with a Curvature Sensitivity Model. ) which says that the **non-identifiability of the latent variables means non-identifiability of the counterfactual queries**. Hence, VAE-based methods can **not** ensure that they correctly learn counterfactual fairness, **only our method does so**.
>
>  **4. Regarding our method**
>
> We diverge from the above practices. Instead, we focus on generating counterfactuals based on the learned counterfactual distribution. Thus, our method is more robust by directly learning the transformation. More importantly, we provide theoretical guarantees in our paper that we correctly learn counterfactual fairness (see **Lemma 1** and **Lemma2**)
>
> * How our method overcomes the above challenges:
>
> We address the above challenges by learning the counterfactuals directly. Thus, our method eliminates the need for a two-step process that learns $U$. To this end, we avoid the complexities and potential inaccuracies of inferring and then using latent variables. More formally, we generate counterfactual samples from the learned counterfactual distribution. For each specific input data $(X=x, A=a,M=m)$, we then generate the counterfactual mediator from the distribution $\mathbb{P}(M_{a'} \mid X=x, A=a, M=m)$. This results in overall more accurate and robust predictions. Further, our **new Lemma 2** even shows that we correctly learn the counterfactual mediators and, unlike VAE-based methods, **we thereby prove that our method is effective in achieving counterfactual fairness.**
>
> **Action (in blue font):** We added a clarification to our paper to distinguish identifiability vs. estimatability (see our revised **Section 2** and **Appendix B.1**). We further spell out clearly that one weakness of the VAE-based methods is that they are non-identifiability, while our method allows for identifiability.
>
>
> We are again thankful for your question. We hope that our answer helped in addressing your questions satisfactorily. Please let us know if there are further things in our presentation that should be improved.
>
> Reference:
>
> [1] D’Amour A. On Multi-Cause Causal Inference with Unobserved Confounding: Counterexamples[J]. Impossibility, and Alternatives, 2019.
>
> [2] Pearl, Judea. Causal inference in statistics: An overview. 2009
>
> [3]Peters, Jonas, Dominik Janzing, and Bernhard Schölkopf. Elements of causal inference: foundations and learning algorithms. The MIT Press, 2017.
>
> [4]Melnychuk, Valentyn, Dennis Frauen, and Stefan Feuerriegel. Partial Counterfactual Identification of Continuous Outcomes with a Curvature Sensitivity Model.  Neurips 2023.
>
> [5]De Brouwer, Edward. "Deep Counterfactual Estimation with Categorical Background Variables." Neurips. 2022.
>
> [6]Khemakhem I, Kingma D, Monti R, et al. Variational autoencoders and nonlinear ica: A unifying framework. International Conference on Artificial Intelligence and Statistics. 2020.
>
> [7]Xia, Kevin, Yushu Pan, and Elias Bareinboim. "Neural causal models for counterfactual identification and estimation." arXiv 2022.

---

> ### Author Response · Authors · 2023-11-22
> **Addressing concerns about the adversarial loss (Response to Reviewer RxKF)**
>
> ### Addressing concerns about the adversarial loss
>
> Thank you for your question. We would like to clarify that the updated version of adversarial loss in Eq.6 is **exactly the same as* the original version (but with better notation). We would also like to explain below why our formalization of the loss is correct.
>
>
> In the original version, our adversarial loss is written as ${L}\_{adv}=\mathbb{E}\_{({X,A,M)\sim \mathbb{P}\_\mathrm{f}}}\left[A \log\big( {D}(X, \tilde{G}\left(X,A,M \right))\_{A}\big)+A^{\prime} \log \big( 1-{D}(X, \tilde{G}\left(X,A,M \right))_{A'}\big)\right]$.
>
> Our current loss is ${L}\_{adv} = \mathbb{E}\_{({X,A,M)\sim \mathbb{P}\_\mathrm{f}}}\left[\log\big( {D}(X, \tilde{G}\left(X,A,M \right))_{A}\big) \right ]$.
>
> It might not look that straightforward to see the updated version is exactly the same (just merge the original two terms into one),  but it is actually **the same as the previous one**. Let us clarify that step by step.
>
>
> In binary setting, $1-{D}(X, \tilde{G}\left(X,A,M \right))_{A'}$ is actually equal to ${D}(X, \tilde{G}\left(X,A,M \right))\_{A}$  . This because the sum of ${D}(X, \tilde{G}\left(X,A,M \right))\_{A’}$ and ${D}(X, \tilde{G}\left(X,A,M \right))\_{A}$ should be 1 (i.e., he discriminator $D$ outputs the probability and the sum of the total probability is 1).
>
> Let’s consider $A=1$ and $A=0$ respectively, for the original version of loss.
>
> - When $A=1$, $A’ = 0$ makes the second term vanish, then the only thing left is
> $\log({D}(X, \tilde{G}\left(X,A,M \right))\_{A})$, which is $\log({D}(X, \tilde{G}\left(X,1,M \right))\_{1})$.
>
> - When $A=0$, then the first term turns to  0 and the second term is $ \log(1-{D}(X, \tilde{G}\left(X,A,M \right))_{A'})$, which is equal to $\log({D}(X, \tilde{G}\left(X,A,M \right))\_{A})$ , and the loss now is $\log({D}(X, \tilde{G}\left(X,0,M \right))\_{0})$ .
>
>
> Our current loss is $\mathbb{E}\_{({X,A,M)\sim \mathbb{P}\_\mathrm{f}}}\left[\log\big( {D}(X, \tilde{G}\left(X,A,M \right))_{A}\big) \right ]$.  Let’s consider $A=1$ and $A=0$ respectively, for the current version of loss.
>
>
> - When$A=1$, the loss is $\log({D}(X, \tilde{G}\left(X,1,M \right))\_{1})$
>
> - When $A=0$, the loss is $\log({D}(X, \tilde{G}\left(X,0,M \right))\_{0})$.
>
> So our updated version is **exactly the same as** the original version.
>
>
> We want to clarify it considers the counterfactual in our formula. **The output of the generator $\tilde{G}\left(X,A,M \right)$ is always fed into the discriminator**. Besides, we consider both classes for $A=1$ and $A=0$.  The generator of the GAN actually minimizes the conditional propensity-weighted Jensen–Shannon divergence (JSD). To show the above more explicitly, we kindly refer the reviewer to our proof, which should make the above more clear.  (see **Eq. (31) in Appendix D.2**).
>
>
> For ease of understanding, one can simply view $A$ as the true label in a classification class, and our loss is no different from the normal **cross-entropy loss** for the classification task.
>
> **Action (in blue font):** We added a brief intuition along the above lines to explain our loss in greater detail.
>
> We are again thankful for your question. We hope that our answer helped in addressing your questions satisfactorily. Please let us know if there are further things in our presentation that should be improved.

---

> > ### Comment · Reviewer_RxKF · 2023-11-22
> > **Addressing concerns about the adversarial loss**
> >
> > Sorry but, if I see the equivalence you detail, I still feel there remains something missing in your formulation. In particular, I cannot see what prevents G to output the trivial solution M for both cases, which prevents D to well distinguish, but does not produce any counterfactual (M never modified)...  please clarify

---

> > > ### Author Response · Authors · 2023-11-22
> > > **Further discussion on loss (Response to Reviewer RxKF)**
> > >
> > > ### Further discussion on adversarial loss
> > >
> > > Thank you for your questions. We see the reviewer’s question of whether the generator $G$ would output a trivial solution and whether it would be super hard for the discriminator $D$ to distinguish. Specifically, the reviewer may think that our method simply learns to reproduce factual mediators in the GAN rather than actually learning the counterfactual mediators. **However, this is not the case**.
> > >
> > > There are four important reasons why this is not the case:
> > >
> > > **Intuitively**: The input of the discriminator $D$ contains the counterfactual and the factual and the order of them is intentionally randomized. Suppose hypothetically that the factual always comes in the first place, then it is easy to distinguish. However, in the design of our framework, this is not the case. In our framework, the data are intentionally shuffled so that factual and counterfactual positions are random.
> > >
> > > **Technical reason**:  If $G$ would just copy the factual of the $M$ (which is the input of the generator $G$), it would be super hard for the discriminator to distinguish, and the loss of $D$ would be super large. We would observe model collapse during training, which we did not.
> > >
> > > **Theoretical reason**: We provide theoretical proof in our Lemma 2. Therein we show: our generator consistently estimates the counterfactual distribution of the mediator $\mathbb{P}(M_{a'} \mid X=x, A=a, M=m)$. For each specific input data $(X=x, A=a,M=m)$, we then generate the counterfactual mediator from the distribution $\mathbb{P}(M_{a'} \mid X=x, A=a, M=m)$. Hence, we offer a theoretical proof that we learn counterfactual fairness correctly.
> > >
> > >
> > >
> > > **Experimental reasons:** We conducted experiments to show your above arguments (see our Table 1 in Sec. 5.2). We find: (1) The factual mediator and the generated counterfactual mediator are highly dissimilar. This is shown by a normalized MSE($M,\hat{M}\_{A'})$ of $\approx1$. (2) The ground-truth counterfactual mediator and our generated counterfactual mediator are highly similar. This shown by a normalized MSE($M\_{A'},\hat{M}\_{A'})$ of close to zero.
> > >
> > > As you see from above, our method does not just copy factual data. On the contrary, it does **not output trivial solutions** but it is **effective in learning counterfactual mediators.**

---

> > > > ### Comment · Reviewer_RxKF · 2023-11-23
> > > > **Further discussion on adversarial loss**
> > > >
> > > > ok in practice you do not observe such convergence toward trivial solutions of G, thanks to an alternated optimization by gradient steps I suppose.
> > > >
> > > > However, you agree that max_D L_{adv}(G,D) is minimal when $G$ maxmizes the uncertainty of $D$, that is to say, when $D=0.5$ for any training sample. no ?   So theoretically I cannot see what prevents from this trivial generation that simply copies the M input...
> > > >
> > > > Lastly, I cannot see why VAE non identifiability of the latent Z would be problematic, while this problem would be miraculously solved in GAN approaches. You could see the parameters of your generator as a random variable $\Theta$ that is driven by $p(\Theta|X,Y,M)$. This variable, as Z for VAE approaches, is not identifiable, since the optimum is not unique. I understand that you can still converge to the true counterfactual when an infinity of data is observed, but sorry I cannot see why the decoder of a VAE couldn't as well...

---

### Official Review · Reviewer_5wUi · 2023-11-01

**Soundness:** 2 fair
**Presentation:** 2 fair
**Contribution:** 2 fair
**Rating:** 5
**Confidence:** 4

**Summary:**

The paper studies the issue of counterfactual fairness. It’s concerned with making predictions that are fair towards individuals, ensuring that the prediction would remain the same if the individual belonged to a different demographic group defined by sensitive attributes like gender or race. The proposed method use GAN for counterfactual mediator generation to ensure fairness while maintaining high prediction performance. Experiments are conducted to evaluate the method, including a real-world case study on recidivism prediction.

**Strengths:**

1. The paper addresses an important aspect of fairness in machine learning, ensuring that predictions are fair at an individual level.

2. The paper presentation includes rich contents, with tables and figures well organized. The writing is generally easy to follow.

3. The proposed method is validated through various experiments, including synthetic datasets and a real-world case study. The paper claims to achieve sota performance, and code is provided.

**Weaknesses:**

1. The theoretical analyses in 4.3 does not provide much insight or guarantee. The lemma states that the level of counterfactual fairness is upper-bounded by the performance of the counterfactual mediator generation and the counterfactual mediator regularization. This does not really takes an equation to be concluded, and it cannot say anything about whether the method would be effective or not.

2. Following the above point, the method does not have guarantee in achieving counterfactual fairness.

3. Though the authors compare this work with CFGAN, the use of GAN for generation-based counterfactual fairness makes the technical contributions largely alike, which impairs the the novelty of the proposed method.

**Questions:**

See weaknesses.

---

> ### Author Response · Authors · 2023-11-20
> **Response to Reviewer 5wUi**
>
> Thank you for your helpful review! We appreciate that you find our paper important, rich, and well-organized. We have carefully addressed your comments below to improve our work.
>
> ### Responses to "Weaknesses"
>
> * **Response to: New theoretical guarantees**
>
> Thank you for your suggestion. We have **entirely reworked our theoretical analyses**:
>
> We now added a new theoretical analysis to show our method’s effectiveness (see our **new Lemma 2**). Our new Lemma 2 states that our method correctly identifies the counterfactual mediators. Specifically, we prove that our method allows for **consistent estimation of the counterfactual distribution**, $\mathbb{P}(M_{a'} \mid X=x, A=a, M=m)$. Lemma 2 thus gives a guarantee on the correctness of the generated counterfactual mediators and, therefore, also a guarantee on the upper bound in Lemma 1, which thus confirms the effectiveness of our method.
>
> We added a new **Corollary 1** to show that the above theoretical results also **generalize to high-dimensional settings**.
> Our new theoretical analysis is important for two reasons. First, it provides a **theoretical justification behind the design of our method**. Second, it ensures that we correctly learn counterfactual fairness. To the best of our knowledge, ours is the **first** neural method for counterfactual fairness that offers such guarantees.
>
> * **Response to: Theoretical guarantee to show our method’s effectiveness**
>
> Thank you for your helpful comment. We added a **new Lemma 2** to give a guarantee on the error of the generated counterfactual ($||M_{A'} - \hat M_{A'}||^2_2$), which is the first term of upper bound terms in Lemma 1. Therefore, when the counterfactual mediator regularization ${R}_\mathrm{{cm}}$ (=the second term of the upper bound), is minimized, the counterfactual fairness level is guaranteed through Lemma1. As such, we achieve a **theoretical guarantee that our method is effective in achieving counterfactual fairness**. To the best of our knowledge, ours is the **first** neural method for counterfactual fairness that offers such guarantees.

---

> ### Author Response · Authors · 2023-11-20
> **Response to Reviewer 5wUi**
>
> ### Responses to "Weaknesses"
>
> * **Response to: Differences from CFGAN [1]**
>
> We appreciate the opportunity to clarify that our method is **significantly different** from CFGAN (in terms of task, architecture, learning objectives, etc.). We highlight these differences below:
>
> **Different tasks**:
>
> CFGAN is designed for fair data generation tasks, while our model is designed for learning predictors to be counterfactual fairness. Hence, both address **different tasks**.
>
> **Different architectures**:
>
> CFGAN employs **two** generators, each aimed at simulating the original causal model and the interventional model, and two discriminators, which ensure data utility and causal fairness. We only employ a streamlined architecture with a **single** generator and discriminator. Further, fairness enters both architectures at **different places**. In CFGAN, fairness is ensured through the GAN setup, whereas our method ensures fairness in a second step through our counterfactual mediator regularization.
>
> **Different training objectives**:
>
> The training objectives are **different**: CFGAN learns to **mimic factual data**. In our method, the generator **learns the counterfactual distribution of the mediator** through the discriminator distinguishing factual from counterfactual mediators.
>
> **No theoretical guarantee for CFGAN**:
>
> CFGAN is proposed to synthesize a dataset that satisfies counterfactual fairness. However, a recent paper [2] has shown that CFGAN is actually considering interventions (=level 2 in Pearl’s causality ladder) and **not** counterfactuals (=level 3). Hence, CFGAN does not fulfill the counterfactual fairness notion, but a different notion based on do-operator (intervention). For details, we refer to reference [2], Definition 5 therein, called ``Discrimination avoiding through causal reasoning''): A generator is said to be fair if the following equation holds: for any context $A=a$ and $X=x$, for all value of $y$ and $a' \in \mathcal{A}$, $P(Y=y \mid X=x, d o(A=a))=P\left(Y=y \mid X=x, d o\left(A=a' \right)\right)$, which is different from the counterfactual fairness
> $P(Y_a=y \mid X=x, A=a)=P\left(Y_{a'}=y \mid X=x, A=a\right)$. (We also discuss CFGAN considers interventions not counterfactual in our paper see **Appendix. I.1**) Moreover, **CFGAN lacks theoretical support** for its methodology (no identifiable guarantee or counterfactual fairness level). In contrast, our method strictly satisfies the principles of counterfactual fairness and provides theoretical guarantees on the counterfactual fairness level. In sum, **only our method offers theoretical guarantees** for the task at hand.
>
> **Suboptimal performance of CFGAN**:
>
> Even though CFGAN can, in principle, be applied to counterfactual fairness prediction, it is **suboptimal**. The reason is the following. Unlike CFGAN, which generates complete synthetic data under causal fairness notions, our method only generates counterfactuals of the mediator as an intermediate step, resulting in minimal information loss and better inference performance than CFGAN. Furthermore, since CFGAN needs to train the dual-generator and dual-discriminator together and optimize two adversarial losses, it is more difficult for stable training, and thus its method is less robust than ours.
>
> In sum, even though CFGAN also employs GANs, it is **vastly different from our method**.
>
> **Action**: We carefully discuss the differences between CFGAN and our method in our revised paper (see **Appendix B.2 and I.1**). Therein, we spell out clearly that CFGAN is different in terms of task, architecture, and theoretical guarantees.
>
> References
>
> [1] Depeng Xu, Yongkai Wu, Shuhan Yuan, Lu Zhang, and Xintao Wu. Achieving causal fairness through generative adversarial networks. In IJCAI, 2019.
>
> [2] Mahed Abroshan, Mohammad Mahdi Khalili, and Andrew Elliott. Counterfactual fairness in synthetic data generation. In Neurips 2022 SyntheticData4ML Research, 2022.

---

### Official Review · Reviewer_Aquc · 2023-11-01

**Soundness:** 1 poor
**Presentation:** 3 good
**Contribution:** 2 fair
**Rating:** 5
**Confidence:** 5

**Summary:**

This work proposes to use GAN to estimate counterfactual mediators. Although the method is shown to have strong empirical performance, there is no theoretical guarantee on the error of estimated counterfactual mediators, which can lead to arbitrarily unfair predictors given the proposed regularization relies on the generated counterfactual mediators.

**Strengths:**

- Empirically, the authors show, in some synthetic/semi-synthetic settings, their method can magically infer counterfactual mediators by learning from observational data. In addition, their method outperforms various baselines from the counterfactual fairness literature.
- Paper is well written, it is easy to read.

**Weaknesses:**

- There is no discussion on the identification of counterfactual mediators. The authors need to show this before developing estimators for counterfactual mediators.
- The generator is trained on factual data only, there is no guarantee on the error of its generated counterfactuals. This would further lead to ineffectiveness of the discriminator as it is trained on generated counterfactuals. Similarly, In Lemma 1, the upperbound has a term ||M_{A'},\hat{M}_{A'}|| which is not computable, so, the minimizing the loss function Eq.(8) cannot guarantee the LHS of Eq.(9) is minimized.

**Questions:**

- In the literature, there are works on issues of methods that learn a ML model to predict counterfactuals for counterfactual fairness [1,2]. [1] points out that one can learn a model to predict counterfactuals iff a specific strong ignorability holds, i.e., A is independent of potential mediators M_a. In another words, the dataset is collected without selection bias, which is represented by a collider S \in \{0,1\} s.t. A->M, A->S, M->S and we can only observe samples from P(A,M|S=1). [2] argues counterfactual fairness is similar to demographic parity as (1) any predictor satisfying counterfactual fairness also satisfies DP and (2) any predictor satisfying DP can be modified trivially to satisfy counterfactual fairness. The authors may want to add a discussion about them.

- It is vague in Fig. 1 that whether the authors allow just correlation between X and A or there has to be a causal relationship X->A.

- The claim that one can include anything in the mediator if the domain knowledge is missing sounds not solid.

- Inconsistent notations: If M_a is the potential outcome, why do we need M_{A-\leftarrow a}?

- Parenthesis mismatch in Eq.(9).

- For Adult, how do the authors know marital status, education level, occupation, hours per week, and work class are mediators? How is the counterfactual fairness metric computed without knowing ground truth counterfactuals?

- [1] Fawkes, Jake, Robin Evans, and Dino Sejdinovic. "Selection, ignorability and challenges with causal fairness." In Conference on Causal Learning and Reasoning, pp. 275-289. PMLR, 2022.
- [2] Rosenblatt, Lucas, and R. Teal Witter. "Counterfactual fairness is basically demographic parity." In Proceedings of the AAAI Conference on Artificial Intelligence, vol. 37, no. 12, pp. 14461-14469. 2023.

---

> ### Author Response · Authors · 2023-11-19
> **Response to Reviewer Aquc**
>
> Thank you for your thorough and insightful feedback! We are glad to hear that you appreciate the strong empirical performance and the clear presentation of our paper.
>
>
> ### Response to Weaknesses
>
> **Identification of counterfactual mediators**
>
> Thank you. We recognize the importance of discussing the identification of counterfactual mediators. To address your point, we have added a **new theoretical analysis**. Specifically, we have revised our manuscript in the following ways:
> * We added a **new Lemma 2** to show that our method correctly identifies the counterfactual mediators. Specifically, we prove that our method allows for **consistent estimation of the counterfactual distribution**. The lemma thus gives a guarantee on the correctness of the generated counterfactual mediators and, therefore, also a guarantee on the upper bound in Lemma 1.
>
> * We added a new **Corollary 1** to show that the above theoretical results also **generalize to high-dimensional settings**.
> Our new theoretical analysis is important for two reasons. First, it provides a **theoretical justification behind the design of our method**. Second, it ensures that we correctly learn counterfactual fairness. To the best of our knowledge, ours is the **first** neural method that offers such guarantees.
>
> **Theoretical Guarantee and Counterfactual Estimation**
>
> Thank you. We address your comment point-by-point。
>
> *Response to: No theoretical guarantee*
>
> To address your point, we now provide **new theoretical guarantees** of the correctness of the generated counterfactual mediators (see our **new Lemma2**). Our new Lemma 2 shows that our generator consistently estimates the counterfactual distribution of the mediator. It gives a guarantee on the correctness of the generated counterfactual mediators. Together with Lemma 1, it provides theoretical results that our method is effective.
>
> *Response to: Correctness of the generated counterfactual mediator*
>
> Our **new Lemma 2** states that our GAN-based approach consistently estimates the counterfactual distribution of mediators. By converging to the counterfactual distribution (point mass distribution), our generator actually learns a deterministic function that transforms from factual to counterfactual: $\mathbb{P}(M_{a'} \mid X=x, A=a, M=m)$. Under consistency, the generated counterfactual can converge to the ground-truth counterfactual.  This confirms the effectiveness and validity of the generated counterfactual mediators.
>
> *Response to: Addressing the upper bound in Lemma 1*
>
> We acknowledge that Lemma 1 was not sufficient to show the effectiveness of our method. As a remedy, our new Lemma 2 gives a guarantee on the error of the generated counterfactual ($||M_{A'} - \hat M_{A'}||^2_2$), which is the first term of upper bound terms in Lemma 1. Then, by minimizing the loss function ${L}(h)=L_{ce}(h)+\lambda R_{cm}(h)$
> , the counterfactual mediator regularization $R_{cm}$ is minimized accordingly, which is the second term in the of upper bound terms in Lemma 1. As a result, we can successfully address your comment: by minimizing the loss, we can guarantee that the counterfactual fairness level is minimized.

---

> ### Author Response · Authors · 2023-11-19
> **Response to Reviewer Aquc**
>
> ### Response to Questions
>
> * **Discussion regarding the literature [1] & [2]:**
>
> Thank you. We appreciate the opportunity to spell out how our work is **different** from [1] and [2]. We also have **included references** to papers [1] & [2] in our revised paper together with a brief discussion. Both [1] & [2] make important contributions to our theoretical understanding of causal fairness. As such, they are **orthogonal** to our work where we develop a new learning algorithm for counterfactual fair predictions with theoretical guarantees.
>
> Regarding [1]:
>
> The setting of the paper [1] is similar to ours, only some notations are different. In [1], what is referred to as $X(a)$ aligns with $M_a$ in our paper. However, there are **several differences in the assumptions** that distinguish [1] from our paper.
>
> 1. Confounder inclusion:
>
> Our model allows for the existence of the confounders between the sensitive attribute $A$ and its descendant $M$, denoted as $X$ in our paper. This is **unlike** [1], where such variables are not explicitly considered, so that our setting can be considered to be more general, For example, consider Figure 1 in reference [1]. Therein, the authors aim to handle additional variables (“covariates”) that are not the sensitive attribute but also can affect the mediators like "Age”. The “Age” variable, even when there are no other sensitive attributes such as “race” or “gender” there, still impacts mediators like “GPA”, “LSAT”, and “FYA”. However, such variables are **not** considered in [1] setting. In contrast, we can put such variables into our covariate $X$. Therefore, we argue that our setting is applicable to a wider range of scenarios.
>
> 2. Selection bias:
> If there exists a collider $S$ between $A$ and $M$ (such that $A \rightarrow M$, $A \rightarrow S$, $M \rightarrow S$), then conditioning on $S$ introduces a correlation between $A$ and $M$. This can be interpreted as the presence of a confounder (such that $A \rightarrow M$, $X \rightarrow A$, $X \rightarrow M$). In [1], the dataset is collected **without** selection bias, which implies that $A$ and $M$ are unconfounded. This is consistent with our assumptions. In [1], structural counterfactuals satisfy ignorability ($ M_a \perp\kern-5pt\perp A $). In our work, since we allow for covariates $X$, the potential outcome $M_a$ is conditionally independent of $A$ given the covariates $X$  ($M_a \perp\kern-5pt\perp A | X$) .
>
> Regarding [2]:
>
> We highly appreciate the theoretical insights from [2], yet there are significant **differences** in our work.
> DP is a non-causal fairness notion. Specifically, DP says the target $Y$ should satisfy the conditional distribution $P(Y \mid A=a)=P(Y \mid A=a’)$, and, thereby, it essentially prohibits the use of any variable correlated with the sensitive attribute $A$ for predictions. This includes $A$ itself, its descendants, causes, and any correlated variables. As we discuss below, prohibiting the use of any variable correlated with the sensitive attribute may be fairly restrictive in practice.
>
> In our causal graph, to achieve fairness under DP, one must exclude not only the sensitive attribute $A$ and mediators $M$ but also any variables in $X$ that directly cause or correlate with A. For instance, consider an employment system in a company. If ‘'gender' is the sensitive attribute and 'age' correlates with gender due to regional cultural norms, DP would forbid the use of 'age' for fair decision-making. Our method, however, allows for such usage.
>
> Consequently, achieving DP often results in a significant loss of predictive accuracy. In contrast, our method ensures counterfactual fairness, while not being bound by these strict restrictions, so that our method can achieve a better prediction performance. We performed a few tests using datasets where $X$ and $A$ are uncorrelated. We used the same classifier for better comparability. A method that needs to achieve DP only records an accuracy of 0.76 (i.e., by only using $X$). In contrast, our method achieves an accuracy 0.94 (by allowing the use of $X$ and $M$). In summary, DP has more strict requirements for using variables, thus sacrificing the accuracy of predictions. This is especially relevant when $M$ contains a lot of useful information for prediction. We thus believe that our method is therefore of great value in practice where not only counterfactual fairness but also a good prediction performance is important.
>
> Reference
>
> [1] Fawkes, Jake, Robin Evans, and Dino Sejdinovic. "Selection, ignorability and challenges with causal fairness." In Conference on Causal Learning and Reasoning. PMLR, 2022.
>
> [2] Rosenblatt, Lucas, and R. Teal Witter. "Counterfactual fairness is basically demographic parity." In AAAI. 2023.

---

> ### Author Response · Authors · 2023-11-20
> **Response to Reviewer Aquc**
>
> ### Response to Questions
>
>
> * **Fig.1 Interpretation:**
>
>  The dashed line in Fig.1 illustrates that a correlation between $X $ and $A$ is permissible/allowed in our framework, but a direct causal relationship $X$->$A$ is not a necessity. Note that, if there is no dashed edge between $X$ and $A$, it is actually a stronger assumption, because it forbids the edge between $X$ and $ A$ to have any hidden confounders. However, our setting is more general and allows the existence of the confounders. Notwithstanding, we added new experimental results (see new results in Appendix H.1), where we analyze the performance of our method in the presence of confounders, finding that our method is highly effective.
>
> * **Choice of mediator variables:**
>
>  The selection of mediator variables, $M$, typically relies on domain knowledge, following established practices in literature such as [3, 4, 5, 6, 7]. We fixed that sentence in our revised paper.
>
> * **Inconsistent notation:**
>
> Thank you! The notation $M_{A-\leftarrow a}$ was an error. We have streamlined our notation so that we now consistently use $M_a$ as the potential outcome in the updated manuscript.
>
> * **Parenthesis mismatch in Eq. (9)**:
>
> Thank you. We fixed this.
>
> * **Adult dataset:**
>
> Thank you for your suggestions. We added a more detailed discussion of our setting to **Appendix E.3**. Below, we discuss how we selected mediators and the counterfactual fairness metric.
>
> *Mediator selection*:
>
> We follow common choices in prior research (such as [3, 4, 5, 6]). We consider 'gender' as a sensitive attribute. Mediators are variables that can potentially be influenced by the sensitive attribute. The  'marital status', 'education level', 'occupation', 'hours per week', and 'work class' are all known to be influenced by ‘gender’, and, therefore, we treat them as ‘mediators’. We classified all other variables as covariates.
>
> *Counterfactual fairness metric*:
>
> We use the Adult dataset to demonstrate the applicability of our method to real-world data, where ground-truth counterfactuals are unavailable. Nevertheless, to understand the implications for counterfactual fairness empirically, we use the generated counterfactuals to measure counterfactual fairness on the test dataset. This allows us to explore the accuracy-fairness trade-off. We have revised our paper to spell out clearly how we computer the counterfactual fairness metric for real-world data. Thereby, we further highlight that our choice is in line with other works from causal fairness as the purpose of the Adult dataset is to demonstrate real-world applicability (and not benchmarking).
>
>
> Reference
>
> [3] Razieh Nabi and Ilya Shpitser. Fair inference on outcomes. In AAAI, 2018.
>
> [4] Hyemi Kim, Seungjae Shin, JoonHo Jang, Kyungwoo Song, Weonyoung Joo, Wanmo Kang, and Il.Chul Moon. Counterfactual fairness with disentangled causal effect variational autoencoder. In AAAI, 2021.
>
> [5] Depeng Xu, Yongkai Wu, Shuhan Yuan, Lu Zhang, and Xintao Wu. Achieving causal fairness through generative adversarial networks. In IJCAI, 2019.
>
> [6] Francesco Quinzan, Cecilia Casolo, Krikamol Muandet, Niki Kilbertus, and Yucen Luo. Learning counterfactually invariant predictors. arXiv preprint, 2022.
>
> [7] Drago Plecko and Elias Bareinboim. Causal fairness analysis. In arXiv preprint, 2022.

---

> ### Comment · Reviewer_Aquc · 2023-11-21
> **Lemma 2**
>
> Thanks for the revision. I updated my score. But I still have some questions.
>
> 1. Can the authors make it clear whether the first statement of Lemma 2 is their contribution or from existing work? With statement 1, the counterfactual distribution is identified. Then it is not surprising we can fit a GAN or other generative models to estimate the counterfactual mediators.
>
> 2. Intuitively, I still think the proposed GAN can only generate good factual data as it is never trained on counterfactual data just like the original GAN can only generate good data from the original training distribution. This implies that the strategy of this work is to mak the distribution of counterfactual mediators equivalent to its corresponding conditional distribution with a different value of $A$. With Lemma 2, it basically says, if the conditional $P(M|X=x,A=a')$ can be estimated precisely, then the counterfactual $P(M_{a\leftarrow a'}|X=x,A=a)$ can be estimated accurately because they are the same thing under the assumptions. In this case, my question is, if we can observe enough samples with $(X=x,A=a')$, why is $M_{a\leftarrow a'}|X=x,A=a$ still a counterfactual quantity?

---

> > ### Author Response · Authors · 2023-11-22
> > **Response to Reviewer Aquc about Lemma2**
> >
> > Thank you very much! We appreciate your help in improving the paper and your constructive feedback!
> >
> > To your questions:
> >
> > Thank you. Yes, the entire **Lemma 2 is our contribution** (and **not** from existing work). This is a **new theoretical result** regarding counterfactual identifiability with GANs. To the best of our knowledge, we are the first to give such a theoretical guarantee on the generated counterfactual with GANs.
> >
> >
> > Thank you again for your question. Lemma 2 basically says: “if the conditional $P(M|X=x,A=a’)$ can be estimated precisely, then the counterfactual $P(M_{a\leftarrow a’}|X=x,A=a)$ can be estimated accurately.” Let us break down that:
> >
> > First, a counterfactual quantity is some probabilistic query that contains a logical contradiction in its formulation (e.g., $M_{a\leftarrow a’}|X=x,A=a$ has a contradiction between observed a and intervened $a’$, i.e., we cannot simultaneously condition on something and intervene on it at the same time).
> >
> > Second,  we would like to emphasize that accurately estimating conditional distributions is a necessary (but not sufficient) condition for the counterfactual inference. The sufficiency in our case is guaranteed by a specific probabilistic model (i.e., non-random continuously-differentiable probabilistic model (GAN generator) with factual $M$ as input) and the assumption of the BGM.  The connections between level 1, level 2, and level 3 quantities in terms of Pearl’s causality ladder can be then seen from the following equalities: $\underbrace{P(M |X=x,A=a’)}\_{L_1} \stackrel{(1)}{=} \underbrace{P(M_{a’} |X=x)}\_{L_2} \stackrel{(2)}{=} \underbrace{P(M_{a’} |X=x, A = a)}\_{L_3} = \int_{m \in \mathcal{M}} \underbrace{P(M_{a’}|X=x,A=a,M=m)}\_{L\_3} \, \underbrace{P(M=m | X=x,A=a)}\_{L_1} dm $, where (1) follows from the consistency assumption,  (2) follows from the ignorability assumption, and the whole equation has only two solutions wrt. $P(M\_{a’}|X=x,A=a,M=m)$ if we either assume a non-random continously-differentiable probabilistic model (GAN generator) with factual $M$ as input or BGMs.
> >
> > Importantly, we would like to emphasize that our new Lemma 2 provides theoretical guarantees that our GAN can even generate correct counterfactual mediators (even if it was trained only on factual values from our setting).
> >
> > **Action:** We added a statement to our paper that Lemma 2 is our contribution (and **not** from existing work).
> >
> > We are again thankful for your question. We hope that our answer helped in addressing your question satisfactorily. Please let us know if there are further things in our presentation that should be improved.

---

> > > ### Comment · Reviewer_Aquc · 2023-11-22
> > > **Continue on Lemma 2**
> > >
> > > Thanks for the reply.
> > >
> > > I do not agree that the first statement of Lemma 2 is related to GAN.
> > > Could the authors explain what is the difference between their first statement in Lemma 2 and the Lemma B.2 in (Nasr-Esfahany et al., 2023) and Corollary 3 in (Melnychuk et al.)? Since there is no complete proof of this statement in Appendix C.2 and the RHS of Eq.(21) is from (Melnychuk et al.).

---

> > > > ### Author Response · Authors · 2023-11-22
> > > > **Response to Reviewer Aquc (Continue on Lemma2)**
> > > >
> > > > Thank you for reading our paper so carefully and thoroughly, and for asking detailed questions about our Lemma 2!
> > > >
> > > >
> > > > About the first statement of Lemma 2:
> > > >
> > > > You are right that the first statement of Lemma 2 is a more general statement, and we properly cited the papers regarding the bijective generation mechanism (BGM) [1, 2]. The difference between the first statement of Lemma 2 and the statements from the Lemma B.2 in (Nasr-Esfahany et al., 2023) and Corollary 3 in (Melnychuk et al.) is that we **additionally condition on the covariates $X = x$**.
> > > >
> > > > The first statement of Lemma 2 thus says that, in the class of continuously differentiable functions, there are only two solutions wrt. possible counterfactual distributions. Notably, in the more general class of all measurable functions, there are infinitely many solutions. Because generators of our GAN mimic a class of continuously differentiable functions, the only possible solutions coincide with the solutions of the BGMs.
> > > >
> > > > The second statement of Lemma 2 is **our main contribution to theory**. It is the first formal connection between BGMs and GANs, and this connection is not trivial. In this proof, we combined the properties of GAN, our three causal assumptions, and the BGM assumption.
> > > >
> > > > The whole Lemma 2 is thus a **new theoretical result** regarding counterfactual identifiability with GANs. **To the best of our knowledge, we are the first to give such a theoretical guarantee on the generated counterfactual with GANs.**
> > > >
> > > > Reference:
> > > >
> > > > [1] Arash Nasr-Esfahany, Mohammad Alizadeh, and Devavrat Shah. Counterfactual identifiability of bijective causal models. In ICML, 2023.
> > > >
> > > > [2] Valentyn Melnychuk, Dennis Frauen, and Stefan Feuerriegel. Partial counterfactual identification of continuous outcomes with a curvature sensitivity model. In NeurIPS, 2023.

---

### Author Response · Authors · 2023-11-19
**Response to all reviewers**

Thank you very much for the constructive evaluation of our paper and your helpful comments! We addressed all of them in the comments below.



Our **main improvements** are the following:

* We provided a **new theoretical analysis** (=> see our **new Lemma 2**, our **new Remark 1**, our **new Corollary 1**, and our **new Remark 2**). Therein, we prove that our method allows for **consistent estimation of the counterfactual distribution**. This is important for two reasons. First, it provides a **theoretical justification behind the design** of our GAN method. Second, it ensures that we correctly learn counterfactual fairness. To the best of our knowledge, ours is the **first** neural method for counterfactual fairness that offers such guarantees.

* We show how our method can **generalize to multiple social groups** (i.e., high-dimensional features). To this end, we added **new experimental results** with multiple categories for the sensitive attribute (see our new **Appendix G**).  We also provide theoretical proof (see our **new Corollary 1**) where we show that our theoretical results also generalize to high-dimensional settings.

* We clarified the advantage of our GAN framework over baselines with latent variables (see our changes in **Sections 1, 2, and 3**, as well as our new materials in **Appendix B.1**)

* We added new experimental results (see new results in **Appendix H**). For example, we now analyze the performance of our method in the presence of confounders, finding that our method is highly effective (in comparison to the baselines).


We uploaded a revised version of our paper, where we highlight key changes colored in **red color**. Given these improvements, we are confident that our paper will be a valuable contribution to the causal machine learning literature and a good fit for ICLR 2024.

---

### Meta-Review · Area_Chair_JqyP · 2023-12-07

**Metareview:**

The use of GANs for problems of fair predictions using counterfactual measures is discussed in the paper. There is definitely a fair amount of work done here, but I feel as is the paper does not properly address to which extent this model is a convincing model for cross-world counterfactuals. I'm aware that sometimes strong assumptions are made about shape of structural equations (such as structural additive error models) that could be useful to illustrate conceptual ideas. However, as we move towards more and more contributions targeting engineering aspects such as adapting GANs maybe we should go back to addressing more directly how different aspects of cross-world distributions described in the manuscript fit the literature at large. The paper appears to suggest in Lemma 2 that any latent variable model can somehow be interpreted as cross-world counterfactual models, which is not a safe take on the topic.

Also, I believe the numerous points raised by the reviewers will be useful in a future revision of the paper.

**Justification For Why Not Higher Score:**

The GAN aspect of it is interesting, although it is not too surprising by itself. The main point of linking latent variable models to sensible causal cross-world models, as in Lemma 2, appears not to consider seriously enough the untestable aspects of this leap, as if a GAN could fix it or be fundamental to it.

**Justification For Why Not Lower Score:**

N/A

---

### Decision · Program_Chairs · 2024-01-16

Reject